# A Stochastic Linearized Augmented Lagrangian Method for Decentralized Bilevel Optimization

**Songtao Lu**[†]    **Siliang Zeng**[‡]    **Xiaodong Cui**[†]    **Mark S. Squillante**[†]
**Lior Horesh**[†]    **Brian Kingsbury**[†]    **Jia Liu**[*]    **Mingyi Hong**[‡]
[†]IBM Research, Thomas J. Watson Research Center, Yorktown Heights, NY 10598
`songtao@ibm.com,{cuix,mss,lhoresh,bedk}@us.ibm.com`
[‡]Dept. of Electrical and Computer Engineering, University of Minnesota, Minneapolis, MN 55455
`{zeng0176,mhong}@umn.edu`
[*]Dept. of Electrical and Computer Engineering, The Ohio State University, Columbus, OH 43210
`liu@ece.osu.edu`

## Abstract

Bilevel optimization has been shown to be a powerful framework for formulating multi-task machine learning problems, e.g., reinforcement learning (RL) and meta-learning, where the decision variables are coupled in both levels of the minimization problems. In practice, the learning tasks would be located at different computing resource environments, and thus there is a need for deploying a decentralized training framework to implement multi-agent and multi-task learning. We develop a stochastic linearized augmented Lagrangian method (SLAM) for solving general nonconvex bilevel optimization problems over a graph, where both upper and lower optimization variables are able to achieve a consensus. We also establish that the theoretical convergence rate of the proposed SLAM to the Karush-Kuhn-Tucker (KKT) points of this class of problems is on the same order as the one achieved by the classical distributed stochastic gradient descent for only single-level nonconvex minimization problems. Numerical results tested on multi-agent RL problems showcase the superiority of SLAM compared with the benchmarks.

## 1 Introduction

In this paper, we consider the following general decentralized bilevel optimization (DBO) framework with applications to machine learning problems. Suppose that there are $n$ nodes over a connected graph $\mathcal{G} = \{\mathcal{E}, \mathcal{V}\}$, where $\mathcal{E}$ and $\mathcal{V}$ represent the edges and vertices. Let $\mathcal{N}_i$ denote the set of neighboring nodes for node $i$. Then the goal of DBO is to have these nodes jointly minimize two levels of optimization problems. More formally, DBO is expressed as

$$\min_{\mathbf{x}_1,\ldots,\mathbf{x}_n} \quad \frac{1}{n}\sum_{i=1}^{n} f_i(\mathbf{x}_i, \mathbf{y}^*_{i,1}(\mathbf{x}_i), \ldots, \mathbf{y}^*_{i,m}(\mathbf{x}_i)) \tag{1a}$$

$$\text{s.t.} \quad \mathbf{x}_i = \mathbf{x}_j, j \in \mathcal{N}_i, \forall i \in [n] \tag{1b}$$

$$\mathbf{y}^*_k(\mathbf{x}) = \arg\min_{\mathbf{y}_{1,k},\ldots,\mathbf{y}_{n,k}} \frac{1}{n}\sum_{i=1}^{n} g_{i,k}(\mathbf{x}_i, \mathbf{y}_{i,k}) \ \text{ s.t. } \ \mathbf{y}_{i,k} = \mathbf{y}_{j,k}, \ j \in \mathcal{N}_i, \forall k \in [m], \tag{1c}$$

where vector $\mathbf{x}_i$ is the upper level (UL) optimization variable at each node $i$, vector $\mathbf{y}_{i,k}$ denotes the lower level (LL) decision variable for the $k$th learning task at node $i$, $f_i(;)$ is a (smooth) UL loss function and possibly nonconvex with respect to (w.r.t.) both the UL and LL variables, $g_{i,k}(,)$ denotes the LL objective function of the $k$th task at node $i$, $m$ represents the total number of LL

36th Conference on Neural Information Processing Systems (NeurIPS 2022).

optimization problems, the consensus constraints $\mathbf{x}_i = \mathbf{x}_j, \mathbf{y}_{i,k} = \mathbf{y}_{j,k}, j \in \mathcal{N}_i, \forall i \in [n], \forall k \in [m]$, enforce the model agreements at each level of the problems and for each LL learning task, and $\mathbf{y}_k^* = [\mathbf{y}_{1,k}^*, \dots, \mathbf{y}_{n,k}^*]^T$ is the optimal solutions of the $k$th LL problem under the consensus constraints.

**Applications of Bilevel Optimization.** Many machine learning problems can be formulated mathematically as a form of bilevel optimization or, more precisely, a special case of problem (1), e.g., meta-learning or meta reinforcement learning (RL), actor-critic (AC) schemes in RL, hyperparameter optimization (HPO), and so on.

Classical bilevel optimization is referred to as the case where there is no consensus constraint but with only two levels of the minimization subproblems, i.e., $\min_{\mathbf{x}} f(\mathbf{x}, \mathbf{y}^*(\mathbf{x}))$, s.t. $\mathbf{y}^*(\mathbf{x}) = \arg\min_{\mathbf{y}} g(\mathbf{x}, \mathbf{y})$, which is also known as Stackelberg games [1] with the UL decision variable as the leader and the LL decision variable as the follower. It turns out that this class of optimization problems is useful in formulating a wide range of hierarchical or nested structured machine learning problems. For example, one of the most popular domain adaption learning models, model-agnostic meta-learning (MAML) [2, 3], can be written as a special case of bilevel programming [4], where the UL model provides a good initialization for accelerating learning procedures by implementing the LL algorithms. The idea behind the model design is that the UL model is considered as the meta learner that searches for a permutation-invariant subspace over multiple task-specific learners at the LL so that the performance of the MAML model can be generalized well for unseen or testing data samples. The theoretical analysis of the generalization performance of this class of bilevel problems has shown that MAML can indeed decrease the generalization error as the number of tasks increases, at least for strongly convex loss functions [5]. Subsequently, a thorough ablation study from the latent representation perspective shows that feature reuse is the actual dominant factor in improving the generalization performance of MAML [6], and the authors propose a neural network-oriented algorithm with almost no inner loop (ANIL) that splits the neural network parameters into two parts corresponding to the UL and LL optimization problems, respectively. Extensive numerical experiments illustrate that ANIL achieves almost the same accuracy as the classical MAML but with significant computational savings. This example further strengthens the necessity of variable splitting in the learning structure by optimizing two levels of objective functions to enhance the generalization performance. Beyond the traditional supervised meta-learning scenarios, MAML has also been applied to increasing the generalization ability of agents in RL problems by replacing the (stochastic) gradient with the (natural) policy gradient (PG) [3] under the same two-level structure.

Besides meta-learning problems, AC structure in RL is another class of common learning frameworks that can be formulated by a bilevel optimization problem in nature [7, 8, 9], where the actor step at the UL aims at optimizing the policy while the critic step at the LL is responsible for value function evaluation. In addition, as the expressiveness of neural networks increased sharply over the past decades, the reuse of large models with adaptation to multi-task learning problems presents promising solutions by leveraging the pre-train and fine-tune strategy, such as in applications of HPO [10, 11] where the hyperparameters are trained at the UL problem so that the downstream learning tasks are learned with low costs including the expense of both computation and memory.

**Applications of Multi-agent Settings.** When multiple computational resources are available and connected, it is well motivated that exploring them solves distributed large-scale problems with a reduced amount of training time or performs multi-task learning. The bilevel structure of the meta-learning (ML) is a good fit in this scenario as either UL/LL or both levels may need to access the networked data samples rather than local ones. For example, a federated learning setting of MAML [12] and bilevel optimization [13] have been built up over multiple nodes recently, where the meta/UL learner finds an initial shared model while the local/LL learners leverage it for adapting data distributions of individual users. In such a way, the federated MAML model can realize personalized learning without sharing heterogeneous data over numerous clients. Once there is no central controller for coordinating the model aggregation, a diffusion-based MAML (Dif-MAML) [14] is proposed by spreading the model parameters over a network, where the UL parameter is updated by one step of stochastic gradient descent (SGD) based on a combination of the parameters of neighbors as the initialization for local model updates.

Decentralized hierarchical structured learning is even more stringent in the multi-agent RL (MARL) setting [15] as the learning tasks are essentially located at scattered sensors and/or controllers. Under this setting, MARL problem becomes a multi-objective optimization problem under provided (approximate) value functions, where the policy of each agent needs to be learned locally by certain

efficient iterative methods, such as multi-agent deep deterministic policy gradient (MADDPG) [16], trust region methods [17], optimal baseline based variance reduced policy gradient [18], and/or improved by more advanced techniques, e.g., constrained policy optimization [19] and large sequence models [20]. In such a way, the total reward can be maximized over the distributed agents through optimizing the networked policy. In a fully collaborative setting, the team-based value function is even required to be shared over all the agents such that each agent is able to improve its policy based on the estimated total reward. For example, the decentralized AC (DAC) scheme has been investigated widely [15, 21, 22], where each agent uses the actor step to optimize its policy while the critic step performs one step [23] or multiple steps of temporal difference learning with mini-batch sampling (MDAC) [22, 24] and communications so that the team-based reward over the network is obtained by each agent. It turns out that DAC can be formulated as a special case of problem (1) as there is no consensus at the UL. Recently, it has been revealed that if there exists homogeneity of the state and action spaces, decentralized policy consensus (or a partial policy parameter sharing strategy) provides significant merits to the centralized training and decentralized execution paradigm in terms of learning scalability and efficiency[23, 25], which motivates the consensus process at both UL and LL DBO problems.

**Related Theoretical Works.** Given the fruitful results across these many applications, the corresponding theoretical analysis has been developing very fast as well for variants of bilevel optimization problems. For example, the convergence behaviors of classical inexact MAML (iMAML) methods have been quantified for both convex [26, 27] and nonconvex [28] cases of the UL loss function, where the LL algorithm only performs one step of stochastic gradient descent (SGD) based on the LL objective functions as the adaptation step. Moreover, the iteration complexity of ANIL with multiple iterations for minimizing the LL problems have been studied in [29], which justifies the significant computational advantages of ANIL compared with MAML in theory. Furthermore, the finite-time analysis of AC algorithms has shown [30] that, once the learning rates at both the actor and critic sides are chosen properly, a two timescale AC algorithm can achieve an $\mathcal{O}(\epsilon^{-2.5})$ iteration complexity for finding the first-order stationary points (FOSPs) of general nonconvex reward functions.

Besides these theoretical analyses in a specific learning setting, the algorithm design and corresponding convergence analysis for general bilevel optimization solvers have been recently advancing at a rapid pace under certain assumptions that the UL objective function is general nonconvex while the LL objective functions are strongly convex, which covers the existing convergence results shown for AC algorithms. The typical algorithms include those with double-loop structure, those with two timescale or single timescale but single-loop, and those with error-correction or accelerated/variance-reduction. To be more specific, double-loop algorithms, such as bilevel stochastic approximation (BSA) methods [31] and stochastic bilevel optimizers (stoBiO) [32], mainly request an inner loop to solve the LL problem up to a certain error tolerance or with a certain number of iterations and then switch back to optimize the UL problem, which can achieve an $\mathcal{O}(\epsilon^{-2})$ convergence rate to the $\epsilon$-FOSPs. In practice, single-loop algorithms are implemented more efficiently in terms of computational complexity and hyperparameter tuning compared to double-loop algorithms. A two-timescale stochastic approximation (TTSA) was analyzed in [33], but it is shown that TTSA needs $\mathcal{O}(\epsilon^{-2.5})$ number of iterations to achieve the $\epsilon$-FOSPs. Later, an error correction method, named the Single-Timescale stochAstic BiLevEl optimization (STABLE) method [34], improves the convergence rate of the single-loop algorithm to $\mathcal{O}(\epsilon^{-2})$ and a tighter analysis for ALternating Stochastic gradient dEscenT (ALSET) shows that the single-loop algorithm can also achieve a convergence of $\mathcal{O}(\epsilon^{-2})$ without the error correction technique. When more advanced momentum-assisted or variance reduction methods are adopted in the algorithm design, the subsequent works, such as the momentum-based recursive bilevel optimizer (MRBO) [35] and the single-timescale double-momentum stochastic approximation (SUSTAIN) [36] and the variance reduced BiAdam (VR-BiAdam) [37], can sharpen the convergence rate of bilevel algorithms to $\mathcal{O}(\epsilon^{-1.5})$.

For the theoretical works on MAML/MARL, it is shown in [22, 24] that when the critic side is allowed the consensus step at each agent to approximate the networked rewards, MDAC algorithms can achieve an $\mathcal{O}(\epsilon^{-2})$ convergence rate to FOSPs, but both of them require an inner loop procedure for the LL problem which makes the algorithms double loop. Dif-MAML [14] is able to perform the UL consensus-based meta learning, but iMAML considered in Dif-MAML is only a very special case of bilevel. Thus, the applicability of Dif-MAML is restrictive. One of the closest works to ours is coordinated AC (CAC) [23], which can realize the consensus on both UL and LL problems with $\mathcal{O}(\epsilon^{-2.5})$ number of iterations and is only for DAC problems. A theoretical comparison between our

Table 1: A comparison with closely related prior work on (decentralized) bilevel optimization learning. "Comm." refer to whether the algorithm only needs one round of communication at either UL or LL per iteration; "Alg." refs to the types of the basic stochastic algorithms adopted in the method.

| Prior work | Consensus UL | LL | Method | Rate | Comm. | Alg. | Setting |
|---|---|---|---|---|---|---|---|
| Ghadimi et al. [31] | | | BSA | $\mathcal{O}(1/\epsilon^2)$ | - | SGD | bilevel |
| Hong et al.[33] | | | TTSA | $\mathcal{O}(1/\epsilon^{2.5})$ | - | SGD | bilevel |
| Chen et al. [43] | | | ALSET | $\mathcal{O}(1/\epsilon^2)$ | - | SGD | bilevel |
| Khanduri et al. [36] | | | SUSTAIN | $\mathcal{O}(1/\epsilon^{1.5})$ | - | Momentum | bilevel |
| Kayaalp et al. [14] | ✓ | | Dif-MAML | $\mathcal{O}(1/\epsilon^2)$ | ✓ | SGD | iMAML |
| Kaiqing et al. [15] | | ✓ | DAC | - | ✓ | PG | MARL |
| Chen et al. [22] | | ✓ | MDAC | $\mathcal{O}(1/\epsilon^2)$ | | PG | MARL |
| Hairi et al. [24] | | ✓ | MDAC | $\mathcal{O}(1/\epsilon^2)$ | | PG | MARL |
| Zeng et al. [23] | ✓ | ✓ | CAC | $\mathcal{O}(1/\epsilon^{2.5})$ | ✓ | PG | MARL |
| **This work** | ✓ | ✓ | **SLAM** | $\mathcal{O}(1/(n\epsilon^2))$ | ✓ | SGD/PG | bilevel |

work and closely related previous works on bilevel programming is shown in Table 1. There is a line of independent work on decentralized optimization. But the existing works are only suitable for single-level minimization of only nonconvex problems, such as distributed SGD [38, 39], stochastic gradient tracking [40, 41] and stochastic primal dual algorithm [42], which can achieve an $\mathcal{O}(1/(n\epsilon^2))$ convergence rate to FOSPs for general nonconvex objective function optimization problems.

**Main Contributions of This Work.** In this work, we consider a very general DBO setting, where both UL and LL problems can include a consensus constraint for model parameter sharing and there would be multiple LL problems coupled with the UL problem. To solve this problem efficiently in a fully decentralized way, we propose a *S*tochastic *L*inearized *A*ugmented Lagrangian *M*ethod (SLAM) for dealing with both of the two levels of the optimization processes and the consensus constraints at each level. Leveraging the linearized augmented Lagrangian function as a surrogate, the design of SLAM is simple and easily implemented as it is a single-loop algorithm with only step sizes to be tuned for convergence. We make the standard assumptions on Lipschitz continuity and convexity for both the UL and LL optimization problems as shown in the existing literature. We establish the conditions of SLAM w.r.t. convergence to $\epsilon$-Karush-Kuhn-Tucker (KKT) points of problem (1) at a rate of $\mathcal{O}(1/(n\epsilon^2))$, matching the standard convergence rate achieved by decentralized SGD type of algorithm to FOSPs for only single-level nonconvex minimization problems. Remarkably, through numerical experiments on MARL problems, it is observed that SLAM can converge faster than the existing MARL methods and even achieve higher rewards in most cases.

To summarize, the main contributions of this work are highlighted as follows:

► Our proposed SLAM algorithm is generic, and thus generalizes the single agent-based bilevel algorithms to the multi-agent setting and is amnable to be specialized to solve multiple consensus-based DBO problems.

► SLAM is a single-timescale and single-loop algorithm that can find the $\epsilon$-KKT points at a rate of $\mathcal{O}(1/(n\epsilon^2))$, which shows a linear speedup w.r.t. the number of nodes. To the best of our knowledge, this is the first work that shows a decentralized stochastic algorithm can achieve this rate under the constraints where any level or both levels of the DBO problem requires the consensus process.

► Numerical results that illustrate the proposed SLAM outperforms the state-of-the-art MARL algorithms over heterogeneous networks in terms of both convergence speed and achievable rewards.

Due to space limitations, all technical proofs are deferred to the supplement.

## 2 Decentralized Bilevel Optimization Framework

**Problem formulation of DBO.** One of the main motivations for performing decentralized joint learning is dealing with large-scale dataset or scattered data samples. At each node, the loss function can be written as $f_i(\mathbf{x}_i, \mathbf{y}_{i,1}^*(\mathbf{x}_i), \ldots, \mathbf{y}_{i,m}^*(\mathbf{x}_i)) \triangleq \mathbb{E}_{\xi \in \mathcal{D}_i^U}[F_i(\mathbf{x}_i, \mathbf{y}_{i,1}^*(\mathbf{x}_i), \ldots, \mathbf{y}_{i,m}^*(\mathbf{x}_i); \xi)]$, where $\mathcal{D}_i^U$ denotes the local data distributions at the UL optimization problem, and $F_i(\mathbf{x}_i, \mathbf{y}_{i,1}^*(\mathbf{x}_i), \ldots, \mathbf{y}_{i,m}^*(\mathbf{x}_i); \xi)$ represents the estimation error of the UL learning model on

data $\xi \in \mathcal{D}_i^{\mathrm{U}}$. Similarly, the LL learning tasks also include randomly sampled data from a local distribution $\mathcal{D}_{i,k}^{\mathrm{L}}$ for task $k$, so the LL cost function at each node can be expressed as $g_{i,k}(\mathbf{x}_i, \mathbf{y}_{i,k}) \triangleq \mathbb{E}_{\zeta \in \mathcal{D}_{i,k}^{\mathrm{L}}}[G_{i,k}(\mathbf{x}_i, \mathbf{y}_i; \zeta)], \forall k$, where $G_{i,k}$ denotes the estimation error of the LL learning model over $\mathbf{y}_{k,i}$ on data $\zeta \in \mathcal{D}_{i,k}^{\mathrm{L}}$. It is well known that SGD is one of the most efficient algorithms for tackling large amounts of data samples. Before showing the algorithm design, we first reformulate problem (1) in a concise and compact way from a global view of the variables. Let $\mathbf{x} \triangleq [\mathbf{x}_1, \ldots, \mathbf{x}_n]^T$ and $\mathbf{y}_k \triangleq [\mathbf{y}_{1,k}, \ldots, \mathbf{y}_{n,k}]^T$. Then, problem (1) can be rewritten by concatenated variables as

$$\min_{\mathbf{x}} \quad f(\mathbf{x}, \mathbf{y}_k^*(\mathbf{x})) \triangleq \frac{1}{n} \sum_{i=1}^n f_i(\mathbf{x}_i, \mathbf{y}_{i,k}^*(\mathbf{x}_i)) \tag{2a}$$

$$\text{s.t.} \quad \mathbf{A}\mathbf{x} = 0, \tag{2b}$$

$$\mathbf{y}_k^*(\mathbf{x}) = \arg\min_{\mathbf{y}_k} g_k(\mathbf{x}, \mathbf{y}_k) \triangleq \frac{1}{n} \sum_{i=1}^n g_{i,k}(\mathbf{x}_i, \mathbf{y}_{i,k}) \quad \text{s.t.} \quad \mathbf{A}\mathbf{y}_k = 0, \forall k \in [m], \tag{2c}$$

where $g_k(\mathbf{x}, \mathbf{y}_k)$ denotes the $k$th LL loss function, $\mathbf{A} \in \mathbb{R}^{|\mathcal{E}| \times n}$ represents the incidence matrix[1] and $f_i(\mathbf{x}_i, \mathbf{y}_{i,k}^*(\mathbf{x}_i))$ abbreviates $f_i(\mathbf{x}_i, \mathbf{y}_{i,1}^*(\mathbf{x}_i), \ldots, \mathbf{y}_{i,m}^*(\mathbf{x}_i))$ for notational brevity.

**Algorithm Design.** Towards this end, it is straightforward to construct a variant of the classical augmented Lagrangian function for the UL optimization problem as

$$\mathcal{L}_{\rho\gamma}(\mathbf{x}, \boldsymbol{\lambda}) = f(\mathbf{x}, \mathbf{y}_k^*(\mathbf{x})) + \gamma\langle \boldsymbol{\lambda}, \mathbf{A}\mathbf{x} \rangle + \frac{\rho\gamma}{2}\|\mathbf{A}\mathbf{x}\|^2, \tag{3}$$

where $\boldsymbol{\lambda}$ denotes the dual variable (Lagrangian multiplier) for the consensus constraint, $\rho > 0$, and $\gamma$ is a scaling factor (which will be determined later).

Motivated by the primal-dual optimization framework [44], one step of gradient descent based on the linearized objective function with a following gradient ascent step is sufficient for the minimization of the general nonconvex loss function under the linear constraints, which means that there is no need to solve an inner optimization problem before updating the Lagrangian multiplier as is done in the classical augmented Lagrangian method.

When both the UL and LL objective functions are differentiable and the inverse of the Hessian matrix at the LL problem exists, i.e., $\nabla^2_{\mathbf{y}_k \mathbf{y}_k} g_k(\mathbf{x}, \mathbf{y}_k^*(\mathbf{x}))$ is invertible, then there exists a closed form for $\nabla f_i(\mathbf{x}_i, \mathbf{y}_{i,k}^*(\mathbf{x}_i))$. Following the existing works on bilevel algorithm designs, replacing $\mathbf{y}_{i,k}^*(\mathbf{x}_i)$ by $\mathbf{y}_{i,k}$ in the gradient of $f_i(\mathbf{x}_i, \mathbf{y}_{i,k}^*(\mathbf{x}_i))$ w.r.t. $\mathbf{x}_i$ can serve as an efficient surrogate for the stochastic gradient estimate. However, in the decentralized setting, only individual loss functions are observable at each agent, therefore, the local UL implicit gradient is computed through replacing $g_k(\mathbf{x}, \mathbf{y}_k)$ by $g_{i,k}(\mathbf{x}_i, \mathbf{y}_{i,k})$, denoted as $\overline{\nabla} f_i(\mathbf{x}_i, \mathbf{y}_{i,k})$. Let $\mathbf{h}_{g,k}^r$ and $\mathbf{h}_f^r$ respectively denote the distributed stochastic gradient estimate of the LL and UL objective functions at points $(\mathbf{x}^r, \mathbf{y}_k^r)$ and $(\mathbf{x}^r, \mathbf{y}_k^{r+1})$, $\forall k$, w.r.t. $\mathbf{y}_k$ and $\mathbf{x}$, where $r$ represents the index of iterations. Thus, our proposed SLAM can be expressed as

$$\mathbf{y}_k^{r+1} = \arg\min_{\mathbf{y}_k}\langle \mathbf{h}_{g,k}^r + \gamma\mathbf{A}^T(\boldsymbol{\omega}_k^r + \rho\mathbf{A}\mathbf{y}_k^r), \mathbf{y}_k - \mathbf{y}_k^r \rangle + \frac{\beta}{2}\|\mathbf{y}_k - \mathbf{y}_k^r\|^2, \quad \forall k, \tag{4a}$$

$$\boldsymbol{\omega}_k^{r+1} = \boldsymbol{\omega}_k^r + \frac{\rho}{\gamma}\mathbf{A}\mathbf{y}_k^{r+1}, \quad \forall k, \tag{4b}$$

$$\mathbf{x}^{r+1} = \arg\min_{\mathbf{x}}\langle \mathbf{h}_f^r + \gamma\mathbf{A}^T(\boldsymbol{\lambda}^r + \rho\mathbf{A}\mathbf{x}^r), \mathbf{x} - \mathbf{x}^r \rangle + \frac{\alpha}{2}\|\mathbf{x} - \mathbf{x}^r\|^2, \tag{4c}$$

$$\boldsymbol{\lambda}^{r+1} = \boldsymbol{\lambda}^r + \frac{\rho}{\gamma}\mathbf{A}\mathbf{x}^{r+1}, \tag{4d}$$

where $\boldsymbol{\omega}_k$ is the dual variable for ensuring the LL consensus process for each learning task, $\alpha$ and $\beta$ are the parameters of the quadratic penalization terms, and $\rho/\gamma$ here is the step-size for the updates of the dual variables.

**Implementation of SLAM.** Noting that the objective functions in each subproblem, i.e., (4a) and (4c), are quadratic, we can easily have the updates of both UL and LL optimization variables as

---

[1]Here, we assume the problem dimension is 1, without loss of generality, to simplify the notation.

$$\mathbf{y}_k^{r+1} = \mathbf{y}_k^r - \frac{1}{\beta}\left(\mathbf{h}_{g,k}^r + \gamma\mathbf{A}^T\boldsymbol{\omega}_k^r + \rho\gamma\mathbf{A}^T\mathbf{A}\mathbf{y}_k^r\right), \forall k, \tag{5a}$$

$$\mathbf{x}^{r+1} = \mathbf{x}^r - \frac{1}{\alpha}\left(\mathbf{h}_f^r + \gamma\mathbf{A}^T\boldsymbol{\lambda}^r + \rho\gamma\mathbf{A}^T\mathbf{A}\mathbf{x}^r\right), \tag{5b}$$

where $1/\alpha$ and $1/\beta$ serve as the step-sizes of updating both UL and LL learning models. Subtracting the equality with the same one from the previous iteration for both (5a) and (5b) ends up with efficient model updates of both the UL and LL learning problems as follows:

$$\mathbf{y}_k^{r+1} = 2\mathbf{W}_g\mathbf{y}_k^r - \mathbf{W}_g'\mathbf{y}_k^{r-1} - \frac{1}{\beta}\left(\mathbf{h}_{g,k}^r - \mathbf{h}_{g,k}^{r-1}\right), \quad \forall k, \tag{6a}$$

$$\mathbf{x}^{r+1} = 2\mathbf{W}_f\mathbf{x}^r - \mathbf{W}_f'\mathbf{x}^{r-1} - \frac{1}{\alpha}\left(\mathbf{h}_f^r - \mathbf{h}_f^{r-1}\right), \tag{6b}$$

where the mixing matrices, with $\tau_g = \beta/\gamma$ and $\tau_f = \alpha/\gamma$, are defined as

$$\mathbf{W}_g \triangleq \mathbf{I} - \frac{(1+\gamma^{-1})\rho}{2\tau_g}\mathbf{A}^T\mathbf{A}, \quad \mathbf{W}_g' \triangleq \mathbf{I} - \frac{\rho}{\tau_g}\mathbf{A}^T\mathbf{A}, \tag{7a}$$

$$\mathbf{W}_f \triangleq \mathbf{I} - \frac{(1+\gamma^{-1})\rho}{2\tau_f}\mathbf{A}^T\mathbf{A}, \quad \mathbf{W}_f' \triangleq \mathbf{I} - \frac{\rho}{\tau_f}\mathbf{A}^T\mathbf{A}. \tag{7b}$$

According to (6a) and (6b), it can be readily observed that SLAM is amenable to a fully decentralized implementation. The detailed algorithm description is provided in Algorithm 1 from a local view of the model update, where $[\mathbf{W}]_{ij}$ denotes the $ij$th entry of matrix $\mathbf{W}$, $[\mathbf{h}_g^r]_{i,k}$ is the gradient estimate of $\nabla g_{i,k}(\mathbf{x}_i^r, \mathbf{y}_{i,k}^r)$ (i.e., $\mathbf{h}_{g,k}^r = [[\mathbf{h}_g^r]_{1,k}, \ldots, [\mathbf{h}_g^r]_{n,k}]^T$), and similarly $[\mathbf{h}_f^r]_i$ is the local gradient estimate of $\overline{\nabla} f_i(\mathbf{x}_i^r, \mathbf{y}_{i,k}^{r+1})$ (i.e., $\mathbf{h}_f^r = [[\mathbf{h}_f^r]_1, \ldots, [\mathbf{h}_f^r]_n]^T$).

---

**Algorithm 1** Decentralized implementation of SLAM

---

**Initialization:** $\alpha, \beta, \gamma, \mathbf{x}_i^1, \mathbf{y}_{i,k}^1, \forall i, k$, and set $\boldsymbol{\lambda}^1 = \boldsymbol{\omega}_k^1 = 0, \forall k$;
1: **for** $r = 1, 2, \cdots, T$ **do**
2:     **for** $i = 1, 2, \cdots, n$ in parallel over the network **do**
3:         Estimate gradient $\nabla g_{i,k}(\mathbf{x}_i^r, \mathbf{y}_{i,k}^r)$ for each task and $\nabla f_i(\mathbf{x}_i^r, \mathbf{y}_{i,k}^{r+1})$ locally
4:         $\mathbf{y}_{i,k}^{r+1} = 2\sum_{j\in\mathcal{N}_i}[\mathbf{W}_g]_{ij}\mathbf{y}_{j,k}^r - [\mathbf{W}_g']_{ij}\mathbf{y}_{j,k}^{r-1} - \beta^{-1}\left([\mathbf{h}_g^r]_{i,k} - [\mathbf{h}_g^{r-1}]_{i,k}\right)$   ▷ LL models
5:         $\mathbf{x}_i^{r+1} = 2\sum_{j\in\mathcal{N}_i}[\mathbf{W}_f]_{ij}\mathbf{x}_j^r - [\mathbf{W}_f']_{ij}\mathbf{x}_j^{r-1} - \alpha^{-1}\left([\mathbf{h}_f^r]_i - [\mathbf{h}_f^{r-1}]_i\right)$       ▷ UL model
6:     **end for**
7: **end for**

---

Besides, if there is a consensus requirement at only one level of the optimization problem, then the problem at the other level becomes one with multiple objective functions. Our proposed SLAM can also be applied for solving any of these problems by a minor revision of the generic SLAM formulation. To be more specific, we provide the following discussion.

**A Special Case of DBO** (1) **(with only consensus in the LL problems).** If there is only a need for consensus of LL model parameters, then problem (2) reduces to the following DBO problem. For example, in solving multi-agent actor-critic RL problems, the UL optimization problem consists of improving the policy for each agent while the LL problem requires all the agents to jointly evaluate the value function over the whole network. The DBO problem is then expressed as

$$\min_{\mathbf{x}_i} \quad f_i(\mathbf{x}_i, \mathbf{y}_{i,k}^*(\mathbf{x}_i)), \quad \forall i \in [n] \tag{8a}$$

$$\text{s.t.} \quad \mathbf{y}_k^*(\mathbf{x}) = \arg\min_{\mathbf{y}_k} g_k(\mathbf{x}, \mathbf{y}_k) \triangleq \frac{1}{n}\sum_{i=1}^n g_{i,k}(\mathbf{x}_i, \mathbf{y}_{i,k}) \quad \text{s.t.} \quad \mathbf{A}\mathbf{y}_k = 0, \forall k \in [m]. \tag{8b}$$

The major difference between problem (2) and (8) is that the UL optimization problem includes multiple objectives over the model parameters $\mathbf{x}_i, \forall i \in [n]$. In this case, the updating rule of variable $\mathbf{x}$ in (6b) reduces to $\mathbf{x}^{r+1} = \mathbf{x}^r - \mathbf{h}_f^r/\alpha$ by forgoing the dual update w.r.t. $\boldsymbol{\lambda}$. The detailed implementation is summarized in Algorithm 2, where we name this special case of SLAM by SLAM-L as the LL consensus process is the main feature in this setting.

---

**Algorithm 2** Decentralized implementation of SLAM-L

---

**Initialization:** $\alpha, \beta, \gamma, \mathbf{x}_i^1, \mathbf{y}_{i,k}^1, \forall i, k$, and set $\boldsymbol{\omega}_k^1 = 0, \forall k$;

1: **for** $r = 1, 2, \cdots, T$ **do**
2:      **for** $i = 1, 2, \cdots, n$ in parallel over the network **do**
3:          Estimate gradient $\nabla g_{i,k}(\mathbf{x}_i^r, \mathbf{y}_{i,k}^r)$ for each task and $\nabla f_i(\mathbf{x}_i^r, \mathbf{y}_{i,k}^{r+1})$ locally
4:          $\mathbf{y}_{i,k}^{r+1} = 2\sum_{j \in \mathcal{N}_i}[\mathbf{W}_g]_{ij}\mathbf{y}_{j,k}^r - [\mathbf{W}_g']_{ij}\mathbf{y}_{j,k}^{r-1} - \beta^{-1}\left([\mathbf{h}_g^r]_{i,k} - [\mathbf{h}_g^{r-1}]_{i,k}\right)$    ▷ LL models
5:          $\mathbf{x}_i^{r+1} = \mathbf{x}_i^r - \alpha^{-1}[\mathbf{h}_f^r]_i, \forall i$                              ▷ UL models
6:      **end for**
7: **end for**

---

**A Special Case of DBO** (1) **(with only consensus in the UL problem).** The other special is analogous to the first one with the difference being the absence of the LL consensus process in comparison to (2), which is written as follows:

$$\min_{\mathbf{x}} \quad f(\mathbf{x}, \mathbf{y}_k^*(\mathbf{x})) \triangleq \frac{1}{n}\sum_{i=1}^n f_i(\mathbf{x}_i, \mathbf{y}_{i,k}^*(\mathbf{x}_i)) \tag{9a}$$

$$\text{s.t.} \quad \mathbf{A}\mathbf{x} = 0, \quad \mathbf{y}_{i,k}^*(\mathbf{x}_i) = \arg\min_{\mathbf{y}_{i,k}} g_{i,k}(\mathbf{x}_i, \mathbf{y}_{i,k}), \forall i \in [n], \forall k \in [m], \tag{9b}$$

where there are multiple objectives in the LL optimization problems. Problem (9) also covers a wide range of applications in machine learning, e.g., multi-task and/or personalized learning, and so on. In this case, the update of variable $\mathbf{y}_k$ shown in (5a) is changed to $\mathbf{y}_k^{r+1} = \mathbf{y}_k^r - \mathbf{h}_g^r/\beta$ as there is no consensus constraint involved. Analogous to the previous case, the implementation of this algorithm is presented in Algorithm 3 and termed as SLAM-U.

---

**Algorithm 3** Decentralized implementation of SLAM-U

---

**Initialization:** $\alpha, \beta, \gamma, \mathbf{x}_i^1, \mathbf{y}_{i,k}^1, \forall i, k$, and set $\boldsymbol{\lambda}^1 = 0, \forall k$;

1: **for** $r = 1, 2, \cdots, T$ **do**
2:      **for** $i = 1, 2, \cdots, n$ in parallel over the network **do**
3:          Estimate gradient $\nabla g_{i,k}(\mathbf{x}_i^r, \mathbf{y}_{i,k}^r)$ for each task and $\nabla f_i(\mathbf{x}_i^r, \mathbf{y}_{i,k}^{r+1})$ locally
4:          $\mathbf{y}_{i,k}^{r+1} = \mathbf{y}_{i,k}^r - \beta^{-1}[\mathbf{h}_g^r]_{i,k}$                                  ▷ LL models
5:          $\mathbf{x}_i^{r+1} = 2\sum_{j \in \mathcal{N}_i}[\mathbf{W}_f]_{ij}\mathbf{x}_j^r - [\mathbf{W}_f']_{ij}\mathbf{x}_j^{r-1} - \alpha^{-1}\left([\mathbf{h}_f^r]_i - [\mathbf{h}_f^{r-1}]_i\right)$    ▷ UL model
6:      **end for**
7: **end for**

---

## 3 Theoretical Convergence Results

Before showing the theoretical results about the convergence guarantees of SLAM, we first need five main classes of assumptions used in showing the descent of some quantifiable function so that SLAM can reach the $\epsilon$-KKT points of the DBO problems. More detailed definitions and properties regarding these assumptions are deferred to the supplement.

### 3.1 Assumptions

Our theoretical results are based on the following assumptions on the properties of the loss functions in both the UL and LL optimization problems, which are mainly related to the continuity of the objective function and stochasticity of the gradient estimates.

     A1. (Lipschitz continuity of both UL and LL objective functions) Assume that functions $f_i(\cdot), \nabla f_i(\cdot,), \nabla g_{i,k}(\cdot), \nabla^2 g_{i,k}(\cdot), \forall i$, are (block-wise) Lipschitz continuous with constants $L_{f,0}, L_{f,1}, L_{g,1}, L_{g,2}$ for both $\mathbf{x}$ and $\mathbf{y}_k, \forall k$, and $\nabla^2_{\mathbf{x}_i \mathbf{y}_{i,k}} g_{i,k}(\cdot), \forall i$ are bounded by $C_{xy}$.

     A2. (Connectivity of graph $\mathcal{G}$) The communication graph $\mathcal{G}$ is assumed to be well-connected, i.e., $\mathbb{1}^T\mathbf{L} = 0$ where $\mathbf{L} = \mathbf{A}^T\mathbf{A}$, and the second-smallest eigenvalue of $\mathbf{L}$ is assumed to be strictly positive, i.e., $\widetilde{\sigma}_{\min}(\mathbf{A}^T\mathbf{A}) > 0$.

A3. (Quality of the stochastic gradient estimate) The stochastic estimates of $\nabla f_i(\mathbf{x}_i, \mathbf{y}_{i,k})$, $\nabla_{\mathbf{y}_i} g_{i,k}(\mathbf{x}_i, \mathbf{y}_{i,k}), \forall i, k$, are unbiased and their variances are bounded by $\sigma_f^2, \sigma_g^2$.

A4. Assume that the UL objective functions $f_i(\mathbf{x}_i, \mathbf{y}_{i,k}^*(\mathbf{x}_i)), \forall i, k$ are lower bounded.

A5. (Strong convexity of $g_{i,k}(\cdot)$ w.r.t. $\mathbf{y}_{i,k}$) Function $g_{i,k}(\cdot)$ is $\mu_g$-strongly convex w.r.t. $\mathbf{y}_{i,k}, \forall i, k$.

Note that these assumptions are commonly used in the convergence analysis for bilevel and decentralized optimization algorithms. Given these assumptions, we are now in a position to provide the following theoretical convergence guarantees.

## 3.2 Convergence Rates of SLAM

**Theorem 1.** *(Convergence rate of SLAM to $\epsilon$-KKT points) Suppose that A1-A5 hold and assume* $\|\nabla_{\mathbf{y}_i\mathbf{y}_i}^2 g_{i,k}(\cdot, \mathbf{y}_i) - n^{-1}\sum_{i=1}^n \nabla_{\mathbf{y}_i\mathbf{y}_i}^2 g_{i,k}(\cdot, \mathbf{y}_i')\| \leq L_g\|\mathbf{y}_i - \mathbf{y}_i'\|, \forall i, k$ *if* $\nabla^2 g_{i,k}(\cdot), \forall i, k$ *are required in computing the UL implicit gradient. When step-sizes are chosen as* $1/\alpha \sim 1/\beta \sim \mathcal{O}(\sqrt{n/T})$, $\tau_f, \tau_g \geq \mathcal{O}(\rho\sigma_{\max}(\mathbf{A}^T\mathbf{A}))$, *the mini-batch size of* $\mathbf{h}_f^r$ *is* $\mathcal{O}(\log(nT))$, *then the iterates* $\{\mathbf{x}^r, \boldsymbol{\lambda}^r, \mathbf{y}_k^r, \boldsymbol{\omega}_k^r, \forall k, r\}$ *generated by SLAM satisfy*

$$\text{UL: } \frac{1}{T}\sum_{r=1}^T \mathbb{E}[\|\nabla f(\mathbb{1}\bar{\mathbf{x}}^r, \mathbf{y}_1^*(\mathbb{1}\bar{\mathbf{x}}^r), \ldots, \mathbf{y}_m^*(\mathbb{1}\bar{\mathbf{x}}^r))\|^2] \sim \frac{1}{T}\sum_{r=1}^T \mathbb{E}[\|\mathbf{A}\mathbf{x}^r\|^2] \sim \mathcal{O}(1/\sqrt{nT}), \quad (10a)$$

$$\text{LL: } \frac{1}{T}\sum_{r=1}^T \mathbb{E}[\|\bar{\mathbf{y}}_k^r - \bar{\mathbf{y}}_k^*(\mathbf{x}^r)\|^2] \sim \frac{1}{T}\sum_{r=1}^T \mathbb{E}[\|\mathbf{A}\mathbf{y}_k^r\|^2] \sim \mathcal{O}(1/\sqrt{nT}), \quad \forall k, \quad (10b)$$

*where* $\bar{\mathbf{x}} = n^{-1}\mathbb{1}^T\mathbf{x}$, *and* $T$ *denotes the total number of iterations.*

*Remark 1.* It is noted in Theorem 1 that the convergence rate achieved by SLAM to find the $\epsilon$-approximate KKT points of (1) (including both the first-order stationarity of the solutions and the violation of constraints) is on the order of $1/(n\epsilon^2)$. Therefore, it follows that a linear speedup w.r.t. the number of learners can be achieved by SLAM for DBO, matching the classical results of distributed SGD for only single-level general nonconvex problems.

*Remark 2.* In comparison with existing bilevel algorithms, SLAM is a single timescale algorithm since the learning rates can be chosen as $1/\alpha \sim 1/\beta$, which is consistent with ALSET [43].

*Remark 3.* The major novelty of obtaining these theoretical results relies on the developed variant of the augmented Lagrangian function and subsequently derived recursion of the successive dual variables, which quantify well the consensus errors resulting from both UL and LL optimization processes in terms of primal variables. Note that this is distinct from the existing theoretical analysis of stochastic algorithms, such as distributed SGD [38, 39], stochastic gradient tracking [40, 41], stochastic primal-dual algorithms [42, 45], etc.

**Corollary 1.** *(Convergence rate of SLAM-L to $\epsilon$-KKT points) Suppose that A1-A5 hold and assume* $\|\nabla_{\mathbf{y}_i\mathbf{y}_i}^2 g_{i,k}(\mathbf{x}_i, \mathbf{y}_i) - \nabla_{\mathbf{y}\mathbf{y}}^2 g_k(\mathbf{x}, \mathbf{y}_i')\| \leq L_g\|\mathbf{y}_i - \mathbf{y}_i'\|, \forall i, k$ *if* $\nabla^2 g_{i,k}(\cdot), \forall i, k$ *are required in computing the UL implicit gradient. When step-sizes are chosen as* $1/\alpha \sim \mathcal{O}(1/\sqrt{T}), 1/\beta \sim \mathcal{O}(\sqrt{n/T})$, $\tau_f, \tau_g \geq \mathcal{O}(\rho\sigma_{\max}(\mathbf{A}^T\mathbf{A}))$, $\rho \geq n$, *the mini-batch size of* $\mathbf{h}_f^r$ *is* $\mathcal{O}(\log(nT))$, *the iterates* $\{\mathbf{x}^r, \mathbf{y}_k^r, \boldsymbol{\omega}_k^r, \forall k, r\}$ *generated by SLAM-L satisfy*

$$\text{UL: } \quad \frac{1}{T}\sum_{r=1}^T \mathbb{E}[\|\nabla f_i(\mathbf{x}_i^r, \mathbf{y}_{i,1}^*(\mathbf{x}_i^r), \ldots, \mathbf{y}_{i,m}^*(\mathbf{x}_i^r)\|^2], \forall i \sim \mathcal{O}(n/\sqrt{T}) \quad \text{and} \quad \text{LL: } (10b).$$

*Remark 4.* Different from Theorem 1, the stationarity of the UL model parameters requires the shrinkage of the gradient size over each individual UL problem as shown in Corollary 1, so there is no speedup on the convergence rate guarantee at UL.

**Corollary 2.** *(Convergence rate of SLAM-U to $\epsilon$-KKT points) Suppose that A1-A5 hold. Given the conditions on* $1/\alpha, 1/\beta, \tau_f, \tau_g$ *and the mini-batch size of* $\mathbf{h}_f^r$ *shown in Theorem 1, the iterates* $\{\mathbf{x}^r, \boldsymbol{\lambda}^r, \mathbf{y}_k^r, \forall k, r\}$ *generated by SLAM-U satisfy*

$$\text{UL: } \quad (10a) \quad \text{and} \quad \text{LL: } \frac{1}{T}\sum_{r=1}^T \mathbb{E}[\|\mathbf{y}_k^r - \mathbf{y}_k^*(\mathbf{x}^r)\|^2] \sim \mathcal{O}(1/\sqrt{nT}), \forall k.$$

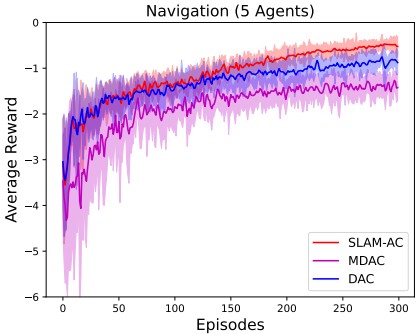
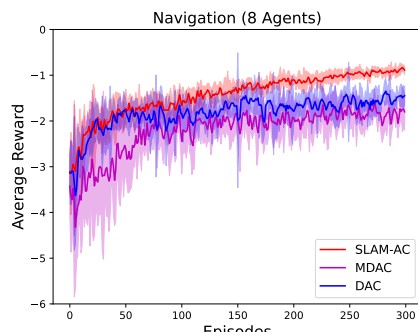

Figure 1: The averaged reward versus the learning process on the cooperative navigation task.

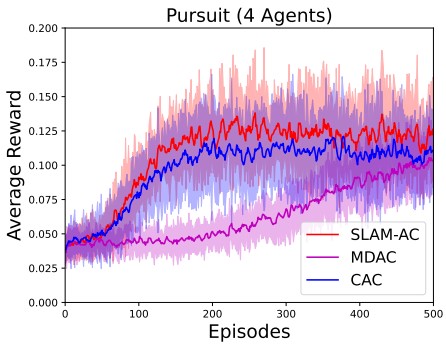
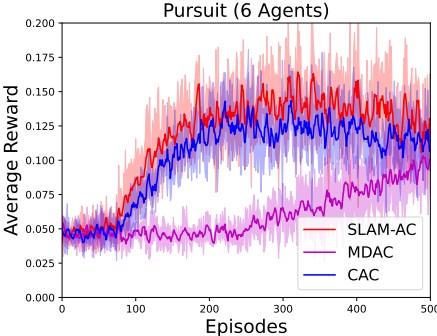

Figure 2: The averaged reward versus the learning process on the pursuit-evasion game. (Consensus with one layer of the actor neural nets and all layers of the critic neural nets.)

## 4 Numerical Results

In this section, we evaluate our proposed algorithm using two MARL environments: 1) the cooperative navigation task [16], which is built on the OpenAI Gym platform [46]; and 2) the pursuit-evasion game [47], which is built on the PettingZoo platform [48]. Detailed experimental settings and additional numerical results are provided in the supplement.

**Cooperative Navigation Task.** In this game, we consider that the $n$ agents are aiming to jointly reach $n$ different landmarks as soon as possible, where the Erdos Renyi Graph is used. We assume that each agent can observe the global state and has 5 possible actions: stay, left, right, up, and down. This task consists of a shared common goal of avoiding collision among the agents while they navigate to the targeting landmarks. In the simulations, each agent locally maintains two fully connected neural networks as the actor network (at UL w.r.t. $\mathbf{x}_i$) and the critic network (at LL w.r.t. $\mathbf{y}_i$), respectively. Moreover, each agent shares its critic network with its neighbors to cooperatively estimate the global value function and independently train its actor network to complete its local task.

We compare the performance of our proposed SLAM with application to the DAC setting, named SLAM-AC, with two benchmark algorithms: DAC [15] and mini-batch DAC (MDAC) [22]. Theoretically, MDAC needs an $\mathcal{O}(\epsilon^{-1} \ln \epsilon^{-1})$ batch size in its inner loop to update critic parameters before each update in policy parameters, which is not practical. Here, we set a small batch $B = 10$ in the inner loop for MDAC to achieve fast convergence. The simulation results on this coordination game are presented in Figure1, where the performance is averaged over 5 independent Monte Carlo (MC) trials for each algorithm.

**Pursuit-Evasion Game.** In the pursuit-evasion game, there are two groups of nodes: pursuers (agents) and evaders. The agents are connected through a ring graph. Pursuers could observe the global state of the video game. An evader is considered caught if two pursuers simultaneously arrive at the evader's location. As each pursuer should learn to cooperate with other pursuers to catch the

evaders, the pursuers share certain similarities with each other since they need to follow similar strategies to achieve their local tasks: simultaneously catching an evader with other pursuers.

We follow the experimental set up in [23], where all agents partially share their actor networks with neighbors for collaborations in their policy spaces and fully share their critic network to cooperatively learn the global value function. In Figure 2, we compare SLAM-AC with two benchmarks, CAC [23] and MDAC [22], with 5 MC trials again. To ensure a fair comparison, all algorithms use the same parameter sharing scheme mentioned above. Note that CAC [23] is a variant of DAC [15] and the only difference is that CAC can partially share its policy parameters while the policy parameters are not shared in DAC. In the experiment, each agent maintains two convolutional neural networks, one for the actor and one for the critic (Please refer to the supplement for detailed structures).

## 5    Concluding Remark

In this paper, we studied a generic form of the DBO problem, which is shown to have three major variants that formulate multiple hierarchical machine learning problems. Targeting these DBO problems, we proposed SLAM – a simple and elegant algorithm to solve DBO in a fully decentralized way. Under mild conditions, we establish theoretical results showing that our proposed SLAM is able to find the $\epsilon$-KKT points with a convergence rate of $\mathcal{O}(1/(n\epsilon^2))$, which matches the standard convergence rate achieved by the classical distributed SGD algorithms for solving only single-level general nonconvex optimization problems. We tested the performance of SLAM numerically on a MARL scenario and found that SLAM outperformed the traditional AC algorithms w.r.t. convergence speed and (in most cases) achievable rewards.

**Societal impact.** To the best of our knowledge, we do not see any ethical or negative immediate societal consequence of this work.

## Acknowledgments

M. Hong and S. Zeng are partially supported by NSF grants CIF-1910385 and CMMI-1727757.

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
