# Supplementary Material

## A Preliminaries

In this section, we provide some technical preliminaries for the proofs of the lemmas and theorems claimed in the main body of this paper, including parameter definitions and supporting results. First, let us define the filtrations

$$\mathcal{F}^r = \{\mathbf{x}^r, \mathbf{y}^r, \boldsymbol{\lambda}^r, \boldsymbol{\omega}_k^r, \xi^{r-1}, \zeta^{r-1}, \ldots, x^0, \boldsymbol{\lambda}^0\}, \tag{11}$$

$$\mathcal{F}'^r = \{\mathbf{x}^r, \mathbf{y}^{r+1}, \boldsymbol{\lambda}^r, \boldsymbol{\omega}_k^r, \xi^{r-1}, \zeta^r, \ldots, x^0, \boldsymbol{\lambda}^0\}, \tag{12}$$

which will often be used in the proofs when taking conditional expectation.

Inequalities used in the proof include

1. Quadrilateral identity:

$$\left\langle \mathbf{x}^{r+1} - \mathbf{x}^r, \mathbf{x}^r - \mathbf{x}^{r-1} \right\rangle = \frac{1}{2} \left( \|\mathbf{x}^{r+1} - \mathbf{x}^r\|^2 + \|\mathbf{x}^r - \mathbf{x}^{r-1}\|^2 - \|\mathbf{w}^{r+1}\|^2 \right) \tag{13}$$

    where

$$\mathbf{w}^{r+1} \triangleq \mathbf{x}^{r+1} - \mathbf{x}^r - (\mathbf{x}^r - \mathbf{x}^{r-1}). \tag{14}$$

2. Young's inequality with parameter $\theta > 0$:

$$\langle \mathbf{x}, \mathbf{y} \rangle \leq \frac{1}{2\theta} \|\mathbf{x}\|^2 + \frac{\theta}{2} \|\mathbf{y}\|^2, \quad \forall \mathbf{x}, \mathbf{y}. \tag{15}$$

3. Given vectors $\mathbf{x}_1, \ldots, \mathbf{x}_n$, the convexity of norm $\|\|^2$ and a trivial application of Jensen's inequality yields the following inequality

$$\left\| \frac{1}{n} \sum_{i=1}^n \mathbf{x}_i \right\|^2 \leq \frac{1}{n} \sum_{i=1}^n \|\mathbf{x}_i\|^2. \tag{16}$$

4. Given that $\mathbf{x}$ is a random vector, then

$$\mathbb{E}\left[\|\mathbf{x}\|^2\right] = \|\mathbb{E}[\mathbf{x}]\|^2 + \mathbb{E}\left[\|\mathbf{x} - \mathbb{E}[\mathbf{x}]\|^2\right]. \tag{17}$$

**Lemma 1.**

*Under A1, A2 and A5, the gradient of $f_i(\mathbf{x}_i, \mathbf{y}_{i,k}^*(\mathbf{x}))$ is given by*

$$\nabla f_i(\mathbf{x}_i, \mathbf{y}_{i,k}^*(\mathbf{x})) = \nabla_{\mathbf{x}_i} f_i(\mathbf{x}_i, \mathbf{y}_{i,k}^*(\mathbf{x}))$$
$$- \sum_{k=1}^m \nabla_{\mathbf{x}_i \mathbf{y}_k}^2 g_{i,k}(\mathbf{x}_i, \mathbf{y}_k^*(\mathbf{x})) \left[\nabla_{\mathbf{y}_k \mathbf{y}_k}^2 g_k(\mathbf{x}, \mathbf{y}_k^*(\mathbf{x}))\right]^{-1} \nabla_{\mathbf{y}_{i,k}} f_i(\mathbf{x}_i, \mathbf{y}_{i,k}^*(\mathbf{x})), \tag{18}$$

*where $\nabla_{\mathbf{x}_i} f_i(\mathbf{x}_i, \mathbf{y}_{i,k}^*(\mathbf{x}))$ denotes the gradient of the objective function w.r.t. the first argument.*

*Proof.* In order to remove the ambiguity of the notation, we first define $F_i(\mathbf{x}) \triangleq f_i(\mathbf{x}_i, \mathbf{y}_{i,k}^*(\mathbf{x}))$. Following the classical proving steps [43, Proposition 1], we obtain the closed form of the implicit gradient by the chain rule:

$$\nabla_{\mathbf{x}_i} F_i(\mathbf{x}) = \nabla_{\mathbf{x}_i} f_i(\mathbf{x}_i, \mathbf{y}_{i,k}^*(\mathbf{x})) + \sum_{k=1}^m \nabla_{\mathbf{x}_i} \mathbf{y}_{i,k}^*(\mathbf{x})^T \nabla_{\mathbf{y}_{i,k}} f_i(\mathbf{x}_i, \mathbf{y}_{i,k}^*(\mathbf{x})). \tag{19}$$

Based on the definition of $\mathbf{y}_k^*(\mathbf{x})$, it follows that

$$\nabla_{\mathbf{y}_k} g_k(\mathbf{x}, \mathbf{y}_k^*(\mathbf{x})) = 0, \quad \mathbf{A}\mathbf{y}_k^*(\mathbf{x}) = 0, \tag{20}$$

and thus we have

$$\nabla_{\mathbf{x}_i} \left( \frac{1}{n} \sum_{i=1}^n \nabla_{\mathbf{y}_k} g_{i,k}(\mathbf{x}, \mathbf{y}_{i,k}^*(\mathbf{x})) \right) = 0, \quad \mathbf{A}\mathbf{y}_k^*(\mathbf{x}) = 0. \tag{21}$$

Therefore, we obtain

$$\nabla_{\mathbf{x}_i \mathbf{y}_k}^2 g_{i,k}(\mathbf{x}_i, \mathbf{y}_k^*(\mathbf{x})) + \nabla_{\mathbf{x}_i} \mathbf{y}_{i,k}^*(\mathbf{x})^T \nabla_{\mathbf{y}_k \mathbf{y}_k}^2 g_k(\mathbf{x}, \mathbf{y}_k^*(\mathbf{x})) = 0. \tag{22}$$

According to A5, the inverse of the Hessian matrix exists. Substituting (22) back into (19) directly yields the desired result. $\square$

Note that $\overline{\nabla} f_i(\mathbf{x}_i, \mathbf{y}_{i,k})$ denotes the surrogate gradient at UL through replacing $\mathbf{y}^*_{i,k}(\mathbf{x})$ in (18) $\mathbf{y}_{i,k}$ with the local loss function, i.e.,

$$\overline{\nabla} f_i(\mathbf{x}_i, \mathbf{y}_{i,k}) = \nabla_{\mathbf{x}_i} f_i(\mathbf{x}_i, \mathbf{y}_{i,k})$$
$$- \sum_{k=1}^m \nabla^2_{\mathbf{x}_i \mathbf{y}_{i,k}} g_{i,k}(\mathbf{x}_i, \mathbf{y}_{i,k}) \left[ \nabla^2_{\mathbf{y}_{i,k} \mathbf{y}_{i,k}} g_{i,k}(\mathbf{x}_i, \mathbf{y}_{i,k}) \right]^{-1} \nabla_{\mathbf{y}_{i,k}} f_i(\mathbf{x}_i, \mathbf{y}_{i,k}). \quad (23)$$

It has been shown in [31, Lemma 3.2] and [33, Lemma 1] that, when the gradient estimator is constructed in a certain way, we can have

$$\|\overline{\nabla} f_i(\mathbf{x}_i^r, \mathbf{y}_{i,k}^{r+1}) - \mathbb{E} \mathbf{h}_{f,i}\| \triangleq b_{r,i} \leq \mathcal{O}(b^{m_b}), \quad \forall i, \quad (24)$$

due to the independence among the LL tasks, where $0 < b < 1$ and $m_b \geq r$ denotes the mini-batch size. Therefore, we only need to choose $m_b = \mathcal{O}(\log(nT))$ to obtain $b_r^2 \leq \mathcal{O}(1/\sqrt{nT})$, where $b_r \triangleq \sum_{i=1}^n b_{r,i}$.

**Lemma 2.**

Under A1 and A5, $\mathbf{y}^*_k(\mathbf{x})$ is Lipschitz continuous, namely
$$\|\bar{\mathbf{y}}^*_k(\mathbf{x}) - \bar{\mathbf{y}}^*_k(\mathbf{x}')\| \leq L_y \|\mathbf{x} - \mathbf{x}'\|, \quad \forall k, \quad (25)$$
where $L_y \triangleq \frac{C_{xy}}{\mu_g}$, and $\nabla \mathbf{y}^*_k(\mathbf{x})$ is also Lipschitz continuous, namely
$$\|\nabla \bar{\mathbf{y}}^*_k(\mathbf{x}) - \nabla \bar{\mathbf{y}}^*_k(\mathbf{x}')\| \leq L_{xy} \|\mathbf{x} - \mathbf{x}'\|, \quad \forall k, \quad (26)$$
where $L_{xy} \triangleq \frac{\sqrt{2} L_{g,2}}{\mu_g}(1 + L_y + C_{xy}(1 + L_y)\mu_g^{-1})$.

*Proof.* **First Part.** According to (20) and (22), we have

$$\|\nabla_{\mathbf{x}_i} \mathbf{y}^*_{i,k}(\mathbf{x})\| \leq \|\nabla^2_{\mathbf{y}_k \mathbf{y}_k} g_k(\mathbf{x}, \mathbf{y}^*_k(\mathbf{x}))^{-1} \nabla_{\mathbf{x}_i \mathbf{y}_k} g_k(\mathbf{x}, \mathbf{y}^*_k(\mathbf{x}))^T\| \leq \frac{C_{xy}}{\mu_g}, \quad (27)$$

where we use A1 and A5. Therefore, we obtain $\|\nabla_{\mathbf{x}} \bar{\mathbf{y}}^*_k(\mathbf{x})\| \leq L_{g,2}/\mu_g$ and

$$\|\bar{\mathbf{y}}^*_k(\mathbf{x}) - \bar{\mathbf{y}}^*_k(\mathbf{x}')\| \leq \frac{C_{xy}}{\mu_g} \|\mathbf{x} - \mathbf{x}'\|. \quad (28)$$

**Second Part.** Next, we can have

$$\|\nabla_{\mathbf{x}} \bar{\mathbf{y}}^*_k(\mathbf{x}) - \nabla_{\mathbf{x}} \bar{\mathbf{y}}^*_k(\mathbf{x}')\|$$
$$= \|\nabla^2_{\mathbf{x} \mathbf{y}_k} g_k(\mathbf{x}, \mathbf{y}^*_k(\mathbf{x}))[\nabla^2_{\mathbf{y}_k \mathbf{y}_k} g_k(\mathbf{x}, \mathbf{y}^*_k(\mathbf{x}))]^{-1} - \nabla^2_{\mathbf{x} \mathbf{y}_k} g_k(\mathbf{x}, \mathbf{y}^*_k(\mathbf{x}'))[\nabla^2_{\mathbf{y}_k \mathbf{y}_k} g_k(\mathbf{x}', \mathbf{y}^*_k(\mathbf{x}'))]^{-1}\| \quad (29)$$
$$\leq \frac{1}{\mu_g} \|\nabla^2_{\mathbf{x} \mathbf{y}_k} g_k(\mathbf{x}, \mathbf{y}^*_k(\mathbf{x})) - \nabla^2_{\mathbf{x} \mathbf{y}_k} g_k(\mathbf{x}', \mathbf{y}^*_k(\mathbf{x}'))\| + \frac{C_{xy}}{\mu_g^2} \|\nabla^2_{\mathbf{y}_k \mathbf{y}_k} g_k(\mathbf{x}, \mathbf{y}^*_k(\mathbf{x})) - \nabla^2_{\mathbf{y}_k \mathbf{y}_k} g_k(\mathbf{x}', \mathbf{y}^*_k(\mathbf{x}'))\|$$
$$\overset{(a)}{\leq} \frac{\sqrt{2} L_{g,2}}{\mu_g} \left( \|\mathbf{x} - \mathbf{x}'\| + \|\mathbf{y}^*_k(\mathbf{x}) - \mathbf{y}^*_k(\mathbf{x}')\| \right) + \frac{\sqrt{2} C_{xy} L_{g,2}}{\mu_g^2} \left( \|\mathbf{x} - \mathbf{x}'\| + \|\mathbf{y}^*_k(\mathbf{x}) - \mathbf{y}^*_k(\mathbf{x}')\| \right) \quad (30)$$
$$\leq \frac{\sqrt{2} L_{g,2}}{\mu_g} \left( 1 + L_y + \frac{C_{xy}(1 + L_y)}{\mu_g} \right) \|\mathbf{x} - \mathbf{x}'\| \quad (31)$$

where in $(a)$ we use

$$\|\nabla^2_{\mathbf{x} \mathbf{y}_k} g_{i,k}(\mathbf{x}_i, \mathbf{y}^*_{i,k}(\mathbf{x}_i)) - \nabla^2_{\mathbf{x} \mathbf{y}_k} g_{i,k}(\mathbf{x}'_i, \mathbf{y}^*_{i,k}(\mathbf{x}'_i))\| \leq L_{g,2} \left( \|\mathbf{x}_i - \mathbf{x}'_i\| + \|\mathbf{y}^*_{i,k}(\mathbf{x}_i) - \mathbf{y}^*_{i,k}(\mathbf{x}'_i)\| \right) \quad (32a)$$
$$\|\nabla^2_{\mathbf{y}_k \mathbf{y}_k} g_{i,k}(\mathbf{x}_i, \mathbf{y}^*_k(\mathbf{x}_i)) - \nabla^2_{\mathbf{y}_k \mathbf{y}_k} g_{i,k}(\mathbf{x}'_i, \mathbf{y}^*_k(\mathbf{x}'_i))\| \leq L_{g,2} \left( \|\mathbf{x}_i - \mathbf{x}'_i\| + \|\mathbf{y}^*_{i,k}(\mathbf{x}_i) - \mathbf{y}^*_{i,k}(\mathbf{x}'_i)\| \right) \quad (32b)$$

by directly applying A1 and (25). $\qquad \square$

**Lemma 3.**

Under A1 and A5, there exists a constant $L_{f,y}$ such that function $\|\overline{\nabla} f_i(\mathbf{x}_i, \mathbf{y}_{i,k}) - \overline{\nabla} f_i(\mathbf{x}_i, \mathbf{y}'_{i,k})\|$ is upper bounded by the sum of $\|\mathbf{y}_{i,k} - \mathbf{y}'_{i,k}\|$, namely
$$\|\overline{\nabla} f_i(\mathbf{x}_i, \mathbf{y}_{1,k}, \dots, \mathbf{y}_{m,k}) - \overline{\nabla} f_i(\mathbf{x}_i, \mathbf{y}'_{1,k}, \dots, \mathbf{y}'_{m,k})\| \leq L_{f,y} \sum_{k=1}^m \|\mathbf{y}_{i,k} - \mathbf{y}'_{i,k}\|, \quad (33)$$
and there also exists a constant $L_{f,x}$ such that $\|\overline{\nabla} f_i(\mathbf{x}_i, \mathbf{y}_{i,k}) - \overline{\nabla} f_i(\mathbf{x}'_i, \mathbf{y}_{i,k})\|$ is upper bounded by $\|\mathbf{x} - \mathbf{x}'\|$, namely
$$\|\overline{\nabla} f_i(\mathbf{x}_i, \mathbf{y}_{1,k}, \dots, \mathbf{y}_{m,k}) - \overline{\nabla} f_i(\mathbf{x}'_i, \mathbf{y}_{1,k}, \dots, \mathbf{y}_{m,k})\| \leq m L_{f,x} \|\mathbf{x}_i - \mathbf{x}'_i\|, \quad (34)$$
where $L_{f,x}, L_{f,y}$ are only dependent on the parameters defined in A1 and A5.

*Proof.* **First Part**. Suppose assumptions A1 and A5 hold. From (23) and [31, Lemma 2.2.], we have

$$\|\nabla^2_{\mathbf{x}_i \mathbf{y}_{i,k}} g_{i,k}(\mathbf{x}_i, \mathbf{y}_{i,k}) \left[\nabla^2_{\mathbf{y}_{i,k} \mathbf{y}_{i,k}} g_{i,k}(\mathbf{x}_i, \mathbf{y}_{i,k})\right]^{-1} \nabla_{\mathbf{y}_{i,k}} f_i(\mathbf{x}_i, \mathbf{y}_{i,k})$$

$$- \nabla^2_{\mathbf{x}_i \mathbf{y}_{i,k}} g_k(\mathbf{x}_i, \mathbf{y}'_{i,k}) \left[\nabla^2_{\mathbf{y}_{i,k} \mathbf{y}_{i,k}} g_{i,k}(\mathbf{x}_i, \mathbf{y}'_{i,k})\right]^{-1} \nabla_{\mathbf{y}_{i,k}} f_i(\mathbf{x}_i, \mathbf{y}'_{i,k})\| \tag{35}$$

$$\leq \left(\frac{L_{f,1} C_{xy}}{\mu_g} + L_{f,0}\left(\frac{L_{g,2}}{\mu_g} + \frac{L_{g,2} C_{xy}}{\mu_g^2}\right)\right) \|\mathbf{y}_{i,k} - \mathbf{y}'_{i,k}\|. \tag{36}$$

Based on the block-wise gradient Lipschitz continuity, we have

$$\left\|\nabla_{\mathbf{x}_i} f_i(\mathbf{x}_i, \mathbf{y}_{i,k}) - \nabla_{\mathbf{x}_i} f_i(\mathbf{x}_i, \mathbf{y}'_{i,k})\right\| \leq \sum_{k=1}^{m} L_{f,1} \|\mathbf{y}_{i,k} - \mathbf{y}'_{i,k}\|. \tag{37}$$

Combing (23) with (36) and (37) gives the desired result immediately.

**Second Part**. Similarly, under A1 and A5, we can also get

$$\left\|\overline{\nabla} f_i(\mathbf{x}_i, \mathbf{y}_{i,k}) - \overline{\nabla} f_i(\mathbf{x}'_i, \mathbf{y}_{i,k})\right\|$$

$$\leq L_{f,1} \|\mathbf{x}_i - \mathbf{x}'_i\| + \frac{m}{\mu_g}\left(L_{f,1} C_{xy} + L_{f,0}\left(L_{g,2} + \frac{L_{g,2} C_{xy}}{\mu_g}\right)\right) \|\mathbf{x}_i - \mathbf{x}'_i\| \tag{38}$$

From (2a), we can get the desired result by requiring $L_{f,x} \geq (L_{f,1} + m\mu_g^{-1}(L_{f,1} C_{xy} + L_{f,0}(L_{g,2} + L_{g,2} C_{xy} \mu_g^{-1})))/m$. □

# B   Convergence Analysis

We now present the proofs, related results, and technical details that establish the lemmas and theorems of our convergence analysis.

We first show that the difference between two successive iterates in Lemma 4 is upper bounded on the order of $1/T$. Then, we begin to quantify the process where one step of the variable updates would make: 1) Lemma 5 measures the closeness from the iterates $\overline{\mathbf{y}}^r$ to its optimal counterpart given the UL variable fixed; 2) Lemma 6 essentially gives the upper bound of the consensus violations from both UL and LL sides or the maximum ascent achieved by the dual update. As the byproducts of Lemma 6, recursion functions in terms of the successive differences of the UL and LL variables are derived in Lemma 7 and Lemma 8 based on the optimality conditions of the UL and LL optimization problems respectively, which serve as the critical role of establishing the potential functions that can evaluate the process of the sequence generated by SLAM to KKT points. Finally, all the above properties are used in the proof of Theorem 1.

## B.1   Upper Bounds of Successive Primal Variables

**Lemma 4.**

*Under A1, A3, A4, suppose that the iterates $\{\mathbf{x}^r, \mathbf{y}_k^r, \forall k, r\}$ are generated by (5a) and (5b) and the step sizes are chosen to be $\alpha = \alpha_0 \sqrt{T}$ and $\beta = \beta_0 \sqrt{T}$. Then, there exist constants $D_x, D_y$ such that, when*

$$\alpha_0 \geq 2\rho\sigma_{\max}(\mathbf{A}^T \mathbf{A})\sqrt{C_x}, \tag{39a}$$

$$\beta_0 \geq \max\{1/\mu_g, \tau_g\}, \tag{39b}$$

$$\tau_g \geq \frac{\rho\sigma_{\max}(\mathbf{A}^T \mathbf{A})}{1 - 1/\sqrt{T}}, \tag{39c}$$

$$\tau_f \geq \rho\sigma_{\max}(\mathbf{A}^T \mathbf{A}), \tag{39d}$$

*the following inequalities hold*

$$\mathbb{E}\|\mathbf{x}^{r+1} - \mathbf{x}^r\|^2 \leq \frac{D_x}{\alpha^2}, \quad \forall r, \tag{40a}$$

$$\frac{1}{T}\sum_{r=1}^{T} \mathbb{E}\|\mathbf{y}_k^{r+1} - \mathbf{y}_k^r\|^2 \leq \frac{D_y}{\beta^2}, \quad \forall k, r, \tag{40b}$$

*where the constants are given by*

$$C_x = \frac{2(1+\theta^{-1})}{\left(1 - (1+\theta)\left(1 - \frac{\rho\widetilde{\sigma}_{\min}(\mathbf{A}^T\mathbf{A})}{\tau_f}\right)^2\right)}, \tag{41a}$$

$$D_x = \frac{4C_x\sigma_f^2}{n} + 4(L_{f,0} + \sigma_f^2) + 2C_x\rho^2\sigma_{\max}^2(\mathbf{A}^T\mathbf{A})B_x, \tag{41b}$$

$$D_y = \frac{\sigma_g^2}{n} + \frac{(D_x\mu_g^{-2} + \mu_g\rho\|\mathbf{y}^1\|^2\alpha_0)\beta_0^2}{\alpha_0^2} + \frac{\sigma_g^2}{n\mu_g\beta_0}, \tag{41c}$$

*and*

$$0 < \theta < \frac{1}{(1 - \frac{\rho\widetilde{\sigma}_{\min}(\mathbf{A}^T\mathbf{A})}{\tau_f})^2} - 1, \tag{42}$$

$$B_x \triangleq \max\left\{\|\mathbf{x}^1\|^2, \frac{\sigma_f^2(2 + 2/\theta) + n(L_{f,0} + \sigma_f^2)(1 - (1+\theta)(1 - \frac{\rho\widetilde{\sigma}_{\min}(\mathbf{A}^T\mathbf{A})}{\tau})^2)}{n(2 + 2/\theta)\rho^2\sigma_{\max}^2(\mathbf{A}^T\mathbf{A})}\right\}. \tag{43}$$

*Proof.* **First Part**. From (6b) and A3, we know that

$$\bar{\mathbf{x}}^{r+1} - \bar{\mathbf{x}}^r = \bar{\mathbf{x}}^r - \bar{\mathbf{x}}^{r-1} + \frac{1}{\alpha}(\bar{\mathbf{h}}_f^r - \bar{\mathbf{h}}_f^{r-1}), \tag{44}$$

where

$$\bar{\mathbf{x}} \triangleq \frac{1}{n}\mathbb{1}^T\mathbf{x}, \qquad \bar{\mathbf{h}} \triangleq \frac{1}{n}\mathbb{1}^T\mathbf{h}. \tag{45}$$

We then derive, from (5b) and (6b), that

$$\|\mathbf{x}^{r+1} - \mathbf{x}^r - (\mathbb{1}\bar{\mathbf{x}}^{r+1} - \mathbb{1}\bar{\mathbf{x}}^r)\|$$
$$\overset{(6b),(44)}{\leq} \left\|\left(\mathbf{I} - (1+\gamma^{-1})\frac{\rho\mathbf{A}^T\mathbf{A}}{\tau}\right)(\mathbf{x}^r - \mathbb{1}\bar{\mathbf{x}}^r) - \left(\mathbf{I} - \frac{\rho\mathbf{A}^T\mathbf{A}}{\tau_f}\right)(\mathbf{x}^{r-1} - \mathbb{1}\bar{\mathbf{x}}^{r-1})\right\|$$
$$+ \frac{1}{\alpha}\left\|\mathbf{h}_f^r - \mathbb{1}\bar{\mathbf{h}}_f^r - (\mathbf{h}_f^{r-1} - \mathbb{1}\bar{\mathbf{h}}_f^{r-1})\right\| \tag{46}$$
$$\leq \left\|\left(\mathbf{I} - \frac{\rho\mathbf{A}^T\mathbf{A}}{\tau_f}\right)(\mathbf{x}^r - \mathbb{1}\bar{\mathbf{x}}^r - (\mathbf{x}^{r-1} - \mathbb{1}\bar{\mathbf{x}}^{r-1}))\right\| + \frac{1}{\alpha}\|\rho\mathbf{A}^T\mathbf{A}\mathbf{x}^r\|$$
$$+ \frac{1}{\alpha}\left\|\mathbf{h}_f^r - \mathbb{1}\bar{\mathbf{h}}_f^r - (\mathbf{h}_f^{r-1} - \mathbb{1}\bar{\mathbf{h}}_f^{r-1})\right\|$$
$$\overset{(a)}{\leq} \left(1 - \frac{\rho\widetilde{\sigma}_{\min}(\mathbf{A}^T\mathbf{A})}{\tau_f}\right)\left\|\mathbf{x}^r - \mathbb{1}\bar{\mathbf{x}}^r - (\mathbf{x}^{r-1} - \mathbb{1}\bar{\mathbf{x}}^{r-1})\right\| + \frac{1}{\alpha}\|\rho\mathbf{A}^T\mathbf{A}\mathbf{x}^r\|$$
$$+ \frac{1}{\alpha}\left\|\mathbf{h}_f^r - \mathbb{1}\bar{\mathbf{h}}_f^r - (\mathbf{h}_f^{r-1} - \mathbb{1}\bar{\mathbf{h}}_f^{r-1})\right\|,$$

where $(a)$ holds because $\mathbb{1}^T(\mathbf{x}^r - \mathbb{1}\bar{\mathbf{x}}^r) = 0, \forall r$, and $\widetilde{\sigma}_{\min}(\mathbf{A}^T\mathbf{A})$ denotes the minimum nonzero eigenvalue of matrix $\mathbf{A}^T\mathbf{A}$. From the definition of $\mathbf{h}_f^r$ and A3, we have

$$\mathbb{E}\|\mathbf{h}_f^r - \mathbb{1}\bar{\mathbf{h}}_f^r\|^2 = \sum_{i=1}^{n}\mathbb{E}\|\mathbf{h}_{f,i}^r - \mathbb{E}\bar{\mathbf{h}}_f^r + \mathbb{E}\bar{\mathbf{h}}_f^r - \bar{\mathbf{h}}_f^r\|^2 \leq \frac{2\sigma_f^2}{n}. \tag{47}$$

Next, we use mathematical induction to prove the boundedness of the variable $\mathbf{x}^r$. When $r = 1$, the size of $\mathbf{x}^1$ is bounded by a constant, i.e., $\|\mathbf{x}^1\|^2$. Applying Young's inequality, we obtain that, for $\theta > 0$,

$$\mathbb{E}\|\mathbf{x}^{r+1} - \mathbf{x}^r - (\mathbb{1}\bar{\mathbf{x}}^{r+1} - \mathbb{1}\bar{\mathbf{x}}^r)\|^2$$
$$\leq (1+\theta)\left(1 - \frac{\rho\widetilde{\sigma}_{\min}(\mathbf{A}^T\mathbf{A})}{\tau_f}\right)^2 \mathbb{E}\left\|\mathbf{x}^r - \mathbb{1}\bar{\mathbf{x}}^r - (\mathbf{x}^{r-1} - \mathbb{1}\bar{\mathbf{x}}^{r-1})\right\|^2$$
$$+ \left(1 + \frac{1}{\theta}\right)\frac{2}{\alpha^2}\left(\frac{2\sigma_f^2}{n} + \rho^2\sigma_{\max}^2(\mathbf{A}^T\mathbf{A})\|\mathbf{x}^r\|^2\right). \tag{48}$$

Define

$$\eta_x \triangleq (1+\theta)\left(1 - \frac{\rho\widetilde{\sigma}_{\min}(\mathbf{A}^T\mathbf{A})}{\tau_f}\right)^2, \qquad \text{and let} \quad \tau_f \geq \rho\widetilde{\sigma}_{\min}(\mathbf{A}^T\mathbf{A}). \tag{49}$$

When

$$\theta < \frac{1}{(1 - \frac{\rho\tilde{\sigma}_{\min}(\mathbf{A}^T\mathbf{A})}{\tau_f})^2} - 1, \tag{50}$$

it is obvious that $\eta_x < 1$. Then, we obtain

$$\mathbb{E}\|\mathbf{x}^{r+1} - \mathbf{x}^r - (\mathbb{1}\bar{\mathbf{x}}^{r+1} - \mathbb{1}\bar{\mathbf{x}}^r)\|^2$$

$$\leq \eta_x \mathbb{E}\left\|\mathbf{x}^r - \mathbf{x}^{r-1} - \left(\mathbb{1}\bar{\mathbf{x}}^r - \mathbb{1}\bar{\mathbf{x}}^{r-1}\right)\right\|^2 + \left(1 + \frac{1}{\theta}\right)\frac{2}{\alpha^2}\left(\frac{2\sigma_f^2}{n} + \rho^2\sigma_{\max}^2(\mathbf{A}^T\mathbf{A})B_1\right) \tag{51}$$

$$\leq \frac{\left(1 + \frac{1}{\theta}\right)\frac{2}{\alpha^2}}{1 - \eta_x}\left(\frac{2\sigma_f^2}{n} + \rho^2\sigma_{\max}^2(\mathbf{A}^T\mathbf{A})B_1\right) = \frac{C_x\left(\frac{2\sigma_f^2}{n} + \rho^2\sigma_{\max}^2(\mathbf{A}^T\mathbf{A})B_1\right)}{\alpha^2}, \tag{52}$$

where we assume that $\|\mathbf{x}^r\|^2 \leq B_1$ and define $C_x \triangleq 2\frac{1+\frac{1}{\theta}}{(1-\eta_x)}$. Further, from (5b) and (4d), we have

$$\bar{\mathbf{x}}^{r+1} = \bar{\mathbf{x}}^r - \frac{1}{\alpha}\bar{\mathbf{h}}_f^r \tag{53}$$

due to $\mathbb{1}^T\mathbf{A}^T\mathbf{A} = 0$, which yields

$$\|\mathbb{1}\bar{\mathbf{x}}^{r+1} - \mathbb{1}\bar{\mathbf{x}}^r\| = \frac{1}{\alpha}\|\mathbb{1}\bar{\mathbf{h}}_f^r\|, \tag{54}$$

and thus we obtain

$$\|\mathbb{1}\bar{\mathbf{x}}^{r+1} - \mathbb{1}\bar{\mathbf{x}}^r\|^2 \leq \frac{1}{\alpha^2}\left(2\|\bar{\mathbf{h}}_f^r - \mathbb{E}\bar{\mathbf{h}}_f^r\|^2 + 2\|\mathbb{E}\bar{\mathbf{h}}_f^r\|^2\right) \leq 2\frac{\sigma_f^2 + L_{f,0}^2}{n\alpha^2} \tag{55}$$

under A1 and A3.

Therefore, combing (52) and (55) renders

$$\mathbb{E}\|\mathbf{x}^{r+1} - \mathbf{x}^r\|^2 = \mathbb{E}\|\mathbf{x}^{r+1} - \mathbb{1}\bar{\mathbf{x}}^{r+1} + \mathbb{1}\bar{\mathbf{x}}^{r+1} - \mathbb{1}\bar{\mathbf{x}}^r + \mathbb{1}\bar{\mathbf{x}}^r - \mathbf{x}^r\|^2$$

$$\leq 2\mathbb{E}\|\mathbf{x}^{r+1} - \mathbb{1}\bar{\mathbf{x}}^{r+1} - (\mathbf{x}^r - \mathbb{1}\bar{\mathbf{x}}^r)\|^2 + 2\mathbb{E}\|\mathbb{1}\bar{\mathbf{x}}^{r+1} - \mathbb{1}\bar{\mathbf{x}}^r\|^2 \tag{56}$$

$$\leq 2\frac{C_x\left(\frac{2\sigma_f^2}{n} + \rho^2\sigma_{\max}^2(\mathbf{A}^T\mathbf{A})B_1\right) + 2(L_{f,0}^2 + \sigma_f^2)/n}{\alpha^2} \tag{57}$$

$$= 2\frac{2C_x\sigma_f^2 + 2(L_{f,0} + \sigma_f^2)}{n\alpha^2} + \frac{2C_x\rho^2\sigma_{\max}^2(\mathbf{A}^T\mathbf{A})B_1}{\alpha^2} \sim \mathcal{O}\left(\frac{1}{\alpha^2}\right), \tag{58}$$

and thus we have

$$\|\mathbf{x}^{r+1}\|^2 \leq r\sum_r \|\mathbf{x}^r\|^2 \leq T\sum_{r=1}^T \|\mathbf{x}^r\|^2 \leq 2\frac{2C_x\sigma_f^2 + 2(L_{f,0} + \sigma_f^2)}{n\alpha_0^2} + \frac{2C_x\rho^2\sigma_{\max}^2(\mathbf{A}^T\mathbf{A})B_1}{\alpha_0^2} \tag{59}$$

where we choose $\alpha = \alpha_0\sqrt{T}$. To show $\|\mathbf{x}^{r+1}\|^2 \leq B_1$, we only need

$$\frac{2C_x\rho^2\sigma_{\max}^2(\mathbf{A}^T\mathbf{A})}{\alpha_0^2} \leq \frac{1}{2} \quad \text{and} \quad \frac{4C_x\sigma_f^2 + 4(L_{f,0} + \sigma_f^2))}{n\alpha_0^2} \leq B_1, \tag{60}$$

which means that

$$\alpha_0 \geq 2\rho\sigma_{\max}(\mathbf{A}^T\mathbf{A})\sqrt{C_x} \quad \text{and} \quad B_1 \geq \frac{C_x\sigma_f^2 + (L_{f,0} + \sigma_f^2)}{n\rho^2\sigma_{\max}^2(\mathbf{A}^T\mathbf{A})C_x}. \tag{61}$$

Therefore, combining the case where $r = 1$, we conclude that, when

$$\alpha_0 \geq 2\rho\sigma_{\max}(\mathbf{A}^T\mathbf{A})\sqrt{\frac{2 + \frac{2}{\theta}}{\left(1 - (1 + \theta)\left(1 - \frac{\rho\tilde{\sigma}_{\min}(\mathbf{A}^T\mathbf{A})}{\tau}\right)^2\right)}}, \tag{62}$$

then

$$\|\mathbf{x}^{r+1}\|^2 \leq \max\left\{\|\mathbf{x}^1\|^2, \frac{\sigma_f^2 + \frac{(L_{f,0} + \sigma_f^2)(1 - (1+\theta)(1 - \frac{\rho\tilde{\sigma}_{\min}(\mathbf{A}^T\mathbf{A})}{\tau})^2)}{2 + 2/\theta}}{n\rho^2\sigma_{\max}^2(\mathbf{A}^T\mathbf{A})}\right\} \triangleq B_x. \tag{63}$$

This directly implies

$$\mathbb{E}\|\mathbf{x}^{r+1} - \mathbf{x}^r\|^2 \leq \overbrace{\frac{4(C_x\sigma_f^2 + L_{f,0} + \sigma_f^2) + 2C_x\rho^2\sigma_{\max}^2(\mathbf{A}^T\mathbf{A})B_x}{\alpha^2}}^{\triangleq D_x} \sim \mathcal{O}\left(\frac{1}{\alpha^2}\right). \tag{64}$$

***Second Part***. From (6a), we have

$$\mathbb{E}[\mathbf{y}_k^{r+1} - \mathbf{y}_k^r] = \mathbb{E}\left[\left(\mathbf{I} - \frac{\rho}{\tau_g}\mathbf{A}^T\mathbf{A}\right)\mathbf{y}_k^r - \left(\mathbf{I} - \frac{\rho}{\tau_g}\mathbf{A}^T\mathbf{A}\right)\mathbf{y}_k^{r-1} - \frac{\rho}{\beta}\mathbf{A}^T\mathbf{A}\mathbf{y}_k^r - \frac{1}{\beta}\left(\mathbf{h}_{g,k}^r - \mathbf{h}_{g,k}^{r-1}\right)\right]. \tag{65}$$

Multiplying $\mathbb{E}[\mathbf{y}_k^{r+1} - \mathbf{y}_k^r]$ on both sides of (65) yields

$$\|\mathbb{E}\mathbf{y}_k^{r+1} - \mathbf{y}_k^r\|^2 = \langle \mathbb{E}[\mathbf{y}_k^r - \mathbf{y}_k^{r-1}], \mathbb{E}[\mathbf{y}_k^{r+1} - \mathbf{y}_k^r]\rangle - \frac{\rho}{\tau_g}\langle \mathbf{A}^T\mathbf{A}(\mathbb{E}[\mathbf{y}_k^r - \mathbf{y}_k^{r-1}]), \mathbb{E}[\mathbf{y}_k^{r+1} - \mathbf{y}_k^r]\rangle$$

$$- \frac{\rho}{\beta}\langle \mathbf{A}^T\mathbf{A}\mathbb{E}[\mathbf{y}_k^r], \mathbb{E}[\mathbf{y}_k^{r+1} - \mathbf{y}_k^r]\rangle - \frac{1}{\beta}\langle g_k(\mathbf{x}^r, \mathbf{y}_k^r) - g_k(\mathbf{x}^{r-1}, \mathbf{y}_k^{r-1}), \mathbb{E}[\mathbf{y}_k^{r+1} - \mathbf{y}_k^r]\rangle$$

$$\leq \frac{\|\mathbb{E}[\mathbf{y}_k^r - \mathbf{y}_k^{r-1}]\|^2}{2} + \frac{\|\mathbb{E}[\mathbf{y}_k^{r+1} - \mathbf{y}_k^r]\|}{2} - \frac{\|\mathbb{E}[\mathbf{v}_k^{r+1}]\|^2}{2}$$

$$- \frac{\rho}{\tau_g}\left(\frac{\|\mathbf{A}\mathbb{E}[\mathbf{y}_k^r - \mathbf{y}_k^{r-1}]\|^2}{2} + \frac{\|\mathbf{A}\mathbb{E}[\mathbf{y}_k^{r+1} - \mathbf{y}_k^r]\|^2}{2} - \frac{\|\mathbf{A}\mathbb{E}[\mathbf{v}_k^{r+1}]\|^2}{2}\right)$$

$$- \frac{\rho}{\beta}\left(\frac{\|\mathbf{A}\mathbb{E}[\mathbf{y}_k^{r+1}]\|^2}{2} - \frac{\|\mathbf{A}\mathbb{E}[\mathbf{y}_k^r]\|^2}{2} - \frac{\|\mathbf{A}\mathbb{E}[\mathbf{y}_k^{r+1} - \mathbf{y}_k^r]\|^2}{2}\right)$$

$$+ \frac{\mathbb{E}\|\mathbf{x}^r - \mathbf{x}^{r-1}\|^2}{\mu_g\beta} + \frac{\mu_g\|\mathbb{E}[\mathbf{y}_k^{r+1} - \mathbf{y}_k^r]\|^2}{4\beta} - \frac{\mu_g}{\beta}\|\mathbb{E}[\mathbf{y}_k^r - \mathbf{y}_k^{r-1}]\|^2$$

$$+ \frac{\sqrt{T}}{2\beta^2}\mathbb{E}\|\mathbf{y}_k^r - \mathbf{y}_k^{r-1}\|^2 + \frac{1}{2\sqrt{T}}\|\mathbb{E}[\mathbf{v}_k^{r+1}]\|^2,$$

where

$$-\frac{1}{\beta}\langle g(\mathbf{x}^{r-1}, \mathbf{y}_k^r) - g(\mathbf{x}^{r-1}, \mathbf{y}_k^{r-1}), \mathbb{E}[\mathbf{y}_k^r - \mathbf{y}_k^{r-1} + \mathbf{v}_k^{r+1}]\rangle$$

$$\overset{(a)}{\leq} -\frac{\mu_g}{\beta}\|\mathbb{E}[\mathbf{y}_k^r - \mathbf{y}_k^{r-1}]\|^2 + \frac{\sqrt{T}}{2\beta^2}\mathbb{E}\|\mathbf{y}_k^r - \mathbf{y}_k^{r-1}\|^2 + \frac{1}{2\sqrt{T}}\|\mathbb{E}\mathbf{v}_k^{r+1}\|^2. \tag{66}$$

Here $(a)$ follows the strong convexity of function $g_k$, Young's inequality, and (17).

Note that

$$\frac{\rho}{\tau_g} - \frac{\rho}{\beta} \geq 0, \tag{67}$$

when $\beta \geq \tau_g$, i.e, $\beta_0\sqrt{T} \geq \tau_g$ or $\beta_0 \geq \tau_g$. Moreover, we know that

$$\mathbb{E}_{\zeta^r}\left[\|\mathbf{y}_k^{r+1} - \mathbf{y}_k^r\|^2|\mathcal{F}^r\right]$$
$$= \|\mathbb{E}_{\zeta^r}[\mathbf{y}_k^{r+1} - \mathbf{y}_k^r]|\mathcal{F}^r\|^2 + \mathbb{E}_{\xi^r}\left[\|\mathbf{y}_k^{r+1} - \mathbf{y}_k^r - \mathbb{E}_{\xi^r}[\mathbf{y}_k^{r+1} - \mathbf{y}_k^r]\|^2|\mathcal{F}^r\right] \tag{68}$$

$$= \|\mathbb{E}_{\zeta^r}[\mathbf{y}_k^{r+1} - \mathbf{y}_k^r]|\mathcal{F}^r\|^2 + \mathbb{E}_{\xi^r}\left[\left\|\frac{1}{\alpha}\left(\mathbb{E}\mathbf{h}_{g,k}^r - \mathbf{h}_{g,k}^r\right)\right\|^2|\mathcal{F}^r\right] \tag{69}$$

$$\leq \|\mathbb{E}_{\zeta^r}[\mathbf{y}_k^{r+1} - \mathbf{y}_k^r]|\mathcal{F}^r\|^2 + \frac{\sigma_g^2}{n\beta^2}. \tag{70}$$

Therefore, we obtain

$$\left(\frac{1}{2} - \frac{\mu_g}{4\beta}\right)\|\mathbb{E}[\mathbf{y}_k^{r+1} - \mathbf{y}_k^r]\|^2$$

$$\leq \left(\frac{1}{2} + \frac{\sqrt{T}}{2\beta^2} - \frac{\mu_g}{\beta}\right)\|\mathbb{E}[\mathbf{y}_k^r - \mathbf{y}_k^{r-1}]\|^2 - \left(\frac{1}{2} - \frac{1}{2\sqrt{T}} - \frac{\rho\sigma_{\max}(\mathbf{A}^T\mathbf{A})}{2\tau_g}\right)\|\mathbb{E}[\mathbf{v}_k^{r+1}]\|^2$$

$$+ \frac{\mathbb{E}\|\mathbf{x}^r - \mathbf{x}^{r-1}\|^2}{\mu_g\beta} - \frac{\rho}{\beta}\left(\frac{\|\mathbf{A}\mathbb{E}[\mathbf{y}_k^{r+1}]\|^2}{2} - \frac{\|\mathbf{A}\mathbb{E}[\mathbf{y}_k^r]\|^2}{2}\right) + \frac{\sigma_g^2\sqrt{T}}{2n\beta^4}.$$

When $\sqrt{T}/\beta \leq \mu_g$, i.e., $\beta_0 \geq 1/\mu_g$, we have $\frac{\sqrt{T}}{2\beta^2} - \frac{\mu_g}{\beta} \leq -\frac{\mu_g}{2\beta}$; further, we need $\tau_g \geq \frac{\rho\sigma_{\max}(\mathbf{A}^T\mathbf{A})}{1-1/\sqrt{T}}$. Then, we obtain

$$\|\mathbb{E}\mathbf{y}_k^{r+1} - \mathbf{y}_k^r\|^2 \leq \frac{1 - \frac{2\mu_g}{\beta}}{1 - \frac{\mu_g}{\beta}}\|\mathbb{E}\mathbf{y}_k^r - \mathbf{y}_k^{r-1}\|^2 + \frac{\mathbb{E}\|\mathbf{x}^r - \mathbf{x}^{r-1}\|^2}{\mu_g\beta} \tag{71}$$

$$- \frac{\rho}{\beta}\left(\frac{\|\mathbf{A}\mathbb{E}\mathbf{y}_k^{r+1}\|^2}{2} - \frac{\|\mathbf{A}\mathbb{E}\mathbf{y}_k^r\|^2}{2}\right) + \frac{\sigma_g^2\sqrt{T}}{2n\beta^4}. \tag{72}$$

Applying the telescoping sum on both sides of (72) yields

$$\frac{1}{T}\sum_{r=1}^T \|\mathbb{E}\mathbf{y}_k^{r+1} - \mathbf{y}_k^r\|^2 \leq \left(1 - \frac{\mu_g}{\beta-\mu_g}\right)\frac{1}{T}\sum_{r=1}^T \|\mathbb{E}\mathbf{y}_k^r - \mathbf{y}_k^{r-1}\|^2 + \frac{D_x}{\mu_g\beta\alpha^2} + \frac{\rho\|\mathbf{y}_k^1\|^2}{T\beta} + \frac{\sigma_g^2\sqrt{T}}{n\beta^4}, \tag{73}$$

and thus there exists a contraction property for the sum of $\|\mathbb{E}[\mathbf{y}_k^r - \mathbf{y}_k^{r-1}]\|^2$. Define $S \triangleq \frac{1}{T}\sum_{r=1}^T \|\mathbb{E}\mathbf{y}_k^r - \mathbf{y}_k^{r-1}\|^2$. When

$$\left(1 - \frac{\mu_g}{\beta-\mu_g}\right)S + \frac{D_x}{\mu_g\beta\alpha^2} + \frac{\rho\|\mathbf{y}_k^1\|^2}{T\beta} + \frac{\sigma_g^2\sqrt{T}}{n\beta^4} \leq S, \tag{74}$$

namely

$$S \geq \frac{\beta - \mu_g}{\mu_g}\left(\frac{D_x}{\mu_g\beta\alpha^2} + \frac{\rho\|\mathbf{y}^1\|^2}{T\beta} + \frac{\sigma_g^2\sqrt{T}}{n\beta^4}\right), \tag{75}$$

then the sum of $\|\mathbb{E}[\mathbf{y}^r - \mathbf{y}^{r-1}]\|^2$ is upper bounded.

Choosing

$$S = \frac{\beta}{\mu_g}\left(\frac{D_x}{\mu_g\beta\alpha^2} + \frac{\rho\|\mathbf{y}_k^1\|^2}{T\beta}\right) = \frac{D_x}{\mu_g^2\alpha^2} + \frac{\mu_g\rho\|\mathbf{y}_k^1\|^2}{T} + \frac{\sigma_g^2\sqrt{T}}{n\mu_g\beta^3}, \tag{76}$$

and applying (70) yield

$$\frac{1}{T}\sum_{r=1}^T \mathbb{E}\|\mathbf{y}_k^{r+1} - \mathbf{y}_k^r\|^2 \leq \frac{D_x}{\mu_g^2\alpha^2} + \frac{\mu_g\rho\|\mathbf{y}_k^1\|^2\alpha_0}{\alpha^2} + \frac{\sigma_g^2}{n\mu_g\beta_0\beta^2} + \frac{\sigma_g^2}{n\beta^2} \tag{77}$$

$$= \frac{D_x/\mu_g^2 + \mu_g\rho\|\mathbf{y}_k^1\|^2\alpha_0}{\alpha^2} + \left(\frac{1}{\mu_g\beta_0} + 1\right)\frac{\sigma_g^2}{n\beta^2}. \tag{78}$$

Combining this with (78) renders

$$\frac{1}{T}\sum_{r=1}^T \mathbb{E}\|\mathbf{y}_k^{r+1} - \mathbf{y}_k^r\|^2 \leq \frac{D_y}{\beta^2}, \quad \forall k, \tag{79}$$

where $D_y \triangleq (D_x/\mu_g^2 + \mu_g\rho\|\mathbf{y}_k^1\|^2\alpha_0)\beta_0^2/\alpha_0^2 + (\frac{1}{\mu_g\beta_0} + 1)\sigma_g^2/n$. $\qquad\square$

## B.2 Contraction of the LL iterates

Now, we will show the contraction property of the recurrence in the LL optimization process. The proof is adapted from [43, Lemma 3], where the main difference is that we evaluate the iterates in the consensus space. To be more precise, we establish the following result.

**Lemma 5.**

*Under A1, A3, A5. Suppose that sequence $\{\mathbf{x}^r, \mathbf{y}_k^r, \boldsymbol{\lambda}_k^r, \boldsymbol{\omega}_k^r, \forall k, r\}$ is generated by SLAM. Then, when $\beta > 2(\mu_g + L_{g,1})^{-1}$, we have*

$$\mathbb{E}\|\bar{\mathbf{y}}_k^{r+1} - \bar{\mathbf{y}}_k^*(\mathbf{x}^r)\|^2 \leq \left(1 - \frac{\rho_g}{\beta}\right)\mathbb{E}\|\bar{\mathbf{y}}_k^r - \bar{\mathbf{y}}_k^*(\mathbf{x}^r)\|^2 + \frac{\sigma_g^2}{n\beta^2}, \tag{80}$$

*and there exist constants $\theta', \vartheta > 0$ such that*

$$\mathbb{E}\|\bar{\mathbf{y}}_k^{r+1} - \bar{\mathbf{y}}_k^*(\mathbf{x}^{r+1})\|^2 \leq \left(1 + \theta' + \frac{\vartheta L_{xy}D_x}{4\alpha^2}\right)\mathbb{E}\|\bar{\mathbf{y}}_k^{r+1} - \bar{\mathbf{y}}_k^*(\mathbf{x}^r)\|^2$$

$$+ \left(L_y^2 + \frac{L_y^2}{4\theta'} + \frac{L_{xy}}{4\vartheta}\right)\|\mathbb{E}[\mathbf{x}^{r+1} - \mathbf{x}^r]\|^2 + \left(\frac{L_{xy}}{4\vartheta} + L_y^2\right)\frac{\sigma_f^2}{n\alpha^2}. \tag{81}$$

*Proof.* First, we expand the error term in the lower level optimization problem as

$$\|\bar{\mathbf{y}}_k^{r+1} - \bar{\mathbf{y}}_k^*(\mathbf{x}^{r+1})\|^2 = \|\bar{\mathbf{y}}_k^{r+1} - \bar{\mathbf{y}}_k^*(\mathbf{x}^r)\|^2 + \|\bar{\mathbf{y}}_k^*(\mathbf{x}^{r+1}) - \bar{\mathbf{y}}_k^*(\mathbf{x}^r)\|^2$$
$$+ 2\left\langle \bar{\mathbf{y}}_k^{r+1} - \bar{\mathbf{y}}_k^*(\mathbf{x}^r), \bar{\mathbf{y}}_k^*(\mathbf{x}^r) - \bar{\mathbf{y}}_k^*(\mathbf{x}^{r+1}) \right\rangle. \tag{82}$$

Then, we will give the upper bound for each term in (82).

$$\mathbb{E}\left[\|\bar{\mathbf{y}}_k^{r+1} - \bar{\mathbf{y}}_k^*(\mathbf{x}^r)\|^2 \mathcal{F}^r\right]$$

$$\overset{(5a)}{=} \mathbb{E}\left[\left\|\bar{\mathbf{y}}_k^{r+1} - \frac{1}{\beta}\bar{\mathbf{h}}_{g,k}^r - \bar{\mathbf{y}}_k^*(\mathbf{x}^r)\right\|^2 \mathcal{F}^r\right] \tag{83}$$

$$= \|\bar{\mathbf{y}}_k^r - \bar{\mathbf{y}}_k^*(\mathbf{x}^r)\|^2 - \frac{2}{\beta}\left\langle \bar{\mathbf{y}}_k^r - \bar{\mathbf{y}}_k^*(\mathbf{x}^r), \mathbb{E}[\bar{\mathbf{h}}_{g,k}^r]|\mathcal{F}^r \right\rangle + \frac{1}{\beta^2}\mathbb{E}[\|\bar{\mathbf{h}}_{g,k}^r\|^2|\mathcal{F}^r] \tag{84}$$

$$\overset{(a)}{\leq} \|\bar{\mathbf{y}}_k^r - \bar{\mathbf{y}}_k^*(\mathbf{x}^r)\|^2 - \frac{2}{\beta}\left\langle \bar{\mathbf{y}}_k^r - \bar{\mathbf{y}}_k^*(\mathbf{x}^r), g_k(\mathbf{x}^r, \bar{\mathbf{y}}_k^r) \right\rangle + \frac{1}{\beta^2}\|\nabla g_k(\mathbf{x}^r, \bar{\mathbf{y}}_k^r)\|^2 + \frac{\sigma_g^2}{n\beta^2} \tag{85}$$

$$\overset{(b)}{\leq} \left(1 - \frac{\rho_g}{\beta}\right)\|\bar{\mathbf{y}}_k^r - \bar{\mathbf{y}}_k^*(\mathbf{x}^r)\|^2 + \frac{\sigma_g^2}{n\beta^2} \tag{86}$$

where in $(a)$ we use A3, $(b)$ follows the $\mu_g$-strong convexity of function $g_k(\mathbf{x}, \mathbf{y}_k)$ [49, Theorem 2.1.11], i.e.,

$$-\left\langle \bar{\mathbf{y}}_k^r - \bar{\mathbf{y}}_k^*(\mathbf{x}^r), g_k(\mathbf{x}^r, \bar{\mathbf{y}}_k) \right\rangle \leq -\frac{\mu_g L_{g,1}}{\mu_g + L_{g,1}}\|\bar{\mathbf{y}}_k^r - \bar{\mathbf{y}}_k^*(\mathbf{x}^r)\|^2$$
$$-\frac{1}{\mu_g + L_{g,1}}\|\nabla g_k(\mathbf{x}^r, \bar{\mathbf{y}}_k^r) - \nabla g_k(\mathbf{x}^r, \bar{\mathbf{y}}_k^*(\mathbf{x}^r))\|^2 \tag{87}$$

and we choose $\beta \geq \frac{2}{\mu_g + L_{g,1}}$ with $\rho_g \triangleq \frac{2\mu_g L_{g,1}}{\mu_g + L_{g,1}}$.

Taking the full expectation over $\mathcal{F}^r$, we have

$$\mathbb{E}\|\bar{\mathbf{y}}_k^{r+1} - \bar{\mathbf{y}}_k^*(\mathbf{x}^r)\|^2 \leq \left(1 - \frac{\rho_g}{\beta}\right)\mathbb{E}\|\bar{\mathbf{y}}_k^r - \bar{\mathbf{y}}_k^*(\mathbf{x}^r)\|^2 + \frac{\sigma_g^2}{n\beta^2}. \tag{88}$$

Next, we split the cross term into two parts

$$\left\langle \bar{\mathbf{y}}_k^{r+1} - \bar{\mathbf{y}}_k^*(\mathbf{x}^r), \bar{\mathbf{y}}_k^*(\mathbf{x}^r) - \bar{\mathbf{y}}_k^*(\mathbf{x}^{r+1}) \right\rangle = -\left\langle \bar{\mathbf{y}}_k^{r+1} - \bar{\mathbf{y}}_k^*(\mathbf{x}^r), \nabla\bar{\mathbf{y}}_k^*(\mathbf{x}^r)(\mathbf{x}^{r+1} - \mathbf{x}^r) \right\rangle$$
$$-\left\langle \bar{\mathbf{y}}_k^{r+1} - \bar{\mathbf{y}}_k^*(\mathbf{x}^r), \bar{\mathbf{y}}_k^*(\mathbf{x}^{r+1}) - \bar{\mathbf{y}}_k^*(\mathbf{x}^r) - \nabla\bar{\mathbf{y}}_k^*(\mathbf{x}^r)(\mathbf{x}^{r+1} - \mathbf{x}^r) \right\rangle. \tag{89}$$

The first part can be upper bounded by

$$-\mathbb{E}\left[\left\langle \bar{\mathbf{y}}_k^{r+1} - \bar{\mathbf{y}}_k^*(\mathbf{x}^r), \nabla\bar{\mathbf{y}}_k^*(\mathbf{x}^r)(\mathbf{x}^{r+1} - \mathbf{x}^r) \right\rangle\right]$$

$$= -\mathbb{E}\left[\left\langle \bar{\mathbf{y}}_k^{r+1} - \bar{\mathbf{y}}_k^*(\mathbf{x}^r), \nabla\bar{\mathbf{y}}_k^*(\mathbf{x}^r)\mathbb{E}[\mathbf{x}^{r+1} - \mathbf{x}^r]\right\rangle|\mathcal{F}'^r\right] \tag{90}$$

$$\overset{(25)}{\leq} L_y\mathbb{E}\|\bar{\mathbf{y}}_k^{r+1} - \bar{\mathbf{y}}_k^*(\mathbf{x}^r)\|\|\mathbb{E}\mathbf{x}^{r+1} - \mathbf{x}^r\| \tag{91}$$

$$\overset{(a)}{\leq} \theta'\mathbb{E}\|\bar{\mathbf{y}}_k^{r+1} - \bar{\mathbf{y}}_k^*(\mathbf{x}^r)\|^2 + \frac{L_y^2}{4\theta'}\|\mathbb{E}\mathbf{x}^{r+1} - \mathbf{x}^r\|^2. \tag{92}$$

where in $(a)$ we apply Young's inequality.

The second part can be upper bounded by

$$-\left\langle \bar{\mathbf{y}}_k^{r+1} - \bar{\mathbf{y}}_k^*(\mathbf{x}^r), \bar{\mathbf{y}}_k^*(\mathbf{x}^{r+1}) - \bar{\mathbf{y}}_k^*(\mathbf{x}^r) - \nabla\bar{\mathbf{y}}_k^*(\mathbf{x}^r)(\mathbf{x}^{r+1} - \mathbf{x}^r) \right\rangle$$

$$\leq \mathbb{E}\|\bar{\mathbf{y}}_k^{r+1} - \bar{\mathbf{y}}_k^*(\mathbf{x}^r)\|\|\bar{\mathbf{y}}_k^*(\mathbf{x}^{r+1}) - \bar{\mathbf{y}}_k^*(\mathbf{x}^r) - \nabla\bar{\mathbf{y}}_k^*(\mathbf{x}^r)(\mathbf{x}^{r+1} - \mathbf{x}^r)\| \tag{93}$$

$$\overset{(a)}{\leq} \frac{L_{xy}}{2}\mathbb{E}\|\bar{\mathbf{y}}_k^{r+1} - \bar{\mathbf{y}}_k^*(\mathbf{x}^r)\|\mathbb{E}\left[\|\mathbf{x}^{r+1} - \mathbf{x}^r\|^2|\mathcal{F}'^r\right] \tag{94}$$

$$\leq \frac{\vartheta L_{xy}}{4}\mathbb{E}\left[\|\bar{\mathbf{y}}_k^{r+1} - \bar{\mathbf{y}}_k^*(\mathbf{x}^r)\|^2\mathbb{E}\left[\|\mathbf{x}^{r+1} - \mathbf{x}^r\|^2|\mathcal{F}'^r\right]\right] + \frac{L_{xy}}{4\vartheta}\mathbb{E}\left[\|\mathbf{x}^{r+1} - \mathbf{x}^r\|^2|\mathcal{F}'^r\right] \tag{95}$$

$$\overset{(b)}{\leq} \frac{\vartheta D_x L_{xy}}{4\alpha^2}\mathbb{E}\left[\|\bar{\mathbf{y}}_k^{r+1} - \bar{\mathbf{y}}_k^*(\mathbf{x}^r)\|^2\right] + \frac{L_{xy}}{4\vartheta}\left(\left[\|\mathbb{E}\mathbf{x}^{r+1} - \mathbf{x}^r\|^2\right] + \frac{\sigma_f^2}{n\alpha^2}\right) \tag{96}$$

where $(a)$ follows the Lipschitz continuity of $\nabla \mathbf{y}_k^*(\mathbf{x})$ shown in (26), in $(b)$ we apply Lemma 4, and

$$\mathbb{E}_{\xi^r}\left[\|\mathbf{x}^{r+1} - \mathbf{x}^r\|^2 | \mathcal{F}'^r\right]$$

$$= \|\mathbb{E}_{\xi^r}[\mathbf{x}^{r+1} - \mathbf{x}^r]|\mathcal{F}'^r\|^2 + \mathbb{E}_{\xi^r}\left[\|\mathbf{x}^{r+1} - \mathbf{x}^r - \mathbb{E}_{\xi^r}[\mathbf{x}^{r+1} - \mathbf{x}^r]\|^2 | \mathcal{F}'^r\right] \tag{97}$$

$$= \|\mathbb{E}_{\xi^r}[\mathbf{x}^{r+1} - \mathbf{x}^r]|\mathcal{F}'^r\|^2 + \mathbb{E}_{\xi^r}\left[\left\|\frac{1}{\alpha}\left(\mathbb{E}\mathbf{h}_f^r - \mathbf{h}_f^r\right)\right\|^2 | \mathcal{F}'^r\right] \tag{98}$$

$$\leq \|\mathbb{E}_{\xi^r}[\mathbf{x}^{r+1} - \mathbf{x}^r]|\mathcal{F}'^r\|^2 + \frac{\sigma_f^2}{n\alpha^2}. \tag{99}$$

Combining (92) and (96), we can have

$$\mathbb{E}\|\bar{\mathbf{y}}_k^{r+1} - \bar{\mathbf{y}}_k^*(\mathbf{x}^{r+1})\|^2 \leq \left(1 + \theta' + \frac{\vartheta L_{xy} D_x}{4\alpha^2}\right)\mathbb{E}\|\bar{\mathbf{y}}_k^{r+1} - \bar{\mathbf{y}}_k^*(\mathbf{x}^r)\|^2$$

$$+ \left(L_y^2 + \frac{L_y^2}{4\theta'} + \frac{L_{xy}}{4\vartheta}\right)\|\mathbb{E}\left[\mathbf{x}^{r+1} - \mathbf{x}^r\right]\|^2 + \left(\frac{L_{xy}}{4\vartheta} + L_y^2\right)\frac{\sigma_f^2}{n\alpha^2}. \tag{100}$$

where we use the gradient Lipschitz continuity, i.e., $\mathbb{E}\|\bar{\mathbf{y}}_k^*(\mathbf{x}^{r+1}) - \bar{\mathbf{y}}_k^*(\mathbf{x}^r)\|^2 \leq L_y^2\mathbb{E}\|\mathbf{x}^{r+1} - \mathbf{x}^r\|^2$. $\qquad\square$

## B.3 Upper Bound of Successive Dual Variables

In what follows, we will show the ascent part after one round of the dual variable update.

**Lemma 6.**

*Under A1-A5, define $\mathbf{D} \triangleq \alpha\mathbf{I} - \rho\gamma\mathbf{A}^T\mathbf{A}$. Suppose that the sequence $\{\mathbf{x}^r, \mathbf{y}_k^r, \boldsymbol{\lambda}^r, \boldsymbol{\omega}_k^r, \forall k\}$ is generated by SLAM. Then, we have*

$$\frac{\gamma}{\rho}\|\mathbb{E}\boldsymbol{\lambda}^{r+1} - \boldsymbol{\lambda}^r\|^2 \leq \frac{4m^2 L_{f,x}^2 D_x}{n^2\rho\gamma\alpha^2\tilde{\sigma}_{\min}(\mathbf{A}^T\mathbf{A})} + \frac{4\left\|\mathbb{E}\mathbf{w}^{r+1}\right\|_{\mathbf{D}^T\mathbf{D}}^2}{\rho\gamma\tilde{\sigma}_{\min}(\mathbf{A}^T\mathbf{A})}$$

$$+ \frac{4mL_{f,y}^2\sum_{k=1}^m\mathbb{E}\left\|\mathbf{y}_k^{r+1} - \mathbf{y}_k^r\right\|^2}{n^2\rho\gamma\tilde{\sigma}_{\min}(\mathbf{A}^T\mathbf{A})} + \frac{4(b_r + b_{r-1})^2}{\rho\gamma\tilde{\sigma}_{\min}(\mathbf{A}^T\mathbf{A})}, \tag{101a}$$

$$\frac{\gamma}{\rho}\|\mathbb{E}\boldsymbol{\omega}_k^{r+1} - \boldsymbol{\omega}_k^r\|^2 \leq \frac{3\left\|\mathbb{E}\mathbf{v}_k^{r+1}\right\|_{\mathbf{D}^T\mathbf{D}}^2}{\rho\gamma\tilde{\sigma}_{\min}(\mathbf{A}^T\mathbf{A})} + \frac{3L_{g,1}^2 D_x}{n^2\rho\gamma\alpha^2\tilde{\sigma}_{\min}(\mathbf{A}^T\mathbf{A})} + \frac{3L_{g,1}^2\mathbb{E}\left\|\mathbf{y}_k^{r+1} - \mathbf{y}_k^r\right\|^2}{n^2\rho\gamma\tilde{\sigma}_{\min}(\mathbf{A}^T\mathbf{A})} \tag{101b}$$

*where $\mathbf{w}^{r+1} \triangleq (\mathbf{x}^{r+1} - \mathbf{x}^r) - (\mathbf{x}^r - \mathbf{x}^{r-1})$ and $\mathbf{v}_k^{r+1} \triangleq (\mathbf{y}_k^{r+1} - \mathbf{y}_k^r) - (\mathbf{y}_k^r - \mathbf{y}_k^{r-1}), \forall k$.*

*Proof.* **First Part**. First, by utilizing the optimality condition of (5b), we have

$$\mathbf{h}_f^r + \gamma\mathbf{A}^T\boldsymbol{\lambda}^{r+1} + \rho\gamma\mathbf{A}^T\mathbf{A}(\mathbf{x}^r - \mathbf{x}^{r+1}) + \alpha(\mathbf{x}^{r+1} - \mathbf{x}^r) = 0. \tag{102}$$

Subtracting the above equality with the same one from the previous iteration, we obtain

$$\mathbf{h}_f^r - \mathbf{h}_f^{r-1} + \gamma\mathbf{A}^T(\boldsymbol{\lambda}^{r+1} - \boldsymbol{\lambda}^r) + \rho\gamma\mathbf{A}^T\mathbf{A}\left((\mathbf{x}^r - \mathbf{x}^{r+1}) - (\mathbf{x}^{r-1} - \mathbf{x}^r)\right)$$

$$+ \alpha\left((\mathbf{x}^{r+1} - \mathbf{x}^r) - (\mathbf{x}^r - \mathbf{x}^{r-1})\right) = 0. \tag{103}$$

Let $\mathbf{w}^{r+1} \triangleq (\mathbf{x}^{r+1} - \mathbf{x}^r) - (\mathbf{x}^r - \mathbf{x}^{r-1})$. We can easily write (103) concisely as

$$\mathbf{h}_f^r - \mathbf{h}_f^{r-1} + \gamma\mathbf{A}^T(\boldsymbol{\lambda}^{r+1} - \boldsymbol{\lambda}^r) + (\alpha - \rho\gamma\mathbf{A}^T\mathbf{A})\mathbf{w}^{r+1} = 0. \tag{104}$$

According to A3, taking expectation on both sides of the above equation, we have

$$\overline{\nabla}f(\mathbf{x}^r, \mathbf{y}_k^{r+1}) - \overline{\nabla}f(\mathbf{x}^r, \mathbf{y}_k^r) + b_r' + b_{r-1}' + \overline{\nabla}f(\mathbf{x}^r, \mathbf{y}_k^r) - \overline{\nabla}f(\mathbf{x}^{r-1}, \mathbf{y}_k^r)$$

$$+ \mathbb{E}[\gamma\mathbf{A}^T(\boldsymbol{\lambda}^{r+1} - \boldsymbol{\lambda}^r)] + \mathbb{E}[(\alpha - \rho\gamma\mathbf{A}^T\mathbf{A})\mathbf{w}^{r+1}] = 0 \tag{105}$$

where $0 < b_r' \leq b_r, 0 < b_{r-1}' \leq b_{r-1}$.

Utilizing the fact that $\boldsymbol{\lambda}^{r+1} - \boldsymbol{\lambda}^r$ lies in the column space of $\mathbf{A}$, we have $\|\mathbf{A}^T(\boldsymbol{\lambda}^{r+1} - \boldsymbol{\lambda}^r)\|^2 \geq \tilde{\sigma}_{\min}(\mathbf{A}^T\mathbf{A})\|\boldsymbol{\lambda}^{r+1} - \boldsymbol{\lambda}^r\|^2$. After applying the triangle inequality, it is easy to see that the following in-

equality is true

$$\frac{\gamma}{\rho}\|\mathbb{E}\boldsymbol{\lambda}^{r+1} - \boldsymbol{\lambda}^r\|^2$$

$$\leq \frac{4}{\rho\gamma\widetilde{\sigma}_{\min}(\mathbf{A}^T\mathbf{A})} \left\|\mathbb{E}[\overline{\nabla} f(\mathbf{x}^r, \mathbf{y}_k^r) - \overline{\nabla} f(\mathbf{x}^{r-1}, \mathbf{y}_k^r)]\right\|^2 + \frac{4}{\rho\gamma\widetilde{\sigma}_{\min}(\mathbf{A}^T\mathbf{A})} \left\|\mathbb{E}\mathbf{w}^{r+1}\right\|_{\mathbf{D}^T\mathbf{D}}^2$$

$$+ \frac{4}{\rho\gamma\widetilde{\sigma}_{\min}(\mathbf{A}^T\mathbf{A})} \left\|\mathbb{E}[\overline{\nabla} f(\mathbf{x}^r, \mathbf{y}_k^{r+1}) - \overline{\nabla} f(\mathbf{x}^r, \mathbf{y}_k^r)]\right\|^2 + \frac{4(b_r' + b_{r-1}')^2}{\rho\gamma\widetilde{\sigma}_{\min}(\mathbf{A}^T\mathbf{A})} \tag{106}$$

$$\overset{(a)}{\leq} \frac{4m^2 L_{f,x}^2}{n^2 \gamma\rho\widetilde{\sigma}_{\min}(\mathbf{A}^T\mathbf{A})} \mathbb{E}\left\|\mathbf{x}^r - \mathbf{x}^{r-1}\right\|^2 + \frac{4}{\rho\gamma\widetilde{\sigma}_{\min}(\mathbf{A}^T\mathbf{A})} \left\|\mathbb{E}\mathbf{w}^{r+1}\right\|_{\mathbf{D}^T\mathbf{D}}^2$$

$$+ \frac{4 L_{f,y}^2 m}{n^2 \rho\gamma\widetilde{\sigma}_{\min}(\mathbf{A}^T\mathbf{A})} \sum_{k=1}^m \mathbb{E}\left\|\mathbf{y}_k^{r+1} - \mathbf{y}_k^r\right\|^2 + \frac{4(b_r' + b_{r-1}')^2}{\rho\gamma\widetilde{\sigma}_{\min}(\mathbf{A}^T\mathbf{A})}$$

$$\overset{(b)}{\leq} \frac{4m^2 L_{f,x}^2 D_x}{n^2 \rho\gamma\alpha^2 \widetilde{\sigma}_{\min}(\mathbf{A}^T\mathbf{A})} + \frac{4\left\|\mathbb{E}\mathbf{w}^{r+1}\right\|_{\mathbf{D}^T\mathbf{D}}^2}{\rho\gamma\widetilde{\sigma}_{\min}(\mathbf{A}^T\mathbf{A})} + \frac{4m L_{f,y}^2 \sum_{k=1}^m \mathbb{E}\left\|\mathbf{y}_k^{r+1} - \mathbf{y}_k^r\right\|^2}{n^2 \rho\gamma\widetilde{\sigma}_{\min}(\mathbf{A}^T\mathbf{A})} + \frac{4(b_r + b_{r-1})^2}{\rho\gamma\widetilde{\sigma}_{\min}(\mathbf{A}^T\mathbf{A})}$$

where in $(a)$ we use gradient Lipschitz continuity of the UL loss function w.r.t. $\mathbf{x}$ and $\mathbf{y}_k$, i.e., (33) and (34), and $(b)$ is true by applying Lemma 4.

***Second Part***. Utilizing the optimality condition of (5a), we have

$$\mathbf{h}_{g,k}^r + \gamma\mathbf{A}^T\boldsymbol{\omega}_k^{r+1} + \rho\gamma\mathbf{A}^T\mathbf{A}(\mathbf{y}_k^r - \mathbf{y}_k^{r+1}) + \alpha(\mathbf{y}_k^{r+1} - \mathbf{y}_k^r) = 0. \tag{107}$$

Similarly, we have

$$\nabla g_k(\mathbf{x}^r, \mathbf{y}_k^{r+1}) - \nabla g_k(\mathbf{x}^r, \mathbf{y}_k^r) + \nabla g_k(\mathbf{x}^r, \mathbf{y}_k^r) - \nabla g_k(\mathbf{x}^{r-1}, \mathbf{y}_k^r)$$
$$+ \gamma\mathbb{E}[\mathbf{A}^T(\boldsymbol{\omega}_k^{r+1} - \boldsymbol{\omega}_k^r)] + \mathbb{E}[(\beta - \rho\gamma\mathbf{A}^T\mathbf{A})\mathbf{v}_k^{r+1}] = 0. \tag{108}$$

where we have defined $\mathbf{v}_k^{r+1} \triangleq (\mathbf{y}_k^{r+1} - \mathbf{y}_k^r) - (\mathbf{y}_k^r - \mathbf{y}_k^{r-1})$.

Following (106), we have

$$\frac{\gamma}{\rho} \left\|\mathbb{E}\left[\boldsymbol{\omega}_k^{r+1} - \boldsymbol{\omega}_k^r\right]\right\|^2$$

$$\leq \frac{3}{\rho\gamma\widetilde{\sigma}_{\min}(\mathbf{A}^T\mathbf{A})} \left\|\mathbb{E}[\nabla g_k(\mathbf{x}^r, \mathbf{y}_k^r) - \nabla g_k(\mathbf{x}^{r-1}, \mathbf{y}_k^r)]\right\|^2 + \frac{3}{\rho\gamma\widetilde{\sigma}_{\min}(\mathbf{A}^T\mathbf{A})} \left\|\mathbb{E}\mathbf{v}_k^{r+1}\right\|_{\mathbf{D}^T\mathbf{D}}^2$$

$$+ \frac{3}{\rho\gamma\widetilde{\sigma}_{\min}(\mathbf{A}^T\mathbf{A})} \left\|\mathbb{E}[\nabla g_k(\mathbf{x}^r, \mathbf{y}_k^{r+1}) - \nabla g_k(\mathbf{x}^r, \mathbf{y}_k^r)]\right\|^2 \tag{109}$$

$$\overset{(a)}{\leq} \frac{3 L_{g,1}^2}{n^2 \rho\gamma\widetilde{\sigma}_{\min}(\mathbf{A}^T\mathbf{A})} \mathbb{E}\left\|\mathbf{x}^r - \mathbf{x}^{r-1}\right\|^2 + \frac{3}{\rho\gamma\widetilde{\sigma}_{\min}(\mathbf{A}^T\mathbf{A})} \left\|\mathbb{E}\mathbf{v}_k^{r+1}\right\|_{\mathbf{D}^T\mathbf{D}}^2$$

$$+ \frac{3 L_{g,1}^2}{n^2 \rho\gamma\widetilde{\sigma}_{\min}(\mathbf{A}^T\mathbf{A})} \mathbb{E}\left\|\mathbf{y}_k^{r+1} - \mathbf{y}_k^r\right\|^2,$$

which gives (101b) directly by applying Lemma 4. $\qquad\square$

## B.4 Recursive Functions

Now, the ascent part measured by the successive difference of the dual variables is upper bounded w.r.t. $\|\mathbb{E}[\mathbf{w}^{r+1}]\|^2$. Using (5b), we can construct the following recursion that establishes descent w.r.t. $\|\mathbb{E}[\mathbf{w}^{r+1}]\|^2$.

**Lemma 7.**

*Under A1-A4, suppose that the sequence is generated by SLAM. Then, there exists a constant $\vartheta > 0$ such that*

$$\mathcal{Q}_w^{r+1} - \mathcal{Q}_w^r \leq -\frac{1}{2}\|\mathbb{E}[\mathbf{w}^{r+1}]\|_{\mathbf{D}}^2 + \left(\frac{L_{f,x} + 1}{2n} + \frac{L_{f,x} m^2}{n}\right)\|\mathbb{E}[\mathbf{x}^{r+1} - \mathbf{x}^r]\|^2$$

$$+ \frac{m L_{f,y}^2}{L_{f,x} n} \sum_{k=1}^m \mathbb{E}\|\mathbf{y}_k^{r+1} - \mathbf{y}_k^r\|^2 + \frac{n(b_r + b_{r-1})^2}{2} + \frac{L_{f,x} m^2 \sigma_f^2}{n\alpha^2} \tag{110}$$

*where*

$$\mathcal{Q}_w^r \triangleq \frac{\rho}{2\sqrt{\gamma}}\|\mathbf{A}\mathbb{E}[\mathbf{x}^r]\|^2 + \frac{1}{2}\|\mathbb{E}[\mathbf{x}^r - \mathbf{x}^{r-1}]\|_{\mathbf{D}}^2 + \frac{L_{f,x} m^2}{n}\mathbb{E}\|\mathbf{x}^r - \mathbf{x}^{r-1}\|^2 \geq 0. \tag{111}$$

*Proof.* We have the following optimality condition for the **x**-update step:

$$\mathbb{E}[\mathbf{h}_f^r + \gamma \mathbf{A}^T \boldsymbol{\lambda}^r + \rho\gamma \mathbf{A}^T \mathbf{A}\mathbf{x}^r + \alpha(\mathbf{x}^{r+1} - \mathbf{x}^r)] = 0, \tag{112a}$$

$$\mathbb{E}[\mathbf{h}_f^{r-1} + \gamma \mathbf{A}^T \boldsymbol{\lambda}^{r-1} + \rho\gamma \mathbf{A}^T \mathbf{A}\mathbf{x}^{r-1} + \alpha(\mathbf{x}^r - \mathbf{x}^{r-1})] = 0. \tag{112b}$$

Hence, we have

$$\overline{\mathbf{h}}_f^r + \gamma \mathbf{A}^T \mathbb{E}[\boldsymbol{\lambda}^r] + \rho\gamma \mathbf{A}^T \mathbf{A}\mathbb{E}[\mathbf{x}^r] + \alpha\mathbb{E}[\mathbf{x}^{r+1} - \mathbf{x}^r] = 0, \tag{113a}$$

$$\overline{\mathbf{h}}_f^{r-1} + \gamma \mathbf{A}^T \mathbb{E}[\boldsymbol{\lambda}^{r-1}] + \rho\gamma \mathbf{A}^T \mathbf{A}\mathbb{E}[\mathbf{x}^{r-1}] + \alpha\mathbb{E}[\mathbf{x}^r - \mathbf{x}^{r-1}] = 0. \tag{113b}$$

Although $\overline{\mathbf{h}}_f^r$ is a biased gradient estimate, we can have

$$\overline{\nabla}f(\mathbf{x}^r, \mathbf{y}_k^{r+1}) + b_r' + \gamma \mathbf{A}^T \mathbb{E}[\boldsymbol{\lambda}^r] + \rho\gamma \mathbf{A}^T \mathbf{A}\mathbb{E}[\mathbf{x}^r] + \alpha\mathbb{E}[\mathbf{x}^{r+1} - \mathbf{x}^r] = 0, \tag{114a}$$

$$\overline{\nabla}f(\mathbf{x}^{r-1}, \mathbf{y}_k^r) + b_{r-1}' + \gamma \mathbf{A}^T \mathbb{E}[\boldsymbol{\lambda}^{r-1}] + \rho\gamma \mathbf{A}^T \mathbf{A}\mathbb{E}[\mathbf{x}^{r-1}] + \alpha\mathbb{E}[\mathbf{x}^r - \mathbf{x}^{r-1}] = 0. \tag{114b}$$

Multiplying $\mathbb{E}[\mathbf{x}^r - \mathbf{x}^{r+1}]$ on both sides of (113a), we get

$$\overline{\nabla}f(\mathbf{x}^r, \mathbf{y}_k^{r+1})\mathbb{E}[\mathbf{x}^r - \mathbf{x}^{r+1}] + \gamma \mathbf{A}^T \mathbb{E}[\boldsymbol{\lambda}^r]\mathbb{E}[\mathbf{x}^r - \mathbf{x}^{r+1}] + b_r'\mathbb{E}[\mathbf{x}^r - \mathbf{x}^{r+1}]$$
$$+ \rho\gamma \mathbf{A}^T \mathbf{A}\mathbb{E}[\mathbf{x}^r]\mathbb{E}[\mathbf{x}^r - \mathbf{x}^{r+1}] + \alpha\mathbb{E}[\mathbf{x}^{r+1} - \mathbf{x}^r]\mathbb{E}[\mathbf{x}^r - \mathbf{x}^{r+1}] = 0, \tag{115}$$

and similarly multiplying $\mathbb{E}[\mathbf{x}^{r+1} - \mathbf{x}^r]$ on both sides of (113b), we can further have

$$\overline{\nabla}f(\mathbf{x}^{r-1}, \mathbf{y}_k^r)\mathbb{E}[\mathbf{x}^{r+1} - \mathbf{x}^r] + \gamma \mathbf{A}^T \mathbb{E}[\boldsymbol{\lambda}^{r-1}]\mathbb{E}[\mathbf{x}^{r+1} - \mathbf{x}^r] + b_{r-1}'\mathbb{E}[\mathbf{x}^{r+1} - \mathbf{x}^r]$$
$$+ \rho\gamma \mathbf{A}^T \mathbf{A}\mathbb{E}[\mathbf{x}^{r-1}]\mathbb{E}[\mathbf{x}^{r+1} - \mathbf{x}^r] + \alpha\mathbb{E}[\mathbf{x}^r - \mathbf{x}^{r-1}]\mathbb{E}[\mathbf{x}^{r+1} - \mathbf{x}^r] = 0. \tag{116}$$

Plugging (4d) into the above two inequalities, we obtain

$$\frac{\rho}{\sqrt{\gamma}}\langle \mathbf{A}^T \mathbf{A}\mathbb{E}[\mathbf{x}^r], \mathbb{E}[\mathbf{x}^{r+1} - \mathbf{x}^r]\rangle$$
$$= \gamma\langle \mathbf{A}^T \mathbb{E}[\boldsymbol{\lambda}^r - \boldsymbol{\lambda}^{r-1}], \mathbb{E}[\mathbf{x}^{r+1} - \mathbf{x}^r]\rangle \tag{117}$$
$$\leq \langle \overline{\nabla}f(\mathbf{x}^{r-1}, \mathbf{y}_k^r) - \overline{\nabla}f(\mathbf{x}^r, \mathbf{y}_k^{r+1}) - \mathbf{D}\mathbb{E}[\mathbf{w}^{r+1}], \mathbb{E}[\mathbf{x}^{r+1} - \mathbf{x}^r]\rangle + (b_r' - b_{r-1}')\mathbb{E}[\mathbf{x}^r - \mathbf{x}^{r+1}],$$

which gives

$$\frac{\rho}{2\sqrt{\gamma}}\|\mathbf{A}\mathbb{E}[\mathbf{x}^{r+1}]\|^2 + \frac{1}{2}\|\mathbb{E}[\mathbf{x}^{r+1} - \mathbf{x}^r]\|_{\mathbf{D}}^2$$

$$\leq \frac{\rho}{2\sqrt{\gamma}}\|\mathbf{A}\mathbb{E}[\mathbf{x}^r]\|^2 + \frac{1}{2}\|\mathbb{E}[\mathbf{x}^r - \mathbf{x}^{r-1}]\|_{\mathbf{D}}^2 - \frac{1}{2}\|\mathbb{E}[\mathbf{w}^{r+1}]\|_{\mathbf{D}}^2$$
$$+ \langle \overline{\nabla}f(\mathbf{x}^{r-1}, \mathbf{y}_k^r) - \overline{\nabla}f(\mathbf{x}^r, \mathbf{y}_k^{r+1}), \mathbb{E}[\mathbf{x}^{r+1} - \mathbf{x}^r]\rangle + (b_r' - b_{r-1}')\mathbb{E}[\mathbf{x}^r - \mathbf{x}^{r+1}] \tag{118}$$

$$\leq \frac{\rho}{2\sqrt{\gamma}}\|\mathbf{A}\mathbb{E}[\mathbf{x}^r]\|^2 + \frac{1}{2}\|\mathbb{E}[\mathbf{x}^r - \mathbf{x}^{r-1}]\|_{\mathbf{D}}^2 - \frac{1}{2}\|\mathbb{E}[\mathbf{w}^{r+1}]\|_{\mathbf{D}}^2$$

$$+ \frac{n}{2L_{f,x}}\mathbb{E}\|\overline{\nabla}f(\mathbf{x}^{r-1}, \mathbf{y}_k^r) - \overline{\nabla}f(\mathbf{x}^r, \mathbf{y}_k^{r+1})\|^2 + \frac{L_{f,x}}{2n}\|\mathbb{E}[\mathbf{x}^{r+1} - \mathbf{x}^r]\|^2 + (b_r' - b_{r-1}')\mathbb{E}[\mathbf{x}^r - \mathbf{x}^{r+1}]$$

$$\overset{(a)}{\leq} \frac{\rho}{2\sqrt{\gamma}}\|A\mathbb{E}[\mathbf{x}^r]\|^2 + \frac{1}{2}\|\mathbb{E}[\mathbf{x}^r - \mathbf{x}^{r-1}]\|_{\mathbf{D}}^2 - \frac{1}{2}\|\mathbb{E}[\mathbf{w}^{r+1}]\|_{\mathbf{D}}^2$$

$$+ \frac{L_{f,x}m^2}{n}\mathbb{E}\|\mathbf{x}^r - \mathbf{x}^{r-1}\|^2 + \frac{mL_{f,y}^2}{nL_{f,x}}\sum_{k=1}^m \mathbb{E}\|\mathbf{y}_k^{r+1} - \mathbf{y}_k^r\|^2 + \frac{L_{f,x}}{2n}\|\mathbb{E}[\mathbf{x}^{r+1} - \mathbf{x}^r]\|^2$$

$$+ \frac{n(b_r' - b_{r-1}')^2}{2} + \frac{\|\mathbb{E}[\mathbf{x}^r - \mathbf{x}^{r+1}]\|^2}{2n} \tag{119}$$

where (a) follows gradient Lipschitz continuity.

Therefore, we have

$$\frac{\rho}{2\sqrt{\gamma}}\|\mathbf{A}\mathbb{E}[\mathbf{x}^{r+1}]\|^2 + \frac{1}{2}\|\mathbb{E}[\mathbf{x}^{r+1} - \mathbf{x}^r]\|_{\mathbf{D}}^2 + \frac{L_{f,x}m^2}{n}\mathbb{E}\|\mathbf{x}^{r+1} - \mathbf{x}^r\|^2$$

$$\leq \frac{\rho}{2\sqrt{\gamma}}\|\mathbf{A}\mathbb{E}[\mathbf{x}^r]\|^2 + \frac{1}{2}\|\mathbb{E}[\mathbf{x}^r - \mathbf{x}^{r-1}]\|_{\mathbf{D}}^2 + \frac{L_{f,x}m^2}{n}\mathbb{E}\|\mathbf{x}^r - \mathbf{x}^{r-1}\|^2$$

$$- \frac{1}{2}\|\mathbb{E}[\mathbf{w}^{r+1}]\|_{\mathbf{D}}^2 + \left(\frac{L_{f,x}+1}{2n} + \frac{L_{f,x}m^2}{n}\right)\|\mathbb{E}[\mathbf{x}^{r+1} - \mathbf{x}^r]\|^2 + \frac{mL_{f,y}^2}{nL_{f,x}}\sum_{k=1}^m \mathbb{E}\|\mathbf{y}_k^{r+1} - \mathbf{y}_k^r\|^2$$

$$+ \frac{n(b_r + b_{r-1})^2}{2} + \frac{L_{f,x}m^2\sigma_f^2}{n\alpha^2} \tag{120}$$

where we use (99) and (17). $\qquad\square$

**Lemma 8.**

*Under A1-A4, suppose that the sequence is generated by SLAM. Let*

$$\mathcal{Q}_{v,k}^r \triangleq \frac{\rho}{2\sqrt{\gamma}}\|\mathbf{A}\mathbb{E}[\mathbf{y}_k^r]\|^2 + \frac{1}{2}\|\mathbb{E}[\mathbf{y}_k^r - \mathbf{y}_k^{r-1}]\|_{\mathbf{D}}^2 + \frac{L_{g,1}}{2n}\mathbb{E}\|\mathbf{x}^r - \mathbf{x}^{r-1}\|^2 + \frac{L_{g,1}}{2n}\mathbb{E}\|\mathbf{y}_k^r - \mathbf{y}_k^{r-1}\|^2 \geq 0, \tag{121}$$

*then, the following is true,*

$$\mathcal{Q}_{v,k}^{r+1} - \mathcal{Q}_{v,k}^r \leq \frac{L_{g,1}}{2n}\|\mathbf{x}^{r+1} - \mathbf{x}^r\|^2 + \frac{3L_{g,1}}{2n}\mathbb{E}\|\mathbf{y}_k^{r+1} - \mathbf{y}_k^r\|^2 - \frac{1}{2}\|\mathbb{E}[\mathbf{v}_k^{r+1}]\|_{\mathbf{D}}^2, \forall k. \tag{122}$$

*Proof.* Following steps from (112) to (117), we can similarly obtain the following series of inequalities for sequence $\{\mathbf{y}_k^r, \forall k\}$.

$$\frac{\rho}{2\sqrt{\gamma}}\|\mathbf{A}\mathbb{E}[\mathbf{y}_k^{r+1}]\|^2 + \frac{1}{2}\|\mathbb{E}[\mathbf{y}_k^{r+1} - \mathbf{y}_k^r]\|_{\mathbf{D}}^2$$

$$\leq \frac{\rho}{2\sqrt{\gamma}}\|\mathbf{A}\mathbb{E}[\mathbf{y}_k^r]\|^2 + \frac{1}{2}\|\mathbb{E}[\mathbf{y}_k^r - \mathbf{y}_k^{r-1}]\|_{\mathbf{D}}^2 - \frac{1}{2}\|\mathbb{E}[\mathbf{v}_k^{r+1}]\|_{\mathbf{D}}^2$$

$$+ \langle \nabla g_k(\mathbf{x}^{r-1}, \mathbf{y}_k^{r-1}) - \nabla g_k(\mathbf{x}^r, \mathbf{y}_k^r), \mathbb{E}[\mathbf{y}_k^{r+1} - \mathbf{y}_k^r] \rangle$$

$$\leq \frac{\rho}{2\sqrt{\gamma}}\|\mathbf{A}\mathbb{E}[\mathbf{y}_k^r]\|^2 + \frac{1}{2}\|\mathbb{E}[\mathbf{y}_k^r - \mathbf{y}_k^{r-1}]\|_{\mathbf{D}}^2 - \frac{1}{2}\|\mathbb{E}[\mathbf{v}_k^{r+1}]\|_{\mathbf{D}}^2$$

$$+ \langle \nabla g_k(\mathbf{x}^{r-1}, \mathbf{y}_k^{r-1}) - \nabla g_k(\mathbf{x}^r, \mathbf{y}_k^{r-1}) + \nabla g_k(\mathbf{x}^r, \mathbf{y}_k^{r-1}) - \nabla g_k(\mathbf{x}^r, \mathbf{y}_k^r), \mathbb{E}[\mathbf{y}_k^{r+1} - \mathbf{y}_k^r] \rangle \tag{123}$$

$$\overset{(a)}{\leq} \frac{\rho}{2\sqrt{\gamma}}\|\mathbf{A}\mathbb{E}[\mathbf{y}_k^r]\|^2 + \frac{1}{2}\|\mathbb{E}[\mathbf{y}_k^r - \mathbf{y}_k^{r-1}]\|_{\mathbf{D}}^2 - \frac{1}{2}\|\mathbb{E}[\mathbf{v}_k^{r+1}]\|_{\mathbf{D}}^2$$

$$+ \frac{L_{g,1}}{2n}\mathbb{E}\|\mathbf{x}^r - \mathbf{x}^{r-1}\|^2 + \frac{L_{g,1}}{n}\|\mathbb{E}\mathbf{y}_k^{r+1} - \mathbf{y}_k^r\|^2 + \frac{L_{g,1}}{2n}\mathbb{E}\|\mathbf{y}_k^r - \mathbf{y}_k^{r-1}\|^2 \tag{124}$$

where in $(a)$ we use Young's inequality

$$\langle \nabla g_k(\mathbf{x}^{r-1}, \mathbf{y}_k^{r-1}) - \nabla g_k(\mathbf{x}^r, \mathbf{y}_k^{r-1}), \mathbb{E}[\mathbf{y}_k^{r+1} - \mathbf{y}_k^r] \rangle \leq \frac{L_{g,1}}{2n}\mathbb{E}\|\mathbf{x}^r - \mathbf{x}^{r-1}\|^2 + \frac{L_{g,1}}{2n}\|\mathbb{E}\mathbf{y}_k^{r+1} - \mathbf{y}_k^r\|^2, \tag{125}$$

and gradient Lipschitz $g(\cdot, \mathbf{y})$

$$\langle \nabla g_k(\mathbf{x}^r, \mathbf{y}_k^{r-1}) - \nabla g_k(\mathbf{x}^r, \mathbf{y}_k^r), \mathbb{E}[\mathbf{y}_k^{r+1} - \mathbf{y}_k^r] \rangle \leq \frac{L_{g,1}}{2n}\mathbb{E}\|\mathbf{y}_k^r - \mathbf{y}_k^{r-1}\|^2 + \frac{L_{g,1}}{2n}\|\mathbb{E}\mathbf{y}_k^{r+1} - \mathbf{y}_k^r\|^2.$$

$\square$

## B.5 Proof of Theorem 1

*Proof.* In this subsection, we will show the convergence rate regarding the stationarity, optimality, and consensus violation w.r.t. both UL and LL optimization variables generated by SLAM as follows.

### B.5.1 Consensus Violation of the UL Variables

Using the fact $\mathbb{E}[X^2] = \mathbb{E}[X]^2 + \text{Var}[X]$ (17) gives

$$\mathbb{E}\left[\|\boldsymbol{\lambda}^{r+1} - \boldsymbol{\lambda}^r\|^2|\mathcal{F}'^r\right] - \|\mathbb{E}[\boldsymbol{\lambda}^{r+1} - \boldsymbol{\lambda}^r]|\mathcal{F}'^r\|^2$$

$$= \mathbb{E}\left[\|\boldsymbol{\lambda}^{r+1} - \boldsymbol{\lambda}^r - \mathbb{E}[\boldsymbol{\lambda}^{r+1} - \boldsymbol{\lambda}^r]|\mathcal{F}'^r\|^2|\mathcal{F}'^r\right] = \frac{\rho^2}{\gamma^2}\mathbb{E}\left[\|\mathbf{A}\mathbf{x}^{r+1} - \mathbb{E}\mathbf{A}\mathbf{x}^{r+1}\|^2|\mathcal{F}'^r\right] \tag{126}$$

$$\leq \frac{\rho^2}{\gamma^2\alpha^2}\sigma_{\max}(\mathbf{A}^T\mathbf{A})\mathbb{E}\|\mathbf{h}_f^r - \mathbb{E}\mathbf{h}_f^r\|^2 \leq \frac{\rho^2\sigma_{\max}(\mathbf{A}^T\mathbf{A})\sigma_f^2}{n\gamma^2\alpha^2}. \tag{127}$$

From (101a), we have

$$\mathbb{E}\|\mathbf{A}^T\mathbf{x}^{r+1}\|^2$$

$$= \frac{\gamma^2}{\rho^2}\|\mathbb{E}\boldsymbol{\lambda}^{r+1} - \boldsymbol{\lambda}^r\|^2 + \frac{\sigma_{\max}(\mathbf{A}^T\mathbf{A})\sigma_f^2}{n\alpha^2} \tag{128}$$

$$\overset{(101a)}{\leq} \frac{\gamma}{\rho}\frac{4\left\|\mathbb{E}\mathbf{w}^{r+1}\right\|_{\mathbf{D}^T\mathbf{D}}^2}{\rho\gamma\widetilde{\sigma}_{\min}(\mathbf{A}^T\mathbf{A})} + \frac{4(m+1)}{n^2\rho^2\widetilde{\sigma}_{\min}(\mathbf{A}^T\mathbf{A})}\left(\frac{(m+1)D_xL_{f,x}^2}{\alpha^2} + L_{f,y}^2\sum_{k=1}^m\mathbb{E}\left\|\mathbf{y}_k^{r+1} - \mathbf{y}_k^r\right\|^2\right)$$

$$+ \frac{4(b_r + b_{r-1})^2}{\rho^2\widetilde{\sigma}_{\min}(\mathbf{A}^T\mathbf{A})} + \frac{\sigma_{\max}(\mathbf{A}^T\mathbf{A})\sigma_f^2}{n\alpha^2}. \tag{129}$$

Considering the recursion derived in (110), we can obtain that there is a constant $C$ such that

$$\mathbb{E}\|\mathbf{A}^T\mathbf{x}^{r+1}\|^2 \leq \frac{\gamma}{\rho}\left(C\mathcal{Q}_w^r - C\mathcal{Q}_w^{r+1} - \mathbb{E}[\mathbf{w}^{r+1}]^T\left(\frac{C\mathbf{D}}{2} - \frac{4\mathbf{D}^T\mathbf{D}}{\rho\gamma\widetilde{\sigma}_{\min}(\mathbf{A}^T\mathbf{A})}\right)\mathbb{E}[\mathbf{w}^{r+1}]\right)$$

$$+ \left(\frac{\gamma}{\rho}\left(\frac{L_{f,1}+1}{2n} + \frac{L_{f,x}m^2}{n}\right) + \frac{4L_{f,x}^2m^2}{n^2\rho^2\widetilde{\sigma}_{\min}(\mathbf{A}^T\mathbf{A})}\right)\frac{D_x}{\alpha^2}$$

$$+ \frac{mL_{f,y}^2}{n}\left(\frac{\gamma}{\rho}\frac{1}{L_{f,x}} + \frac{4}{n\rho^2\widetilde{\sigma}_{\min}(\mathbf{A}^T\mathbf{A})}\right)\sum_{k=1}^m\mathbb{E}\left\|\mathbf{y}_k^{r+1} - \mathbf{y}_k^r\right\|^2$$

$$+ \frac{\gamma}{\rho}\left(\frac{n(b_r + b_{r-1})^2}{2} + \frac{L_{f,x}m^2\sigma_f^2}{n^2\alpha^2}\right) + \frac{4(b_r + b_{r-1})^2}{\rho^2\widetilde{\sigma}_{\min}(\mathbf{A}^T\mathbf{A})} + \frac{\sigma_{\max}(\mathbf{A}^T\mathbf{A})\sigma_f^2}{n\alpha^2} \quad (130)$$

To show the descent of the right-hand side of the above inequality, we need

$$\frac{C\mathbf{D}}{2} - \frac{4\mathbf{D}^T\mathbf{D}}{\rho\gamma\widetilde{\sigma}_{\min}(\mathbf{A}^T\mathbf{A})} \succ 0, \quad (131)$$

so, it is sufficient to show

$$\frac{C}{2}\left(\alpha\mathbf{I} - \rho\gamma\mathbf{A}^T\mathbf{A}\right) - \frac{4}{\rho\gamma\widetilde{\sigma}_{\min}(\mathbf{A}^T\mathbf{A})}\left(\alpha\mathbf{I} - \rho\gamma\mathbf{A}^T\mathbf{A}\right)^T\left(\alpha\mathbf{I} - \rho\gamma\mathbf{A}^T\mathbf{A}\right) \succ 0 \quad (132)$$

$$\Rightarrow \frac{C}{2}\left(\alpha\mathbf{I} - \rho\gamma\mathbf{A}^T\mathbf{A}\right) - \frac{4\alpha^2\mathbf{I}}{\rho\gamma\widetilde{\sigma}_{\min}(\mathbf{A}^T\mathbf{A})} + \frac{8\alpha\mathbf{A}^T\mathbf{A}}{\widetilde{\sigma}_{\min}(\mathbf{A}^T\mathbf{A})} - \frac{4\rho\gamma(\mathbf{A}^T\mathbf{A})^2}{\widetilde{\sigma}_{\min}(\mathbf{A}^T\mathbf{A})} \succ 0 \quad (133)$$

$$\Rightarrow \frac{C}{2}\alpha\mathbf{I} - \frac{4\alpha^2\mathbf{I}}{\rho\gamma\widetilde{\sigma}_{\min}(\mathbf{A}^T\mathbf{A})} - \frac{4\rho\gamma(\mathbf{A}^T\mathbf{A})^2}{\widetilde{\sigma}_{\min}(\mathbf{A}^T\mathbf{A})} \succ 0 \quad \text{and} \quad \frac{8\alpha\mathbf{A}^T\mathbf{A}}{\widetilde{\sigma}_{\min}(\mathbf{A}^T\mathbf{A})} - \frac{C}{2}\rho\mathbf{A}^T\mathbf{A} \succ 0 \quad (134)$$

$$\Rightarrow \frac{C}{2}\alpha\mathbf{I} - \frac{4\alpha^2\mathbf{I}}{\rho\gamma\widetilde{\sigma}_{\min}(\mathbf{A}^T\mathbf{A})} - \frac{4\rho\gamma(\mathbf{A}^T\mathbf{A})^2}{\widetilde{\sigma}_{\min}(\mathbf{A}^T\mathbf{A})} \succ 0 \quad \text{and} \quad \frac{8\alpha\mathbf{A}^T\mathbf{A}}{\widetilde{\sigma}_{\min}(\mathbf{A}^T\mathbf{A})} - \frac{C}{2}\rho\mathbf{A}^T\mathbf{A} \succ 0 \quad (135)$$

$$\Rightarrow \frac{C}{2}\alpha - \frac{4\alpha}{\widetilde{\sigma}_{\min}(\mathbf{A}^T\mathbf{A})}\left(\frac{\alpha}{\rho\gamma} + \frac{\rho\gamma\sigma_{\max}^2(\mathbf{A}^T\mathbf{A})}{\alpha}\right) \geq 0 \quad \text{and} \quad \alpha \geq \frac{C\widetilde{\sigma}_{\min}(\mathbf{A}^T\mathbf{A})}{16}, \quad (136)$$

which means

$$\frac{\tau_f}{\rho} + \frac{\rho\sigma_{\max}^2(\mathbf{A}^T\mathbf{A})}{\tau_f} \leq \frac{\widetilde{\sigma}_{\min}(\mathbf{A}^T\mathbf{A})C}{8} \quad \text{and} \quad \alpha \geq \frac{C\widetilde{\sigma}_{\min}(\mathbf{A}^T\mathbf{A})}{16}. \quad (137)$$

Applying (131) and the telescoping sum, we have

$$\frac{1}{T}\sum_{r=1}^T\mathbb{E}\|\mathbf{A}^T\mathbf{x}^{r+1}\|^2 \leq \frac{C\gamma\mathcal{Q}_w^1}{T\rho} + \left(\frac{\gamma}{\rho}\left(\frac{L_{f,x}+1}{2n} + \frac{L_{f,x}m^2}{n}\right) + \frac{4L_{f,x}^2m^2}{n^2\rho^2\widetilde{\sigma}_{\min}(\mathbf{A}^T\mathbf{A})}\right)\frac{D_x}{\alpha^2}$$

$$+ \left(\frac{\gamma}{\rho}\frac{mL_{f,y}^2}{nL_{f,x}} + \frac{4L_{f,y}^2m}{n^2\rho^2\widetilde{\sigma}_{\min}(\mathbf{A}^T\mathbf{A})}\right)\frac{D_y}{\beta^2}$$

$$+ \frac{\gamma}{\rho}\left(\frac{n(b_r + b_{r-1})^2}{2} + \frac{L_{f,x}m^2\sigma_f^2}{n\alpha^2}\right) + \frac{4(b_r + b_{r-1})^2}{\rho^2\widetilde{\sigma}_{\min}(\mathbf{A}^T\mathbf{A})} + \frac{\sigma_{\max}(\mathbf{A}^T\mathbf{A})\sigma_f^2}{n\alpha^2}$$

$$\sim \mathcal{O}\left(\frac{1}{\sqrt{nT}}\right) \quad (138)$$

where we choose $\alpha = \alpha_0\sqrt{T/n}, \beta = \beta_0\sqrt{T/n}$.

### B.5.2 Consensus Violation of the LL Optimization Variables

From Lemma 8, we know that $\mathcal{Q}_{v,k}^r$ is lower bounded by 0. According to (122), we have

$$\mathcal{Q}_{v,k}^{r+1} - \mathcal{Q}_{v,k}^r \leq \frac{L_{g,1}}{n}\left(\frac{D_x}{2\alpha^2} + \frac{3}{2}\mathbb{E}\|\mathbf{y}_k^{r+1} - \mathbf{y}_k^r\|^2\right) - \frac{1}{2}\|\mathbb{E}[\mathbf{v}_k^{r+1}]\|_{\mathbf{D}}^2. \quad (139)$$

Multiplying (139) by constant $C$ and adding (101b) together, we have

$$\frac{\gamma}{\rho}\|\mathbb{E}\boldsymbol{\omega}_k^{r+1} - \boldsymbol{\omega}_k^r\|^2 \leq C\mathcal{Q}_{v,k}^r - C\mathcal{Q}_{v,k}^{r+1} - \mathbb{E}[\mathbf{v}_k^{r+1}]^T\left(\frac{C\mathbf{D}}{2} - \frac{3\mathbf{D}^T\mathbf{D}}{\rho\gamma\widetilde{\sigma}_{\min}(\mathbf{A}^T\mathbf{A})}\right)\mathbb{E}[\mathbf{v}_k^{r+1}]$$

$$+ \frac{CL_{g,1}}{n}\left(\frac{D_x}{2\alpha^2} + \frac{3}{2}\mathbb{E}\|\mathbf{y}_k^{r+1} - \mathbf{y}_k^r\|^2\right) + \frac{3L_{g,1}^2}{n^2\rho\gamma\widetilde{\sigma}_{\min}(\mathbf{A}^T\mathbf{A})}\left(\frac{D_x}{\alpha^2} + \mathbb{E}\left\|\mathbf{y}_k^{r+1} - \mathbf{y}_k^r\right\|^2\right). \quad (140)$$

Note that

$$\mathbb{E}\left[\|\boldsymbol{\omega}_k^{r+1} - \boldsymbol{\omega}_k^r\|^2 | \mathcal{F}'^r\right] - \|\mathbb{E}[\boldsymbol{\omega}_k^{r+1} - \boldsymbol{\omega}_k^r]|\mathcal{F}'^r\|^2$$

$$=\mathbb{E}\left[\|\boldsymbol{\omega}_k^{r+1} - \boldsymbol{\omega}_k^r - \mathbb{E}[\boldsymbol{\omega}_k^{r+1} - \boldsymbol{\omega}_k^r]|\mathcal{F}'^r\|^2|\mathcal{F}'^r\right] = \frac{\rho^2}{\gamma^2\beta^2}\mathbb{E}\left[\|\mathbf{A}\mathbf{y}_k^{r+1} - \mathbb{E}\mathbf{A}\mathbf{y}_k^{r+1}\|^2|\mathcal{F}'^r\right] \tag{141}$$

$$\leq \frac{\rho^2}{\gamma^2\beta^2}\sigma_{\max}(\mathbf{A}^T\mathbf{A})\|\mathbf{h}_{g,k}^r - \mathbb{E}\mathbf{h}_{g,k}^r\|^2 \leq \frac{\rho^2\sigma_{\max}(\mathbf{A}^T\mathbf{A})\sigma_g^2}{n\gamma^2\beta^2}. \tag{142}$$

From (4b), we have

$$\mathbb{E}\|\mathbf{A}\mathbf{y}_k^{r+1}\|^2$$

$$\overset{(4b)}{\leq} \frac{\gamma^2}{\rho^2}\mathbb{E}\|\boldsymbol{\omega}_k^{r+1} - \boldsymbol{\omega}_k^r\|^2 \tag{143}$$

$$\overset{(142)}{\leq} \frac{\gamma^2}{\rho^2}\|\mathbb{E}\boldsymbol{\omega}_k^{r+1} - \boldsymbol{\omega}_k^r\|^2 + \frac{\sigma_{\max}(\mathbf{A}^T\mathbf{A})\sigma_g^2}{n\beta^2} \tag{144}$$

$$\overset{(140)}{\leq} \frac{\gamma}{\rho}\left(C\mathcal{Q}_{v,k}^r - C\mathcal{Q}_{v,k}^{r+1} - \mathbb{E}[\mathbf{v}_k^{r+1}]^T\left(\frac{C\mathbf{D}}{2} - \frac{3\mathbf{D}^T\mathbf{D}}{\rho\gamma\widetilde{\sigma}_{\min}(\mathbf{A}^T\mathbf{A})}\right)\mathbb{E}[\mathbf{v}_k^{r+1}]\right) + \frac{\sigma_{\max}(\mathbf{A}^T\mathbf{A})\sigma_g^2}{n\beta^2}$$

$$+ \frac{\gamma}{\rho n}\left(\frac{CL_{g,1}}{2} + \frac{3L_{g,1}^2}{n\rho\gamma\widetilde{\sigma}_{\min}(\mathbf{A}^T\mathbf{A})}\right)\frac{D_x}{\alpha^2} + \frac{\gamma}{\rho}\left(\frac{3CL_{g,1}}{2n} + \frac{3L_{g,1}^2}{n^2\rho\gamma\widetilde{\sigma}_{\min}(\mathbf{A}^T\mathbf{A})}\right)\mathbb{E}\left\|\mathbf{y}_k^{r+1} - \mathbf{y}_k^r\right\|^2$$

$$\overset{(131),(137)}{\leq} \frac{\gamma}{\rho}\left(C\mathcal{Q}_{v,k}^r - C\mathcal{Q}_{v,k}^{r+1}\right) + \frac{\gamma}{\rho}\left(\frac{3CL_{g,1}}{2n} + \frac{3L_{g,1}^2}{n^2\rho\gamma\widetilde{\sigma}_{\min}(\mathbf{A}^T\mathbf{A})}\right)\mathbb{E}\left\|\mathbf{y}_k^{r+1} - \mathbf{y}_k^r\right\|^2 + N_1 \tag{145}$$

where

$$N_1 \triangleq \frac{C\gamma L_{g,1}D_x}{2n\rho\alpha^2} + \frac{\sigma_{\max}(\mathbf{A}^T\mathbf{A})\sigma_g^2}{n\beta^2} + \frac{3L_{g,1}^2 D_x}{n^2\rho^2\alpha^2\widetilde{\sigma}_{\min}(\mathbf{A}^T\mathbf{A})}. \tag{146}$$

Applying telescoping sum, we can get

$$\frac{1}{T}\sum_{r=1}^T\mathbb{E}\|\mathbf{A}\mathbf{y}_k^{r+1}\|^2$$

$$\leq \frac{C\gamma\mathcal{Q}_{v,k}^1}{T\rho} + N_1 + \frac{\gamma}{\rho}\left(\frac{3CL_{g,1}}{2n} + \frac{3L_{g,1}^2}{n^2\rho\gamma\widetilde{\sigma}_{\min}(\mathbf{A}^T\mathbf{A})}\right)\frac{1}{T}\sum_{r=1}^T\mathbb{E}\left\|\mathbf{y}_k^{r+1} - \mathbf{y}_k^r\right\|^2 \tag{147}$$

$$\leq \frac{C\gamma\mathcal{Q}_{v,k}^1}{T\rho} + N_1 + \frac{\gamma}{\rho}\left(\frac{3CL_{g,1}}{2n} + \frac{3L_{g,1}^2}{n^2\rho\gamma\widetilde{\sigma}_{\min}(\mathbf{A}^T\mathbf{A})}\right)\frac{D_y}{\beta^2} \tag{148}$$

$$\sim \mathcal{O}\left(\frac{1}{\sqrt{nT}}\right), \tag{149}$$

when $\gamma \sim \beta \sim \alpha \sim \mathcal{O}(\sqrt{T/n})$.

### B.5.3 Stationarity of the UL Optimization Variable

From (5a) and (5b), we have

$$\bar{\mathbf{x}}^{r+1} = \bar{\mathbf{x}}^r - \frac{1}{\alpha}\bar{\mathbf{h}}_f^r, \quad \bar{\mathbf{y}}_k^{r+1} = \bar{\mathbf{y}}_k^r - \frac{1}{\beta}\bar{\mathbf{h}}_{g,k}^r. \tag{150}$$

Note that $\mathbf{y}_k^*(\mathbb{1}\bar{\mathbf{x}}^r) = \mathbb{1}\bar{\mathbf{y}}_k^*(\mathbb{1}\bar{\mathbf{x}}^r)$ due to (2c). From the notations shown in (2), we know that

$$f(\mathbb{1}\bar{\mathbf{x}}, \mathbb{1}\bar{\mathbf{y}}_k^*(\mathbb{1}\bar{\mathbf{x}})) \triangleq \frac{1}{n}\sum_{i=1}^n f_i(\bar{\mathbf{x}}, \mathbf{y}_{i,k}^*(\bar{\mathbf{x}})), \tag{151a}$$

$$g_k(\mathbb{1}\bar{\mathbf{x}}, \mathbb{1}\bar{\mathbf{y}}_k) \triangleq \frac{1}{n}\sum_{i=1}^n g_{i,k}(\bar{\mathbf{x}}, \bar{\mathbf{y}}_k), \quad \forall k \in [m]. \tag{151b}$$

To simply the notations, we ignore $\mathbb{1}$. For example, we just use $f(\bar{\mathbf{x}}, \bar{\mathbf{y}}_k^*(\bar{\mathbf{x}}))$ and $g_k(\bar{\mathbf{x}}, \bar{\mathbf{y}}_k)$ as the abbreviations of $f(\mathbb{1}\bar{\mathbf{x}}, \mathbb{1}\bar{\mathbf{y}}_k^*(\mathbb{1}\bar{\mathbf{x}}))$ and $g_k(\mathbb{1}\bar{\mathbf{x}}, \mathbb{1}\bar{\mathbf{y}}_k)$ in the following derivation in this proof. Similarly, we have

$$\overline{\nabla}f(\mathbf{x}, \mathbf{y}_k) \triangleq \frac{1}{n}\sum_{i=1}^n \overline{\nabla}f_i(\mathbf{x}_i, \mathbf{y}_{i,k}). \tag{152}$$

Then, we can have the descent at the consensus space as follows

$$
\mathbb{E}_{\xi^r}\left[f(\bar{\mathbf{x}}^{r+1}, \bar{\mathbf{y}}_k^*(\bar{\mathbf{x}}^{r+1}))|\mathcal{F}'^r\right]
$$

$$
\overset{(a)}{\leq} f(\bar{\mathbf{x}}^r, \bar{\mathbf{y}}_k^*(\bar{\mathbf{x}}^r)) + \mathbb{E}\left[\langle\nabla f(\bar{\mathbf{x}}^r, \bar{\mathbf{y}}_k^*(\bar{\mathbf{x}}^r)), \bar{\mathbf{x}}^{r+1} - \bar{\mathbf{x}}^r\rangle |\mathcal{F}'^r\right] + \mathbb{E}\left[\frac{L_f}{2}\|\bar{\mathbf{x}}^{r+1} - \bar{\mathbf{x}}^r\|^2|\mathcal{F}'^r\right] \tag{153}
$$

$$
= f(\bar{\mathbf{x}}^r, \bar{\mathbf{y}}_k^*(\bar{\mathbf{x}}^r)) + \langle\nabla f(\bar{\mathbf{x}}^r, \bar{\mathbf{y}}_k^*(\bar{\mathbf{x}}^r)), \mathbb{E}[\bar{\mathbf{x}}^{r+1} - \bar{\mathbf{x}}^r]|\mathcal{F}'^r\rangle
$$
$$
+ \frac{L_f}{2}\|\mathbb{E}[\bar{\mathbf{x}}^{r+1} - \bar{\mathbf{x}}^r]\|^2 + \frac{L_f}{2}\mathbb{E}\left[\|\bar{\mathbf{x}}^{r+1} - \bar{\mathbf{x}}^r - \mathbb{E}[\bar{\mathbf{x}}^{r+1} - \bar{\mathbf{x}}^r]\|^2|\mathcal{F}'^r\right] \tag{154}
$$

$$
\overset{(b)}{\leq} f(\bar{\mathbf{x}}^r, \bar{\mathbf{y}}_k^*(\bar{\mathbf{x}}^r)) - \frac{1}{2\alpha}\|\nabla f(\bar{\mathbf{x}}^r, \bar{\mathbf{y}}_k^*(\bar{\mathbf{x}}^r))\|^2 - \left(\frac{\alpha}{2} - \frac{L_f}{2}\right)\|\mathbb{E}[\bar{\mathbf{x}}^{r+1} - \bar{\mathbf{x}}^r]\|^2
$$
$$
+ \frac{1}{2\alpha}\|\nabla f(\bar{\mathbf{x}}^r, \mathbf{y}_k^*(\bar{\mathbf{x}}^r)) - \bar{\mathbf{h}}_f^r\|^2 + \frac{L_f}{2\alpha^2}\mathbb{E}\left[\|\bar{\mathbf{h}}_f^r - \mathbb{E}\bar{\mathbf{h}}_f^r\|^2\mathcal{F}'^r\right] \tag{155}
$$

$$
\overset{(c)}{\leq} f(\bar{\mathbf{x}}^r, \bar{\mathbf{y}}_k^*(\bar{\mathbf{x}}^r)) - \frac{1}{2\alpha}\|\nabla f(\bar{\mathbf{x}}^r, \bar{\mathbf{y}}_k^*(\bar{\mathbf{x}}^r))\|^2 - \left(\frac{\alpha}{2} - \frac{L_f}{2}\right)\|\mathbb{E}[\bar{\mathbf{x}}^{r+1} - \bar{\mathbf{x}}^r]\|^2
$$
$$
+ \left(\frac{2(m+1)L_f^2}{\alpha n^2} + \frac{10(m+1)}{\alpha n^2}\left(\frac{C_{xy}L_{f,0}L_{g,2}}{\mu_g^2}\right)^2 + \frac{8(m+1)L_{y,c}^2 L_y^2}{\alpha n}\right)\|\mathbf{x}^r - \mathbb{1}\bar{\mathbf{x}}^r\|^2
$$
$$
+ \sum_{k=1}^m \frac{8(m+1)L_{y,c}^2}{n\alpha}\|\bar{\mathbf{y}}_k^r - \bar{\mathbf{y}}_k^*(\bar{\mathbf{x}}^r)\|^2 + \left(\frac{4(mL_{f,y}^2 + (m+1)L_{y,c}^2)}{\alpha n^2}\right)\|\mathbf{y}_k^r - \mathbb{1}\bar{\mathbf{y}}_k^r\|^2
$$
$$
+ \frac{2mL_{f,y}^2}{n^2\alpha}\sum_{k=1}^m \mathbb{E}\|\mathbf{y}_k^{r+1} - \mathbf{y}_k^r\|^2 + \frac{2b_r^2}{n\alpha} + \frac{L_f\sigma_f^2}{2n\alpha^2} \tag{156}
$$

where $(a)$ follows the gradient Lipschitz continuity of the UL loss function with constant $L_f$ at the consensus space (which is the same as the centralized case, e.g., [31, Lemma 2.2.]), $(b)$ is true because

$$
\langle\nabla f(\bar{\mathbf{x}}^r, \bar{\mathbf{y}}_k^*(\bar{\mathbf{x}}^r)), \mathbb{E}[\bar{\mathbf{x}}^{r+1} - \bar{\mathbf{x}}^r]|\mathcal{F}'^r\rangle
$$
$$
= -\frac{1}{\alpha}\langle\nabla f(\bar{\mathbf{x}}^r, \bar{\mathbf{y}}_k^*(\bar{\mathbf{x}}^r)), \mathbb{E}\bar{\mathbf{h}}_f^r\rangle
$$
$$
= -\frac{1}{2\alpha}\|\nabla f(\bar{\mathbf{x}}^r, \bar{\mathbf{y}}_k^*(\bar{\mathbf{x}}^r))\|^2 - \frac{1}{2\alpha}\|\mathbb{E}\bar{\mathbf{h}}_f^r\|^2 + \frac{1}{2\alpha}\|\nabla f(\bar{\mathbf{x}}^r, \bar{\mathbf{y}}_k^*(\bar{\mathbf{x}}^r)) - \mathbb{E}\bar{\mathbf{h}}_f^r\|^2,
$$

and in $(c)$ we use the following steps: *1)* the difference between the UL gradient and its distributed stochastic estimate can be quantified by

$$
\|\nabla f(\bar{\mathbf{x}}^r, \bar{\mathbf{y}}_k^*(\bar{\mathbf{x}}^r)) - \mathbb{E}\bar{\mathbf{h}}_f^r\|^2
$$
$$
\overset{(a.1)}{\leq} 4\|\nabla f(\bar{\mathbf{x}}^r, \mathbf{y}_k^*(\bar{\mathbf{x}}^r)) - \nabla f(\mathbf{x}^r, \mathbf{y}_k^*(\mathbf{x}^r))\|^2 + 4\|\nabla f(\mathbf{x}^r, \mathbf{y}_k^*(\mathbf{x}^r)) - \overline{\nabla}f(\mathbf{x}^r, \mathbf{y}_k^r)\|^2
$$
$$
+ 4\|\overline{\nabla}f(\mathbf{x}^r, \mathbf{y}_k^r) - \overline{\nabla}f(\mathbf{x}^r, \mathbf{y}_k^{r+1})\|^2 + 4\|\overline{\nabla}f(\mathbf{x}^r, \mathbf{y}_k^{r+1}) - \mathbb{E}\bar{\mathbf{h}}_f^r\|^2
$$
$$
\leq \left(\frac{4(m+1)L_f^2}{n^2} + \frac{10(m+1)}{n^2}\left(\frac{C_{xy}L_{f,0}L_{g,2}}{\mu_g^2}\right)^2\right)\|\mathbf{x}^r - \mathbb{1}\bar{\mathbf{x}}^r\|^2
$$
$$
+ \sum_{k=1}^m \frac{4(m+1)L_{y,c}^2}{n^2}\|\mathbf{y}_k^*(\mathbf{x}^r) - \mathbf{y}_k^r\|^2 + \frac{4mL_{f,y}^2}{n^2}\|\mathbf{y}_k^r - \mathbb{1}\bar{\mathbf{y}}_k^r\|^2
$$
$$
+ \frac{4mL_{f,y}^2}{n^2}\sum_{k=1}^m \mathbb{E}\|\mathbf{y}_k^{r+1} - \mathbf{y}_k^r\|^2 + \frac{4b_r^2}{n}
$$
$$
\overset{(a.2)}{\leq} \left(\frac{4(m+1)L_f^2}{n^2} + \frac{10(m+1)}{n^2}\left(\frac{C_{xy}L_{f,0}L_{g,2}}{\mu_g^2}\right)^2 + \frac{16(m+1)L_{y,c}^2 L_y^2}{n}\right)\|\mathbf{x}^r - \mathbb{1}\bar{\mathbf{x}}^r\|^2
$$
$$
+ \sum_{k=1}^m \frac{16(m+1)L_{y,c}^2}{n}\|\bar{\mathbf{y}}_k^*(\bar{\mathbf{x}}^r) - \bar{\mathbf{y}}_k^r\|^2 + \left(\frac{8(mL_{f,y}^2 + (m+1)L_{y,c}^2)}{n^2}\right)\|\mathbf{y}_k^r - \mathbb{1}\bar{\mathbf{y}}_k^r\|^2
$$
$$
+ \frac{4mL_{f,y}^2}{n^2}\sum_{k=1}^m \mathbb{E}\|\mathbf{y}_k^{r+1} - \mathbf{y}_k^r\|^2 + \frac{4b_r^2}{n}, \tag{157}
$$

and in $(a.1)$ the Lipschitz continuity of $\nabla f_i(\mathbf{x}_i^r, \mathbf{y}_{i,k}^*(\mathbf{x}_i^r))$ w.r.t. $\mathbf{x}_i$ with a constant (defined as $L_f$ here) follows the centralized case directly (which has been shown in [31, Lemma 2.2.]); *2)* constant $L_{y,c}$ is computed as

follows

$$\|\nabla f(\mathbf{x}, \mathbf{y}_k^*(\mathbf{x}^r)) - \overline{\nabla} f(\mathbf{x}, \mathbf{y}_k^r)\|^2$$

$$\leq \frac{m+1}{n^2} \sum_{k=1}^m \left\| \sum_{i=1}^n \nabla_{\mathbf{x}} f_i(\mathbf{x}_i, \mathbf{y}_{i,k}) - \nabla_{\mathbf{x}} f_i(\mathbf{x}_i, \mathbf{y}_{i,k}^*(\mathbf{x})) \right\|^2 + \frac{m+1}{n^2} \left\| \sum_{i=1}^n \Delta_{i,k}^{(1)} + \Delta_{i,k}^{(2)} + \Delta_{i,k}^{(3)} + \Delta_{i,k}^{(4)} + \Delta_{i,k}^{(5)} \right\|^2$$

$$\leq \frac{m+1}{n^2} \sum_{k=1}^m \underbrace{L_{f,1}^2 + 5\left(\frac{L_{f,0}L_{g,2}}{\mu_g}\right)^2 + 5\left(\frac{C_{xy}L_{f,0}L_g}{\mu_g^2}\right)^2 + 5\left(\frac{C_{xy}L_{f,1}}{\mu_g}\right)^2}_{\triangleq L_{y,c}^2} \|\mathbf{y}_k - \mathbf{y}_k^*(\mathbf{x})\|^2$$

$$+ \frac{10(m+1)}{n^2}\left(\frac{C_{xy}L_{f,0}L_{g,2}}{\mu_g^2}\right)^2 \|\mathbf{x}^r - \mathbb{1}\bar{\mathbf{x}}^r\|^2, \tag{158}$$

and terms $\Delta_{i,k}^{(1)}, \Delta_{i,k}^{(2)}, \Delta_{i,k}^{(3)}, \Delta_{i,k}^{(4)}, \Delta_{i,k}^{(5)}$ are defined as

$$\Delta_{i,k}^{(1)} \triangleq \left[ \nabla^2_{\mathbf{x}_i\mathbf{y}_{i,k}} g_{i,k}(\mathbf{x}_i, \mathbf{y}_{i,k}) - \nabla^2_{\mathbf{x}_i\mathbf{y}_{i,k}} g_{i,k}(\mathbf{x}_i, \mathbf{y}_{i,k}^*(\mathbf{x})) \right] \left[ \nabla^2_{\mathbf{y}_{i,k}\mathbf{y}_{i,k}} g_{i,k}(\mathbf{x}, \mathbf{y}_{i,k}) \right]^{-1} \nabla_{\mathbf{y}_{i,k}} f_i(\mathbf{x}_i, \mathbf{y}_{i,k}),$$

$$\Delta_{i,k}^{(2)} \triangleq \nabla^2_{\mathbf{x}_i\mathbf{y}_{i,k}} g_{i,k}(\mathbf{x}_i, \mathbf{y}_{i,k}^*(\mathbf{x})) \left[ \left[ \nabla^2_{\mathbf{y}_{i,k}\mathbf{y}_{i,k}} g_{i,k}(\mathbf{x}_i, \mathbf{y}_{i,k}) \right]^{-1} - \left[ \nabla^2_{\mathbf{y}_k\mathbf{y}_k} g_{i,k}(\bar{\mathbf{x}}, \mathbf{y}_{i,k}) \right]^{-1} \right] \nabla_{\mathbf{y}_{i,k}} f_i(\mathbf{x}_i, \mathbf{y}_{i,k}),$$

$$\Delta_{i,k}^{(3)} \triangleq \nabla^2_{\mathbf{x}_i\mathbf{y}_{i,k}} g_{i,k}(\mathbf{x}_i, \mathbf{y}_{i,k}^*(\mathbf{x})) \left[ \left[ \nabla^2_{\mathbf{y}_{i,k}\mathbf{y}_{i,k}} g_{i,k}(\bar{\mathbf{x}}, \mathbf{y}_{i,k}) \right]^{-1} - \left[ \nabla^2_{\mathbf{y}_k\mathbf{y}_k} g_k(\bar{\mathbf{x}}, \mathbf{y}_k^*(\mathbf{x})) \right]^{-1} \right] \nabla_{\mathbf{y}_{i,k}} f_i(\mathbf{x}_i, \mathbf{y}_{i,k}),$$

$$\Delta_{i,k}^{(4)} \triangleq \nabla^2_{\mathbf{x}_i\mathbf{y}_{i,k}} g_{i,k}(\mathbf{x}_i, \mathbf{y}_{i,k}^*(\mathbf{x})) \left[ \left[ \nabla^2_{\mathbf{y}_k\mathbf{y}_k} g_k(\bar{\mathbf{x}}, \mathbf{y}_k^*(\mathbf{x})) \right]^{-1} - \left[ \nabla^2_{\mathbf{y}_k\mathbf{y}_k} g_k(\mathbf{x}, \mathbf{y}_k^*(\mathbf{x})) \right]^{-1} \right] \nabla_{\mathbf{y}_{i,k}} f_i(\mathbf{x}_i, \mathbf{y}_{i,k}),$$

$$\Delta_{i,k}^{(5)} \triangleq \nabla^2_{\mathbf{x}_i\mathbf{y}_{i,k}} g_{i,k}(\mathbf{x}_i, \mathbf{y}_{i,k}^*(\mathbf{x})) \left[ \nabla^2_{\mathbf{y}_k\mathbf{y}_k} g_k(\mathbf{x}, \mathbf{y}_k^*(\mathbf{x})) \right]^{-1} \left[ \nabla_{\mathbf{y}_{i,k}} f_i(\mathbf{x}_i, \mathbf{y}_{i,k}) - \nabla_{\mathbf{y}_{i,k}} f_i(\mathbf{x}_i, \mathbf{y}_{i,k}^*(\mathbf{x})) \right];$$

*3)* and in $(a.2)$ we apply the triangle inequality and the fact that

$$\|\bar{\mathbf{y}}_k^*(\mathbf{x}^r) - \bar{\mathbf{y}}_k^r\|^2 \leq 2\|\bar{\mathbf{y}}_k^*(\bar{\mathbf{x}}^r) - \bar{\mathbf{y}}_k^r\|^2 + 2\|\bar{\mathbf{y}}_k^*(\mathbf{x}_k) - \bar{\mathbf{y}}_k^*(\bar{\mathbf{x}}_k)\|^2 \tag{159}$$

$$\leq 2\|\bar{\mathbf{y}}_k^*(\bar{\mathbf{x}}^r) - \bar{\mathbf{y}}_k^r\|^2 + 2L_y^2\|\mathbf{x}^r - \mathbb{1}\bar{\mathbf{x}}^r\|^2 \tag{160}$$

based on (25).

Further, from Lemma 5, we know that

$$\|\bar{\mathbf{y}}_k^{r+1} - \bar{\mathbf{y}}_k^*(\bar{\mathbf{x}}^{r+1})\|^2 \leq \left(1 - \frac{\rho_g}{\beta}\right)\left(1 + \frac{2(m+1)nL_y^2}{\alpha} + \frac{\vartheta L_{xy}D_x}{4\alpha^2}\right)\|\bar{\mathbf{y}}_k^r - \bar{\mathbf{y}}_k^*(\bar{\mathbf{x}}^r)\|^2$$

$$+ \left(L_y^2 + \frac{\alpha}{8(m+1)n} + \frac{L_{xy}}{4\vartheta}\right)\|\mathbb{E}\left[\bar{\mathbf{x}}^{r+1} - \bar{\mathbf{x}}^r\right]\|^2$$

$$+ \left(1 + \frac{2(m+1)nL_y^2}{\alpha} + \frac{\vartheta L_{xy}D_x}{4\alpha^2}\right)\frac{\sigma_g^2}{n\beta^2} + \left(\frac{L_{xy}}{4\vartheta} + L_y^2\right)\frac{\sigma_f^2}{n\alpha^2} \tag{161}$$

where we choose $\theta' \triangleq 2(m+1)nL_y^2/\alpha$.

Next, let us define potential function at the consensus space as

$$\mathcal{P}_c^r = \mathbb{E}\left[f(\bar{\mathbf{x}}^r, \bar{\mathbf{y}}_k^*(\bar{\mathbf{x}}^r))|\mathcal{F}^r\right] + \frac{m+1}{n}\sum_{k=1}^m \|\bar{\mathbf{y}}_k^r - \bar{\mathbf{y}}_k^*(\bar{\mathbf{x}}^r)\|^2. \tag{162}$$

Subsequently, we can have

$$\mathcal{P}_c^{r+1} - \mathcal{P}_c^r$$

$$\leq -\frac{1}{2\alpha}\|\nabla f(\bar{\mathbf{x}}^r, \mathbf{y}_k^*(\bar{\mathbf{x}}^r))\|^2 - \underbrace{\left(\frac{\alpha}{2} - \left(\frac{L_f}{2} + \frac{m+1}{n}L_y^2 + \frac{\alpha}{8n^2} + \frac{(m+1)L_{xy}}{4n\vartheta}\right)\right)}_{\triangleq C_1}\|\mathbb{E}[\bar{\mathbf{x}}^{r+1} - \bar{\mathbf{x}}^r]\|^2$$

$$+ \frac{m+1}{n}\left(\underbrace{\frac{8L_{y,c}^2}{\alpha} + \left(1 - \frac{\rho_g}{\beta}\right)\left(1 + \frac{2n(m+1)L_y^2}{\alpha} + \frac{\vartheta L_{xy}D_x}{4\alpha^2}\right) - 1}_{\triangleq C_2}\right)\sum_{k=1}^m \|\bar{\mathbf{y}}_k^r - \bar{\mathbf{y}}_k^*(\bar{\mathbf{x}}^r)\|^2$$

$$+ \left(\frac{2(m+1)L_f^2}{\alpha n^2} + \frac{10(m+1)}{\alpha n^2}\left(\frac{C_{xy}L_{f,0}L_{g,2}}{\mu_g^2}\right)^2 + \frac{8(m+1)L_{y,c}^2 L_y^2}{\alpha n}\right)\|\mathbf{x}^r - \mathbb{1}\bar{\mathbf{x}}^r\|^2$$

$$+ \left(\frac{4(mL_{f,y}^2 + (m+1)L_{y,c}^2)}{\alpha n^2}\right)\sum_{k=1}^m \|\mathbf{y}_k^r - \mathbb{1}\bar{\mathbf{y}}_k^r\|^2 + \frac{2mL_{f,y}^2}{n^2\alpha}\sum_{k=1}^m \mathbb{E}\|\mathbf{y}_k^{r+1} - \mathbf{y}_k^r\|^2$$

$$+ \underbrace{\frac{m+1}{n}\left(\left(1 + \frac{2(m+1)nL_y^2}{\alpha} + \frac{\vartheta L_{xy}D_x}{4\alpha^2}\right)\frac{\sigma_g^2}{n\beta^2} + \left(\frac{L_{xy}}{4\vartheta} + L_y^2\right)\frac{\sigma_f^2}{n\alpha^2}\right) + \frac{4b_r^2}{n\alpha} + \frac{L_f\sigma_f^2}{2n\alpha^2}}_{\triangleq N_2 \sim \mathcal{O}\left(\frac{1}{n\alpha^2}\right)}.$$

1) to show $C_1 > 0$: we need

$$\frac{\alpha}{2} - \left(\frac{L_f}{2} + \frac{m+1}{n}L_y^2 + \frac{\alpha}{8n^2} + \frac{(m+1)L_{xy}}{4n\vartheta}\right) > 0, \tag{163}$$

which requires

$$\alpha > 4\left(\frac{L_f}{2} + \frac{m+1}{n}L_y^2 + \frac{(m+1)L_{xy}}{4n\vartheta}\right). \tag{164}$$

2) to show $C_2 < 0$: we need

$$\frac{2L_{y,c}^2}{\alpha} + \left(1 - \frac{\rho_g}{\beta}\right)\left(1 + \frac{2(m+1)nL_y^2}{\alpha} + \frac{\vartheta L_{xy}D_x}{4\alpha^2}\right) \leq 1, \tag{165}$$

which is

$$\beta < \frac{\rho_g\left(1 + \frac{2(m+1)nL_y^2}{\alpha} + \frac{\vartheta L_{xy}D_x}{4\alpha^2}\right)}{\frac{1}{\alpha}(8L_{y,c}^2 + 2(m+1)L_y^2) + \frac{\vartheta L_{xy}D_x}{4\alpha^2}} = \frac{\rho_g(\alpha^2 + 2(m+1)n\alpha L_y^2 + \vartheta L_{xy}D_x/4)}{2\alpha((m+1)L_y^2 + 4L_{y,c}^2) + \vartheta L_{xy}D_x/4} \tag{166}$$

$$\leq \frac{\rho_g(\alpha + 2(m+1)nL_y^2 + \frac{\vartheta L_{xy}D_x}{4\alpha})}{2(L_y^2 + 4L_{y,c}^2)}. \tag{167}$$

From (164) and (167), we can have constants $C_1 > 0$ and $C_2 < 0$ such that

$$\mathcal{P}_c^{r+1} - \mathcal{P}_c^r \leq -\frac{1}{2\alpha}\|\nabla f(\bar{\mathbf{x}}^r, \mathbf{y}_k^*(\bar{\mathbf{x}}^r))\|^2 - C_1\|\mathbb{E}[\bar{\mathbf{x}}^{r+1} - \bar{\mathbf{x}}^r]\|^2 + \frac{m+1}{n}C_2\sum_{k=1}^m \|\bar{\mathbf{y}}_k^r - \bar{\mathbf{y}}_k^*(\bar{\mathbf{x}}^r)\|^2 + N_2$$

$$+ \left(\frac{2(m+1)L_f^2}{\alpha n^2} + \frac{10(m+1)}{\alpha n^2}\left(\frac{C_{xy}L_{f,0}L_{g,2}}{\mu_g^2}\right)^2 + \frac{8(m+1)L_{y,c}^2 L_y^2}{\alpha n}\right)\|\mathbf{x}^r - \mathbb{1}\bar{\mathbf{x}}^r\|^2$$

$$+ \left(\frac{4(mL_{f,y}^2 + (m+1)L_{y,c}^2)}{\alpha n^2}\right)\sum_{k=1}^m \|\mathbf{y}_k^r - \mathbb{1}\bar{\mathbf{y}}_k^r\|^2 + \frac{2mL_{f,y}^2}{n^2\alpha}\sum_{k=1}^m \mathbb{E}\|\mathbf{y}_k^{r+1} - \mathbf{y}_k^r\|^2. \tag{168}$$

Note that the consensus errors have been quantified in (138) and (149). Let $\widetilde{\mathbf{x}} \triangleq \mathbf{x} - \mathbb{1}\bar{\mathbf{x}}$. Then, we can have

$$\widetilde{\sigma}_{\min}(\mathbf{A}^T\mathbf{A})\|\mathbf{x} - \mathbb{1}\bar{\mathbf{x}}\|^2 \leq \|\mathbf{A}(\mathbf{x} - \mathbb{1}\bar{\mathbf{x}})\|^2 = \widetilde{\mathbf{x}}^T\mathbf{A}^T\mathbf{A}\widetilde{\mathbf{x}} = \sum_{i,j} \|\widetilde{\mathbf{x}}_i - \widetilde{\mathbf{x}}_j\|^2 \tag{169}$$

$$= \sum_{j\in\mathcal{N}_i, \forall i} \|\mathbf{x}_i - \mathbf{x}_j\|^2 = \mathbf{x}^T\mathbf{A}^T\mathbf{A}\mathbf{x} = \|\mathbf{A}\mathbf{x}\|^2, \tag{170}$$

Applying the telescoping sum, we have

$$\frac{1}{T}\sum_{r=1}^{T}\|\nabla f(\bar{\mathbf{x}}^r, \mathbf{y}_k^*(\bar{\mathbf{x}}^r))\|^2$$

$$\leq 2\alpha\frac{\mathcal{P}_c^1 - \mathcal{P}_c}{T} + \frac{4mL_{f,y}^2}{n^2}\sum_{k=1}^{m}\mathbb{E}\|\mathbf{y}_k^{r+1} - \mathbf{y}_k^r\|^2 + \left(\frac{8(mL_{f,y}^2 + (m+1)L_{y,c}^2)}{n^2}\right)\sum_{k=1}^{m}\|\mathbf{y}_k^r - \mathbb{1}\bar{\mathbf{y}}_k^r\|^2$$

$$+ 2\left(\frac{2(m+1)L_f^2}{n^2} + \frac{10(m+1)}{n^2}\left(\frac{C_{xy}L_{f,0}L_{g,2}}{\mu_g^2}\right)^2 + \frac{8(m+1)L_{y,c}^2L_y^2}{n}\right)\|\mathbf{x}^r - \mathbb{1}\bar{\mathbf{x}}^r\|^2 + 2\alpha N_2$$

$$\sim \mathcal{O}\left(\frac{1}{\sqrt{nT}}\right), \tag{171}$$

since $N_2 \sim \mathcal{O}(1/(n\alpha^2))$, where $\underline{\mathcal{P}_c}$ denotes the lower bound of $\mathcal{P}_c$.

### B.5.4 Optimality of the LL Optimization Variables

From (165) and condition $\beta \geq 2(\mu_g + L_{g,1})^{-1}$, we know that $-1 < C_2 < 0$. Combining (80) and (81) with $\theta' = 2(m+1)nL_y^2/\alpha$, so we can have

$$(1 + C_2)\mathbb{E}\|\bar{\mathbf{y}}_k^r - \bar{\mathbf{y}}_k^*(\mathbf{x}^r)\|^2$$

$$\leq \mathbb{E}\|\bar{\mathbf{y}}_k^r - \bar{\mathbf{y}}_k^*(\mathbf{x}^r)\|^2 - \mathbb{E}\|\bar{\mathbf{y}}_k^{r+1} - \bar{\mathbf{y}}_k^*(\mathbf{x}^{r+1})\|^2 + \left(L_y^2 + \frac{\alpha}{8(m+1)n} + \frac{L_{xy}}{4\vartheta}\right)\left\|\mathbb{E}\left[\mathbf{x}^{r+1} - \mathbf{x}^r\right]\right\|^2$$

$$+ \left(1 + \frac{2(m+1)nL_y^2}{\alpha} + \frac{\vartheta L_{xy}D_x}{4\alpha^2}\right)\frac{\sigma_g^2}{n\beta^2} + \left(\frac{L_{xy}}{4\vartheta} + L_y^2\right)\frac{\sigma_f^2}{n\alpha^2}. \tag{172}$$

Applying the telescoping sum on (172), we have

$$\frac{1}{T}\sum_{r=1}^{T}\mathbb{E}\|\bar{\mathbf{y}}_k^r - \bar{\mathbf{y}}_k^*(\mathbf{x}^r)\|^2$$

$$\leq \frac{\|\bar{\mathbf{y}}_k^1 - \bar{\mathbf{y}}_k^*(\mathbf{x}^1)\|^2}{T(1 + C_2)} + \frac{1}{1 + C_2}\left(L_y^2 + \frac{\alpha}{8(m+1)n} + \frac{L_{xy}}{4\vartheta}\right)\frac{1}{T}\sum_{r=1}^{T}\mathbb{E}\left\|\mathbf{x}^{r+1} - \mathbf{x}^r\right\|^2$$

$$+ \frac{1}{1 + C_2}\left(1 + \frac{2(m+1)nL_y^2}{\alpha} + \frac{\vartheta L_{xy}D_x}{4\alpha^2}\right)\frac{\sigma_g^2}{n\beta^2} + \frac{1}{1 + C_2}\left(\frac{L_{xy}}{4\vartheta} + L_y^2\right)\frac{\sigma_f^2}{n\alpha^2} \tag{173}$$

$$\leq \frac{\|\bar{\mathbf{y}}_k^1 - \bar{\mathbf{y}}_k^*(\mathbf{x}^1)\|^2}{T(1 + C_2)} + \frac{1}{1 + C_2}\left(L_y^2 + \frac{\alpha}{8(m+1)n} + \frac{L_{xy}}{4\vartheta}\right)\frac{D_x}{\alpha^2}$$

$$+ \frac{1}{1 + C_2}\left(1 + \frac{2(m+1)nL_y^2}{\alpha} + \frac{\vartheta L_{xy}D_x}{4\alpha^2}\right)\frac{\sigma_g^2}{n\beta^2} + \frac{1}{1 + C_2}\left(\frac{L_{xy}}{4\vartheta} + L_y^2\right)\frac{\sigma_f^2}{n\alpha^2} \sim \mathcal{O}\left(\frac{1}{\sqrt{nT}}\right), \tag{174}$$

since we choose $\alpha \sim \beta \sim \mathcal{O}(\sqrt{T/n})$. $\qquad\square$

### B.6 Proof of Corollary 1

In this case, we have

$$\mathbf{x}^{r+1} = \mathbf{x}^r - \frac{\mathbf{h}_f^r}{\alpha}. \tag{175}$$

Then, Lemma 4 holds because

$$\mathbb{E}\|\mathbf{x}^{r+1} - \mathbf{x}^r\|^2 \leq \frac{2}{\alpha^2}\left(\|\mathbf{h}_f^r - \mathbb{E}\mathbf{h}_f^r\|^2 + \|\mathbb{E}\mathbf{h}_f^r\|^2\right) \leq n\frac{\overbrace{2\sigma_f^2 + 2L_{f,0}}^{\triangleq D_x}}{\alpha^2}. \tag{176}$$

### B.6.1 Consensus Violation of the LL Optimization Variables

From (176), we know that $\mathbb{E}\|\mathbf{x}^{r+1} - \mathbf{x}^r\|^2 \leq nD_x/\alpha^2$. Following (139) to (149), we can still have $\frac{1}{T}\sum_{r=1}^{T}\mathbb{E}\|\mathbf{A}\mathbf{y}_k^r\|^2 \sim \mathcal{O}\left(\frac{1}{\sqrt{nT}}\right)$ when $\rho \geq n$.

### B.6.2 Stationarity of the UL Optimization Variables

Note that there will be $n$ objective functions corresponding to $\{\mathbf{x}_i, \forall n\}$. We will show the convergence of SLAM-L for each one of them. According to the gradient Lipschitz continuity and by following (156), we have

$$\sum_{i=1}^{n} \mathbb{E}_{\xi^r} \left[ f_i(\mathbf{x}_i^{r+1}, \mathbf{y}_{i,k}^*(\mathbf{x}_i^{r+1})) | \mathcal{F}'^r \right]$$

$$\leq \sum_{i=1}^{n} f_i(\mathbf{x}_i^r, \mathbf{y}_{i,k}^*(\mathbf{x}_i^r)) + \mathbb{E}\left[ \langle \nabla f_i(\mathbf{x}_i^r, \mathbf{y}_{i,k}^*(\mathbf{x}_i^r)), \mathbf{x}_i^{r+1} - \mathbf{x}_i^r \rangle | \mathcal{F}'^r \right] + \mathbb{E}\left[ \frac{L_{f,1}}{2} \|\mathbf{x}_i^{r+1} - \mathbf{x}_i^r\|^2 | \mathcal{F}'^r \right]$$

$$= \sum_{i=1}^{n} f_i(\mathbf{x}_i^r, \mathbf{y}_{i,k}^*(\mathbf{x}_i^r)) + \langle \nabla f_i(\mathbf{x}_i^r, \mathbf{y}_{i,k}^*(\mathbf{x}_i^r)), \mathbb{E}[\mathbf{x}_i^{r+1} - \mathbf{x}_i^r] | \mathcal{F}'^r \rangle + \frac{L_{f,1}}{2} \|\mathbb{E}[\mathbf{x}_i^{r+1} - \mathbf{x}_i^r]\|^2$$

$$+ \frac{L_{f,1}}{2} \mathbb{E}\left[ \|\mathbf{x}_i^{r+1} - \mathbf{x}_i^r - \mathbb{E}[\mathbf{x}_i^{r+1} - \mathbf{x}_i^r]\|^2 | \mathcal{F}'^r \right] \tag{177}$$

$$\overset{(a)}{\leq} \sum_{i=1}^{n} f_i(\mathbf{x}_i^r, \mathbf{y}_{i,k}^*(\mathbf{x}_i^r)) - \frac{1}{2\alpha} \|\nabla f_i(\mathbf{x}_i^r, \mathbf{y}_{i,k}^*(\mathbf{x}_i^r))\|^2 - \left( \frac{\alpha}{2} - \frac{L_{f,1}}{2} \right) \|\mathbb{E}[\mathbf{x}_i^{r+1} - \mathbf{x}_i^r]\|^2$$

$$+ \frac{1}{2\alpha} \|\nabla f_i(\mathbf{x}_i^r, \mathbf{y}_{i,k}^*(\mathbf{x}_i^r)) - [\mathbf{h}_f^r]_i\|^2 + \frac{L_{f,1}}{2\alpha^2} \mathbb{E}\left[ \|[\mathbf{h}_f^r]_i - \mathbb{E}[\mathbf{h}_f^r]_i\|^2 \mathcal{F}'^r \right] \tag{178}$$

$$\overset{(b)}{\leq} \sum_{i=1}^{n} f_i(\mathbf{x}_i^r, \mathbf{y}_{i,k}^*(\mathbf{x}_i^r)) - \frac{1}{2\alpha} \|\nabla f_i(\mathbf{x}_i^r, \mathbf{y}_{i,k}^*(\mathbf{x}_i^r))\|^2 - \left( \frac{\alpha}{2} - \frac{L_{f,1}}{2} \right) \|\mathbb{E}[\mathbf{x}^{r+1} - \mathbf{x}^r]\|^2$$

$$+ \frac{2n(m+1)}{\alpha} \sum_{k=1}^{m} L_{f,l}^2 \|\bar{\mathbf{y}}_k^r - \bar{\mathbf{y}}_k^*(\mathbf{x}^r)\|^2 + \frac{2mL_{f,y}^2}{\alpha} \sum_{k=1}^{m} \|\mathbf{y}_k^r - \mathbb{1}\bar{\mathbf{y}}_k^r\|^2$$

$$+ \frac{2mL_{f,y}^2}{\alpha} \sum_{k=1}^{m} \mathbb{E}\|\mathbf{y}_k^{r+1} - \mathbf{y}_k^r\|^2 + \frac{2b_r^2}{\alpha} + \frac{nL_{f,1}\sigma_f^2}{2\alpha^2} \tag{179}$$

where $(a)$ is true because

$$\langle \nabla f_i(\mathbf{x}_i^r, \mathbf{y}_{i,k}^*(\mathbf{x}_i^r)), \mathbb{E}[\mathbf{x}_i^{r+1} - \mathbf{x}_i^r] | \mathcal{F}'^r \rangle = -\frac{1}{\alpha} \langle \nabla f_i(\mathbf{x}_i^r, \mathbf{y}_{i,k}^*(\mathbf{x}_i^r)), \mathbb{E}[\mathbf{h}_f^r]_i \rangle$$

$$= -\frac{1}{2\alpha} \|\nabla f_i(\mathbf{x}_i^r, \mathbf{y}_{i,k}^*(\mathbf{x}_i^r))\|^2 - \frac{1}{2\alpha} \|\mathbb{E}[\mathbf{h}_f^r]_i\|^2 + \frac{1}{2\alpha} \|\nabla f_i(\mathbf{x}_i^r, \mathbf{y}_{i,k}^*(\mathbf{x}_i^r)) - \mathbb{E}[\mathbf{h}_f^r]_i\|^2, \tag{180}$$

and in $(b)$ we apply the following facts,

$$\sum_{i=1}^{n} \|\nabla f_i(\mathbf{x}_i^r, \mathbf{y}_{i,k}^*(\mathbf{x}_i^r)) - \mathbb{E}[\mathbf{h}_f^r]_i\|^2$$

$$\leq \sum_{i=1}^{n} 4\|\nabla f_i(\mathbf{x}_i^r, \mathbf{y}_{i,k}^*(\mathbf{x}_i^r)) - \bar{\nabla} f_i(\mathbf{x}_i^r, \bar{\mathbf{y}}_k^r)\|^2 + 4\|\bar{\nabla} f_i(\mathbf{x}_i^r, \bar{\mathbf{y}}_k^r) - \bar{\nabla} f_i(\mathbf{x}_i^r, \mathbf{y}_{i,k}^r)\|^2$$

$$+ 4\|\bar{\nabla} f_i(\mathbf{x}_i^r, \mathbf{y}_{i,k}^{r+1}) - \bar{\nabla} f_i(\mathbf{x}_i^r, \mathbf{y}_{i,k}^r)\|^2 + 4\|\bar{\nabla} f_i(\mathbf{x}_i^r, \mathbf{y}_{i,k}^{r+1}) - \mathbb{E}[\mathbf{h}_f^r]_i\|^2 \tag{181}$$

$$\leq 4n(m+1) \sum_{k=1}^{m} L_{y,l}^2 \|\bar{\mathbf{y}}_k^r - \bar{\mathbf{y}}_k^*(\mathbf{x}^r)\|^2 + 4mL_{f,y}^2 \|\mathbf{y}_k^r - \mathbb{1}\bar{\mathbf{y}}_k^r\|^2$$

$$+ 4mL_{f,y}^2 \sum_{k=1}^{m} \mathbb{E}\|\mathbf{y}_k^{r+1} - \mathbf{y}_k^r\|^2 + 4b_r^2, \tag{182}$$

and constant $L_{y,l}$ is computed as follows:

$$\sum_{i=1}^{n} \|\bar{\nabla} f_i(\mathbf{x}_i, \bar{\mathbf{y}}_k) - \nabla f_i(\mathbf{x}_i^r, \bar{\mathbf{y}}_k^*(\mathbf{x}))\|^2$$

$$\leq (m+1) \sum_{k=1}^{m} \sum_{i=1}^{n} \|\nabla_{\mathbf{x}} f_i(\mathbf{x}_i, \bar{\mathbf{y}}_k) - \nabla_{\mathbf{x}} f_i(\mathbf{x}_i, \bar{\mathbf{y}}_k^*(\mathbf{x}))\|^2 + (m+1) \left\| \Delta_{i,k}'^{(1)} + \Delta_{i,k}'^{(2)} + \Delta_{i,k}'^{(3)} \right\|^2$$

$$\leq n(m+1) \sum_{k=1}^{m} \underbrace{L_{f,1}^2 + 3\left( \left( \frac{L_{f,0}L_{g,2}}{\mu_g} \right)^2 + \left( \frac{L_{f,0}C_{xy}L_g}{\mu_g^2} \right)^2 + \left( \frac{C_{xy}L_{f,1}}{\mu_g} \right)^2 \right)}_{\triangleq L_{y,l}^2} \|\bar{\mathbf{y}}_k - \bar{\mathbf{y}}_k^*(\mathbf{x})\|^2,$$

and terms $\Delta_{i,k}^{\prime(1)}, \Delta_{i,k}^{\prime(2)}, \Delta_{i,k}^{\prime(3)}$ are defined as

$$
\begin{aligned}
\Delta_{i,k}^{\prime(1)} &\triangleq \nabla^2_{\mathbf{x}_i \mathbf{y}_{i,k}} g_{i,k}(\mathbf{x}_i, \bar{\mathbf{y}}_k) \left[ \nabla^2_{\mathbf{y}_{i,k} \mathbf{y}_{i,k}} g_{i,k}(\mathbf{x}_i, \bar{\mathbf{y}}_k) \right]^{-1} \nabla_{\mathbf{y}_{i,k}} f_i(\mathbf{x}_i, \bar{\mathbf{y}}_k) \\
&\quad - \nabla^2_{\mathbf{x}_i \mathbf{y}_{i,k}} g_{i,k}(\mathbf{x}_i, \bar{\mathbf{y}}_k^*(\mathbf{x})) \left[ \nabla^2_{\mathbf{y}_{i,k} \mathbf{y}_{i,k}} g_{i,k}(\mathbf{x}_i, \bar{\mathbf{y}}_k) \right]^{-1} \nabla_{\mathbf{y}_{i,k}} f_i(\mathbf{x}_i, \bar{\mathbf{y}}_k), \\
\Delta_{i,k}^{\prime(2)} &\triangleq \nabla^2_{\mathbf{x}_i \mathbf{y}_{i,k}} g_{i,k}(\mathbf{x}_i, \bar{\mathbf{y}}_k^*(\mathbf{x})) \left[ \nabla^2_{\mathbf{y}_{i,k} \mathbf{y}_{i,k}} g_{i,k}(\mathbf{x}_i, \bar{\mathbf{y}}_k) \right]^{-1} \nabla_{\mathbf{y}_{i,k}} f_i(\mathbf{x}_i, \bar{\mathbf{y}}_k) \\
&\quad - \nabla^2_{\mathbf{x}_i \mathbf{y}_{i,k}} g_{i,k}(\mathbf{x}_i, \bar{\mathbf{y}}_k^*(\mathbf{x})) \left[ \nabla^2_{\mathbf{y}_k \mathbf{y}_k} g_k(\mathbf{x}, \bar{\mathbf{y}}_k^*(\mathbf{x})) \right]^{-1} \nabla_{\mathbf{y}_{i,k}} f_i(\mathbf{x}_i, \bar{\mathbf{y}}_k), \\
\Delta_{i,k}^{\prime(3)} &\triangleq \nabla^2_{\mathbf{x}_i \mathbf{y}_{i,k}} g_{i,k}(\mathbf{x}_i, \bar{\mathbf{y}}_k^*(\mathbf{x})) \left[ \nabla^2_{\mathbf{y}_k \mathbf{y}_k} g_k(\mathbf{x}, \bar{\mathbf{y}}_k^*(\mathbf{x})) \right]^{-1} \nabla_{\mathbf{y}_{i,k}} f_i(\mathbf{x}_i, \bar{\mathbf{y}}_k) \\
&\quad - \nabla^2_{\mathbf{x}_i \mathbf{y}_{i,k}} g_{i,k}(\mathbf{x}_i, \bar{\mathbf{y}}_k^*(\mathbf{x})) \left[ \nabla^2_{\mathbf{y}_k \mathbf{y}_k} g_k(\mathbf{x}, \bar{\mathbf{y}}_k^*(\mathbf{x})) \right]^{-1} \nabla_{\mathbf{y}_{i,k}} f_i(\mathbf{x}_i, \bar{\mathbf{y}}_k^*(\mathbf{x})).
\end{aligned}
$$

Let us define potential function as

$$
\mathcal{P}_l^r = \sum_{i=1}^n \mathbb{E}\left[ f_i(\mathbf{x}_i^r, \bar{\mathbf{y}}_k^*(\bar{\mathbf{x}}^r)) | \mathcal{F}^r \right] + n(m+1) \sum_{k=1}^m \| \bar{\mathbf{y}}_k^r - \bar{\mathbf{y}}_k^*(\mathbf{x}^r) \|^2. \tag{183}
$$

Then, we can have

$$
\begin{aligned}
&\mathcal{P}_l^{r+1} - \mathcal{P}_l^r \\
\leq &-\frac{1}{2\alpha} \sum_{i=1}^n \| \nabla f_i(\mathbf{x}_i^r, \mathbf{y}_{i,k}^*(\mathbf{x}_i^r)) \|^2 \\
&- \Bigg( \underbrace{\frac{\alpha}{2} - \left( \frac{L_{f,1}}{2} + n(m+1)L_y^2 + \frac{\alpha}{8} + \frac{n(m+1)L_{xy}}{4\vartheta} \right)}_{\triangleq C_1} \Bigg) \| \mathbb{E}[\mathbf{x}^{r+1} - \mathbf{x}^r] \|^2 \\
&+ n(m+1) \Bigg( \underbrace{\frac{2L_{y,l}^2}{\alpha} + \left(1 - \frac{\rho_g}{\beta}\right)\left(1 + \frac{2(m+1)nL_y^2}{\alpha} + \frac{\vartheta L_{xy} D_x}{4\alpha^2}\right) - 1}_{\triangleq C_2} \Bigg) \sum_{k=1}^m \| \bar{\mathbf{y}}_k^r - \bar{\mathbf{y}}_k^*(\mathbf{x}^r) \|^2 \\
&+ \frac{2mL_{f,y}^2}{\alpha} \sum_{k=1}^m \| \mathbf{y}_k^r - \mathbb{1}\bar{\mathbf{y}}_k^r \|^2 + \frac{2mL_{f,y}^2}{\alpha} \sum_{k=1}^m \mathbb{E}\| \mathbf{y}_k^{r+1} - \mathbf{y}_k^r \|^2 \\
&+ \underbrace{n(m+1)\left( \left(1 + \frac{2(m+1)nL_y^2}{\alpha} + \frac{\vartheta L_{xy} D_x}{4\alpha^2}\right) \frac{\sigma_g^2}{n\beta^2} + \left(\frac{L_{xy}}{4\vartheta} + L_y^2\right)\frac{\sigma_f^2}{n\alpha^2} \right) + \frac{2b_r^2}{\alpha} + \frac{nL_{f,1}\sigma_f^2}{2\alpha^2}}_{\triangleq N_3 \sim \mathcal{O}\left(\frac{n}{\alpha^2}\right)}.
\end{aligned}
$$

1) to show $C_1 > 0$: we need

$$
\frac{\alpha}{2} - \left( \frac{L_{f,1}}{2} + n(m+1)L_y^2 + \frac{\alpha}{8} + \frac{n(m+1)L_{xy}}{4\vartheta} \right) > 0, \tag{184}
$$

which requires

$$
\alpha > 4\left( \frac{L_{f,1}}{2} + n(m+1)L_y^2 + \frac{n(m+1)L_{xy}}{4\vartheta} \right). \tag{185}
$$

2) to show $C_2 < 0$: we need

$$
\frac{2L_{y,l}^2}{\alpha} + \left(1 - \frac{\rho_g}{\beta}\right)\left(1 + \frac{2(m+1)nL_y^2}{\alpha} + \frac{\vartheta L_{xy} D_x}{4\alpha^2}\right) \leq 1, \tag{186}
$$

which is

$$
\beta < \frac{\rho_g(\alpha + 2(m+1)nL_y^2 + \frac{\vartheta L_{xy} D_x}{4\alpha})}{2(L_y^2 + L_{y,l}^2)}. \tag{187}
$$

Then, we can have constants $C_1 > 0$ and $C_2 < 0$ such that

$$
\begin{aligned}
\mathcal{P}_l^{r+1} - \mathcal{P}_l^r \leq &-\frac{1}{2\alpha} \| \nabla f_i(\mathbf{x}_i^r, \mathbf{y}_{i,k}^*(\mathbf{x}_i^r)) \|^2 - C_1 \| \mathbb{E}[\mathbf{x}^{r+1} - \mathbf{x}^r] \|^2 + n(m+1)C_2 \sum_{k=1}^m \| \bar{\mathbf{y}}_k^r - \bar{\mathbf{y}}_k^*(\mathbf{x}^r) \|^2 \\
&+ \frac{2mL_{f,y}^2}{\alpha} \sum_{k=1}^m \| \mathbf{y}_k^r - \mathbb{1}\bar{\mathbf{y}}_k^r \|^2 + \frac{2mL_{f,y}^2}{\alpha} \sum_{k=1}^m \mathbb{E}\| \mathbf{y}_k^{r+1} - \mathbf{y}_k^r \|^2 + N_3. \tag{188}
\end{aligned}
$$

Applying the telescoping sum, we can obtain

$$
\frac{1}{T}\sum_{r=1}^{T}\mathbb{E}\|\nabla f_i(\mathbf{x}_i^r,\mathbf{y}_{i,k}^*(\mathbf{x}_i^r))\|^2
$$

$$
\overset{(40\text{b})}{\leq} \frac{2\alpha\left(\mathcal{P}_l^1 - \underline{\mathcal{P}_l}\right)}{T} + \frac{4mL_{f,y}^2}{T\widetilde{\sigma}_{\min}^2(\mathbf{A}^T\mathbf{A})}\sum_{r=1}^{T}\sum_{i=1}^{m}\|\mathbf{A}\mathbf{y}_k^r\|^2 + 4m^2 L_{f,y}^2\frac{D_y}{\beta^2} + \alpha N_3 \sim \mathcal{O}\left(\frac{n}{\sqrt{T}}\right), \qquad (189)
$$

where $\underline{\mathcal{P}_l}$ denotes the lower bound of $\mathcal{P}_l$, which gives the result shown in Corollary 1.

### B.6.3  Optimality of the LL Optimization Variables

We know that $-1 < C_2 < 0$ based on (187) and condition $\beta \geq 2(\mu_g + L_{g,1})^{-1}$, and $\mathbb{E}_{\varepsilon^r}\left[\left\|\frac{1}{\alpha}\left(\mathbb{E}\mathbf{h}_f^r - \mathbf{h}_f^r\right)\right\|^2 | \mathcal{F}'^r\right] \leq n\sigma_f^2/\alpha$ in Lemma 5. Following (172) to (174) and choosing $\theta' = 2(m+1)n^{3/2}L_y^2/\alpha$, we have

$$
\frac{1}{T}\sum_{r=1}^{T}\mathbb{E}\|\bar{\mathbf{y}}_k^r - \bar{\mathbf{y}}_k^*(\mathbf{x}^r)\|^2 \sim \mathcal{O}\left(\frac{1}{\sqrt{nT}}\right), \qquad (190)
$$

when $\alpha \sim \mathcal{O}(\sqrt{T})$, $\beta \sim \mathcal{O}(\sqrt{T/n})$, and $T \gg n$.

### B.7  Proof of Corollary 2

In this case, (5a) reduces to

$$
\mathbf{y}_k^{r+1} = \mathbf{y}_k^r - \frac{\mathbf{h}_g^r}{\beta} \qquad (191)
$$

where $\mathbf{h}_g^r$ is the stochastic estimate of $[g_{1,k}(\mathbf{x}_1^r,\mathbf{y}_{1,k}^r),\ldots,g_{n,k}(\mathbf{x}_n^r,\mathbf{y}_{n,k}^r)]^T$.

#### B.7.1  Consensus Violation of the UL Variables

Following (138), we have $\frac{1}{T}\sum_{r=1}^{T}\mathbb{E}\|\mathbf{A}^T\mathbf{x}^r\|^2 \leq \mathcal{O}(1/\sqrt{nT})$, where $\alpha \sim \mathcal{O}(\sqrt{T/n})$ and $\beta \sim \mathcal{O}(\sqrt{T/n})$.

#### B.7.2  Stationarity of the UL Optimization Variables

Based on (191), (80) becomes

$$
\mathbb{E}\|\mathbf{y}_k^{r+1} - \mathbf{y}_k^*(\mathbf{x}^r)\|^2 \leq \left(1 - \frac{\rho_g}{\beta}\right)\mathbb{E}\|\mathbf{y}_k^r - \mathbf{y}_k^*(\mathbf{x}^r)\|^2 + \frac{n\sigma_g^2}{\beta^2}, \qquad (192)
$$

and there exist $\theta', \vartheta > 0$ such that

$$
\mathbb{E}\|\mathbf{y}_k^{r+1} - \mathbf{y}_k^*(\mathbf{x}^{r+1})\|^2 \leq \left(1 + \theta' + \frac{\vartheta L_{xy}D_x}{4\alpha^2}\right)\mathbb{E}\|\mathbf{y}_k^{r+1} - \mathbf{y}_k^*(\mathbf{x}^r)\|^2
$$
$$
+ \left(L_y^2 + \frac{L_y^2}{4\theta'} + \frac{L_{xy}}{4\vartheta}\right)\|\mathbb{E}\mathbf{x}^{r+1} - \mathbf{x}^r\|^2 + \left(\frac{L_{xy}}{4\vartheta} + L_y^2\right)\frac{\sigma_f^2}{n\alpha^2}. \qquad (193)
$$

Therefore, (161) becomes

$$
\|\mathbf{y}_k^{r+1} - \mathbf{y}_k^*(\mathbf{x}^{r+1})\|^2 \leq \left(1 - \frac{\rho_g}{\beta}\right)\left(1 + \frac{2mnL_y^2}{\alpha} + \frac{\vartheta L_{xy}D_x}{4\alpha^2}\right)\|\mathbf{y}_k^r - \mathbf{y}_k^*(\mathbf{x}^r)\|^2
$$
$$
+ \left(L_y^2 + \frac{\alpha}{8mn} + \frac{L_{xy}}{4\vartheta}\right)\left\|\mathbb{E}\left[\mathbf{x}^{r+1} - \mathbf{x}^r\right]\right\|^2
$$
$$
+ \left(1 + \frac{2mnL_f^2}{\alpha} + \frac{\vartheta L_{xy}D_x}{4\alpha^2}\right)\frac{n\sigma_g^2}{\beta^2} + \left(\frac{L_{xy}}{4\vartheta} + L_y^2\right)\frac{\sigma_f^2}{n\alpha^2} \qquad (194)
$$

where $\theta' = 2mnL_y^2/\alpha$.

According to the gradient Lipschitz continuity of the UL objective function, we can have

$$\mathbb{E}_{\xi^r}\left[f(\overline{\mathbf{x}}^{r+1}, \mathbf{y}_k^*(\overline{\mathbf{x}}^{r+1}))|\mathcal{F}'^r\right]$$

$$\overset{(a)}{\leq} f(\overline{\mathbf{x}}^r, \mathbf{y}_k^*(\overline{\mathbf{x}}^r)) + \mathbb{E}\left[\langle\nabla f(\overline{\mathbf{x}}^r, \mathbf{y}_k^*(\overline{\mathbf{x}}^r)), \overline{\mathbf{x}}^{r+1} - \overline{\mathbf{x}}^r\rangle|\mathcal{F}'^r\right] + \mathbb{E}\left[\frac{L_f}{2}\|\overline{\mathbf{x}}^{r+1} - \overline{\mathbf{x}}^r\|^2|\mathcal{F}'^r\right] \tag{195}$$

$$= f(\overline{\mathbf{x}}^r, \mathbf{y}_k^*(\overline{\mathbf{x}}^r)) + \langle\nabla f(\overline{\mathbf{x}}^r, \mathbf{y}_k^*(\overline{\mathbf{x}}^r)), \mathbb{E}[\overline{\mathbf{x}}^{r+1} - \overline{\mathbf{x}}^r]|\mathcal{F}'^r\rangle$$
$$+ \frac{L_f}{2}\|\mathbb{E}[\overline{\mathbf{x}}^{r+1} - \overline{\mathbf{x}}^r]\|^2 + \frac{L_f}{2}\mathbb{E}\left[\|\overline{\mathbf{x}}^{r+1} - \overline{\mathbf{x}}^r - \mathbb{E}[\overline{\mathbf{x}}^{r+1} - \overline{\mathbf{x}}^r]\|^2|\mathcal{F}'^r\right] \tag{196}$$

$$\overset{(b)}{\leq} f(\overline{\mathbf{x}}^r, \mathbf{y}_k^*(\overline{\mathbf{x}}^r)) - \frac{1}{2\alpha}\|\nabla f(\overline{\mathbf{x}}^r, \mathbf{y}_k^*(\overline{\mathbf{x}}^r))\|^2 - \left(\frac{\alpha}{2} - \frac{L_f}{2}\right)\|\mathbb{E}[\overline{\mathbf{x}}^{r+1} - \overline{\mathbf{x}}^r]\|^2$$
$$+ \frac{1}{2\alpha}\|\nabla f(\overline{\mathbf{x}}^r, \mathbf{y}_k^*(\overline{\mathbf{x}}^r)) - \overline{\mathbf{h}}_f^r\|^2 + \frac{L_f}{2\alpha^2}\mathbb{E}\left[\|\overline{\mathbf{h}}_f^r - \mathbb{E}\overline{\mathbf{h}}_f^r\|^2\mathcal{F}'^r\right] \tag{197}$$

$$\overset{(c)}{\leq} f(\overline{\mathbf{x}}^r, \overline{\mathbf{y}}_k^*(\overline{\mathbf{x}}^r)) - \frac{1}{2\alpha}\|\nabla f(\overline{\mathbf{x}}^r, \mathbf{y}_k^*(\overline{\mathbf{x}}^r))\|^2 - \left(\frac{\alpha}{2} - \frac{L_f}{2}\right)\|\mathbb{E}[\overline{\mathbf{x}}^r - \overline{\mathbf{x}}^r]\|^2 + \frac{2L_{f,x}^2 m^2}{n^2\alpha}\|\mathbf{x}^r - \mathbb{1}\overline{\mathbf{x}}^r\|^2$$
$$+ \frac{2mL_{y,u}^2}{n^2\alpha}\sum_{k=1}^m\|\mathbf{y}_k^r - \mathbf{y}_k^*(\overline{\mathbf{x}}^r)\|^2 + \frac{2mL_{f,y}^2}{n^2\alpha}\mathbb{E}\|\mathbf{y}_k^{r+1} - \mathbf{y}_k^r\|^2 + \frac{2b_r^2}{n\alpha} + \frac{L_f\sigma_f^2}{2n\alpha^2}$$

where $(a)$ follows the gradient Lipschitz continuity of the UL loss function with constant $L_f$ at the consensus space, $(b)$ is true because

$$\langle\nabla f(\overline{\mathbf{x}}^r, \mathbf{y}_k^*(\overline{\mathbf{x}}^r)), \mathbb{E}[\overline{\mathbf{x}}^{r+1} - \overline{\mathbf{x}}^r]|\mathcal{F}'^r\rangle$$
$$= -\frac{1}{\alpha}\langle\nabla f(\overline{\mathbf{x}}^r, \mathbf{y}_k^*(\overline{\mathbf{x}}^r)), \mathbb{E}\overline{\mathbf{h}}_f^r\rangle$$
$$= -\frac{1}{2\alpha}\|\nabla f(\overline{\mathbf{x}}^r, \mathbf{y}_k^*(\overline{\mathbf{x}}^r))\|^2 - \frac{1}{2\alpha}\|\mathbb{E}\overline{\mathbf{h}}_f^r\|^2 + \frac{1}{2\alpha}\|\nabla f(\overline{\mathbf{x}}^r, \mathbf{y}_k^*(\overline{\mathbf{x}}^r)) - \mathbb{E}\overline{\mathbf{h}}_f^r\|^2,$$

and $(c)$ we can apply the following facts,

$$\|\nabla f(\overline{\mathbf{x}}^r, \mathbf{y}_k^*(\overline{\mathbf{x}}^r)) - \mathbb{E}\overline{\mathbf{h}}_f^r\|^2$$
$$\leq 4\|\nabla f(\overline{\mathbf{x}}^r, \mathbf{y}_k^*(\overline{\mathbf{x}}^r)) - \overline{\nabla}f(\overline{\mathbf{x}}^r, \mathbf{y}_k^r)\|^2 + 4\|\overline{\nabla}f(\overline{\mathbf{x}}^r, \mathbf{y}_k^r) - \overline{\nabla}f(\mathbf{x}^r, \mathbf{y}_k^r)\|^2$$
$$+ 4\|\overline{\nabla}f(\mathbf{x}^r, \mathbf{y}_k^r) - \overline{\nabla}f(\mathbf{x}^r, \mathbf{y}_k^{r+1})\|^2 + 4\|\overline{\nabla}f(\mathbf{x}^r, \mathbf{y}_k^{r+1}) - \mathbb{E}\overline{\mathbf{h}}_f^r\|^2 \tag{198}$$
$$\leq \frac{4mL_{y,u}^2}{n^2}\sum_{k=1}^m\|\mathbf{y}_k^r - \mathbf{y}_k^*(\overline{\mathbf{x}}^r)\|^2 + \frac{4L_{f,x}^2 m^2}{n^2}\|\mathbf{x}^r - \mathbb{1}\overline{\mathbf{x}}^r\|^2$$
$$+ \frac{4mL_{f,y}^2}{n^2}\sum_{k=1}^m\mathbb{E}\|\mathbf{y}_k^{r+1} - \mathbf{y}_k^r\|^2 + \frac{4b_r^2}{n}, \tag{199}$$

and the continuity of $\nabla f_i(\mathbf{x}_i^r, \mathbf{y}_{i,k}^*(\mathbf{x}_i^r))$ is the same as the centralized case (e.g., constant $C$ in [31, Lemma 2.2.]) as variable $\mathbf{y}_{i,k}, \forall i$ are decoupled over the nodes, namely $\|\nabla f_i(\mathbf{x}_i^r, \mathbf{y}_{i,k}^*(\mathbf{x}_i^r)) - \nabla f_i(\mathbf{x}_i^r, \mathbf{y}_{i,k})\| \leq L_{y,u}\|\mathbf{y}_{i,k}^*(\mathbf{x}_i^r) - \mathbf{y}_{i,k}\|, \forall i, k$.

Define potential function as

$$\mathcal{P}_u^r = \mathbb{E}\left[f(\overline{\mathbf{x}}^r, \mathbf{y}_k^*(\overline{\mathbf{x}}^r))|\mathcal{F}^r\right] + \frac{m}{n^2}\sum_{k=1}^m\|\mathbf{y}_k^r - \mathbf{y}_k^*(\overline{\mathbf{x}}^r)\|^2. \tag{200}$$

Combining (193), we have Then, we can have

$$\mathcal{P}_u^{r+1} - \mathcal{P}_u^r$$

$$\leq -\frac{1}{2\alpha}\|\nabla f(\bar{\mathbf{x}}^r, \mathbf{y}_k^*(\bar{\mathbf{x}}^r))\|^2 - \underbrace{\left(\frac{\alpha}{2} - \left(\frac{L_f}{2} + \frac{m}{n^2}L_y^2 + \frac{\alpha}{8n^3} + \frac{mL_{xy}}{4n^2\vartheta}\right)\right)}_{\triangleq C_1}\|\mathbb{E}[\bar{\mathbf{x}}^{r+1} - \bar{\mathbf{x}}^r]\|^2$$

$$+\frac{m}{n^2}\left(\underbrace{\frac{2L_{y,u}^2}{\alpha} + \left(1 - \frac{\rho_g}{\beta}\right)\left(1 + \frac{2nmL_y^2}{\alpha} + \frac{\vartheta L_{xy}D_x}{4\alpha^2}\right) - 1}_{\triangleq C_2}\right)\sum_{k=1}^m \|\mathbf{y}_k^r - \mathbf{y}_k^*(\bar{\mathbf{x}}^r)\|^2$$

$$+\frac{2L_{f,x}^2 m^2}{n^2\alpha}\|\mathbf{x}^r - \mathbb{1}\bar{\mathbf{x}}^r\|^2 + \frac{2mL_{f,y}^2}{n^2\alpha}\sum_{k=1}^m \mathbb{E}\|\mathbf{y}_k^{r+1} - \mathbf{y}_k^r\|^2$$

$$+\underbrace{\frac{m}{n^2}\left(\left(1 + \frac{2mnL_y^2}{\alpha} + \frac{\vartheta L_{xy}D_x}{4\alpha^2}\right)\frac{n\sigma_g^2}{\beta^2} + \left(\frac{L_{xy}}{4\vartheta} + L_y^2\right)\frac{\sigma_f^2}{n\alpha^2}\right) + \frac{2b_r^2}{n\alpha} + \frac{L_f\sigma_f^2}{2n\alpha^2}}_{\triangleq N_4 \sim \mathcal{O}\left(\frac{1}{n\alpha^2}\right)}.$$

Following (164) and (167), we can have constants $C_1 > 0$ and $C_2 < 0$ such that

$$\mathcal{P}_u^{r+1} - \mathcal{P}_u^r \leq -\frac{1}{2\alpha}\|\nabla f(\bar{\mathbf{x}}^r, \mathbf{y}_k^*(\bar{\mathbf{x}}^r))\|^2 - C_1\|\mathbb{E}[\bar{\mathbf{x}}^{r+1} - \bar{\mathbf{x}}^r]\|^2 + \frac{m}{n^2}C_2\sum_{k=1}^m \|\bar{\mathbf{y}}_k^r - \bar{\mathbf{y}}_k^*(\bar{\mathbf{x}}^r)\|^2$$

$$+\frac{2L_{f,x}^2 m^2}{n^2\alpha}\|\mathbf{x}^r - \mathbb{1}\bar{\mathbf{x}}^r\|^2 + \frac{2mL_{f,y}^2}{n^2\alpha}\sum_{k=1}^m \mathbb{E}\|\mathbf{y}_k^{r+1} - \mathbf{y}_k^r\|^2 + N_4. \tag{201}$$

Applying the telescoping sum, we have that when $\alpha \sim \mathcal{O}(\sqrt{T/n})$ and $\beta \sim \mathcal{O}(\sqrt{T/n})$,

$$\frac{1}{T}\sum_{r=1}^T \|\nabla f(\bar{\mathbf{x}}^r, \mathbf{y}_k^*(\bar{\mathbf{x}}^r))\|^2 \leq 2\alpha\frac{\mathcal{P}_u^1 - \underline{\mathcal{P}_u}}{T} + \frac{4L_{f,x}^2 m^2}{n^2\widetilde{\sigma}_{\min}(\mathbf{A}^T\mathbf{A})}\frac{1}{T}\sum_{r=1}^T \mathbb{E}\|\mathbf{A}\mathbf{x}_k^r\|^2$$

$$+\frac{4m^2 L_{f,y}^2}{n^2}\frac{D_y}{\beta^2} + 2\alpha N_4 \sim \mathcal{O}\left(\frac{1}{\sqrt{nT}}\right), \tag{202}$$

where $\underline{\mathcal{P}_u}$ denotes the lower bound of $\mathcal{P}_u$.

### B.7.3 Optimality of the LL Optimization Variables

From (193) and condition $\beta \geq 2(\mu_g + L_{g,1})^{-1}$, we know that $-1 < C_2 < 0$. With $\theta' = 2mnL_y^2/\alpha$, so (194) can be written as

$$(1 + C_2)\mathbb{E}\|\mathbf{y}_k^r - \mathbf{y}_k^*(\mathbf{x}^r)\|^2$$

$$\leq \mathbb{E}\|\mathbf{y}_k^r - \mathbf{y}_k^*(\mathbf{x}^r)\|^2 - \mathbb{E}\|\mathbf{y}_k^{r+1} - \mathbf{y}_k^*(\mathbf{x}^{r+1})\|^2 + \left(L_y^2 + \frac{\alpha}{8mn} + \frac{L_{xy}}{4\vartheta}\right)\|\mathbb{E}[\mathbf{x}^{r+1} - \mathbf{x}^r]\|^2$$

$$+\left(1 + \frac{2mnL_y^2}{\alpha} + \frac{\vartheta L_{xy}D_x}{4\alpha^2}\right)\frac{n\sigma_g^2}{\beta^2} + \frac{L_{xy}}{4\vartheta}\frac{\sigma_f^2}{n\alpha^2}. \tag{203}$$

Applying the telescoping sum on (203), we have

$$\frac{1}{T}\sum_{r=1}^T \mathbb{E}\|\mathbf{y}_k^r - \mathbf{y}_k^*(\mathbf{x}^r)\|^2$$

$$\leq \frac{\|\mathbf{y}_k^1 - \mathbf{y}_k^*(\mathbf{x}^1)\|^2}{T(1 + C_2)} + \frac{1}{1 + C_2}\left(L_y^2 + \frac{\alpha}{8mn} + \frac{L_{xy}}{4\vartheta}\right)\frac{1}{T}\sum_{r=1}^T \mathbb{E}\|\mathbf{x}^{r+1} - \mathbf{x}^r\|^2$$

$$+\frac{1}{1 + C_2}\left(1 + \frac{2mnL_y^2}{\alpha} + \frac{\vartheta L_{xy}D_x}{4\alpha^2}\right)\frac{n\sigma_g^2}{\beta^2} + \frac{1}{1 + C_2}\frac{L_{xy}}{4\vartheta}\frac{\sigma_f^2}{n\alpha^2} \tag{204}$$

$$\leq \frac{\|\mathbf{y}_k^1 - \mathbf{y}_k^*(\mathbf{x}^1)\|^2}{T(1 + C_2)} + \frac{1}{1 + C_2}\left(L_y^2 + \frac{\alpha}{8mn} + \frac{L_{xy}}{4\vartheta}\right)\frac{D_x}{\alpha^2}$$

$$+\frac{1}{1 + C_2}\left(1 + \frac{2mnL_y^2}{\alpha} + \frac{\vartheta L_{xy}D_x}{4\alpha^2}\right)\frac{n\sigma_g^2}{\beta^2} + \frac{1}{1 + C_2}\frac{L_{xy}}{4\vartheta}\frac{\sigma_f^2}{n\alpha^2} \sim \mathcal{O}\left(\frac{1}{\sqrt{nT}}\right), \tag{205}$$

as we choose $\alpha \sim \mathcal{O}(\sqrt{T/n})$, $\beta \sim \mathcal{O}(\sqrt{T/n})$, and $T \gg n$.

## C   Additional Numerical Results

### C.1   SLAM for MARL: SLAM-AC

In this section, we will introduce how to use SLAM for solving MARL problems. Consider a networked Markov Decision Process (nMDP) $(\mathcal{S}, \{\mathcal{A}_i, \forall i\}, \mathcal{P}, \{R_i, \forall i\}, \eta, \mathcal{G}, \gamma)$, where $\mathcal{S}$ denotes the global states shared by all agents, $\mathcal{A}_i$ is the action space of agent $i$, subsequently $\mathcal{A} = \Pi_{i=1}^n \mathcal{A}_i$ is the joint action space of all agents, $\mathcal{P} : \mathcal{S} \times \mathcal{A} \times \mathcal{S} \to [0, 1]$ is the state transition probability of the nMDP, $R_i(s, \mathbf{a}) : \mathcal{S} \times \mathcal{A} \to \mathbb{R}, \forall i$ denote the local rewards, $\eta(s)$ denotes the initial state, $\mathcal{G}$ is the communication graph, and $\gamma \in (0, 1)$ stands for the discount factor. We assume that the states $s$ and actions $\mathbf{a}$ are globally observable. The goal of this problem is to learn a joint policy $\pi_\theta$ parametrized by $\theta$ such that the networked reward function is maximized. Let $\theta_i$ be the local policy at each agent and the concatenation of all local policies be $\theta = [\theta_1, \ldots, \theta_n]^T$ (which will be the UL optimization variables in the DAC formulation). Under this setting, $\mu_\theta(s)$ is the stationary distribution induced by the policy $\pi_\theta$ at each state, and $d_\theta(s) = (1 - \gamma) \sum_{r=0}^{\infty} \gamma^r \mathcal{P}(s^r = s | s^0 \sim \eta(s))$ denotes the discounted visitation measure. Therefore, given the initial state $\eta(s)$, the policy $\pi_\theta$ can generate a trajectory according to the nMDP. Then, the discounted accumulative reward function maximization problem w.r.t. optimizing the policy is $\max_\theta J(\theta)$, and the objective function is

$$J(\theta) \triangleq \frac{1}{n} \sum_{i=1}^n \mathbb{E}_{\pi_\theta} \left[ \sum_{r=0}^{\infty} \gamma^r R_i(s^r, \mathbf{a}^r) \right] \tag{206}$$

$$= \mathbb{E}_{\pi_\theta} \left[ \sum_{r=0}^{\infty} \gamma^r \bar{R}(s^r, \mathbf{a}^r) \right] \tag{207}$$

$$= \mathbb{E}_{s \sim \eta(\cdot)} \left[ V_{\pi_\theta}(s) \right] \tag{208}$$

where $\bar{R}(s, \mathbf{a}) = n^{-1} \sum_{i=1}^n R_i(s, \mathbf{a})$, and the expectation is taken over all the trajectories generated by policy $\pi_\theta$, and value function $V_{\pi_\theta}(s) \triangleq \mathbb{E} \left[ \sum_{r=0}^{\infty} \gamma^r \bar{R}(s^r, \mathbf{a}^r) | s^0 = s \right]$. Note that given policy $\pi_\theta$, value function $V_{\pi_\theta}(s)$ satisfies the Bellman equation [7] and can be written as

$$V_{\pi_\theta}(s) = \mathbb{E}_{\mathbf{a} \sim \pi_\theta(\cdot|s), s' \sim \mathcal{P}(\cdot|s,\mathbf{a})} \left[ \bar{R}(s, \mathbf{a}) + \gamma V_{\pi_\theta}(s') \right], \tag{209}$$

so the policy gradient [50] of $J(\theta)$ w.r.t. $\theta$ can be expressed by

$$\nabla J(\theta) = \frac{1}{1 - \gamma} \mathbb{E}_{s \sim d_\theta(\cdot), \mathbf{a} \sim \pi_\theta(\cdot|s), s' \sim \mathcal{P}(\cdot|s,\mathbf{a})} \left[ \left( \bar{R}(s, \mathbf{a}) + \gamma V_{\pi_\theta}(s') \right) \nabla_\theta \log \pi_\theta(\mathbf{a}|s) \right]. \tag{210}$$

Based on the finite sum structure of the rewards over the network and policy gradient theorem, it is motivated that the policy gradient at each node can be estimated locally when a consensus process is allowed and the global action and state are observable. As the value function is not explicitly known at each agent, we apply the critic step to approximate the global valuation function by formulating the LL function estimation process. Following the existing AC works [30, 23, 15], we also adopt the linear function approximation, i.e., $\widehat{V}(s, w) = \varphi(s)^T w$, to estimate the global value function, where $\varphi(s)$ denotes the feature mapping and $w$ is the model parameter. Then, in this decentralized setting, the global value functions are estimated by $\widehat{V}(s, w_i) = \varphi(s)^T w_i, \forall i$ and subsequently the global reward functions can be estimated through $\widehat{R}_i(s, \mathbf{a}, \phi_i) = \psi(s, \mathbf{a})^T \phi_i, \forall i$, where $\{w_i, \phi_i, \forall i\}$ are local model parameters and $\psi(s, \mathbf{a})$ is the given feature mapping. Towards this end, the DAC problem can be formulated as the following DBO problem,

$$\max_\theta \quad J(\theta) = \mathbb{E}_{\pi_\theta} \left[ \sum_{r=0}^{\infty} \gamma^r \bar{R}(s^r, \mathbf{a}^r) \right] \tag{211a}$$

$$\text{s.t.} \quad \theta_i^p = \theta_j^p, \quad j \in \mathcal{N}_i, \forall i \tag{211b}$$

$$w^*(\theta) = \arg\min_w \frac{1}{n} \sum_{i=1}^n \mathbb{E} \left( R_i(s, \mathbf{a}) + \gamma \widehat{V}(s', w_i) - \widehat{V}(s, w_i) \right)^2, \text{s.t.} \ w_i = w_j, \ j \in \mathcal{N}_i, \forall i \tag{211c}$$

$$\phi^*(\theta) = \arg\min_\phi \frac{1}{n} \sum_{i=1}^n \mathbb{E} \left( R_i(s, \mathbf{a}) - \widehat{R}_i(s, \mathbf{a}, \phi_i) \right)^2, \text{s.t.} \ \phi_i = \phi_j \ j \in \mathcal{N}_i, \forall i \tag{211d}$$

where $\theta_i^p$ is the shared part of the agent i's policy parameter, $w = [w_1, \ldots, w_n]^T$ and $\phi = [\phi_1, \ldots, \phi_n]^T$. The LL problem (211c) is used for approximating the network value function and (211d) for the averaged reward function, and both of them are needed at the UL problem (211a) for computing the policy gradient w.r.t. local policies at each iteration, which can be explicitly approximated by

$$\widehat{\nabla}_{\theta_i} J(\theta) = (1 - \gamma)^{-1} \left( \psi(s^r, \mathbf{a}^r)^T \phi_i^r + \gamma \varphi(s^{r+1})^T w_i^r - \varphi(s^r)^T w_i^r \right) \nabla_{\theta_i} \log \pi_{\theta_i}(\mathbf{a}_i^r | s^r). \tag{212}$$

Given the above setting, we implement SLAM for solving DAC problem (211) in a decentralized way. The results are shown in both main text and the following. All the experiments are executed on an Apple MacBook Pro (8 GB Memory, M1 processor).

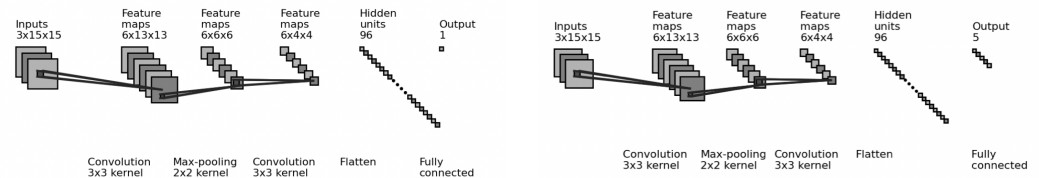

Figure 3: **Neural Network Architecture Diagrams in the Pursuit-Evasion Game.** (Left) The diagram of the **critic** network. (Right) The diagram of the **actor** network.

## C.2 Cooperative Navigation Task

In our simulations, the dimension of state $s^r$ is $4 \times n$, since it includes the two-dimensional coordinates of $n$ agents and $n$ landmarks. Both the actor and critic networks maintained at each agent have one hidden layer with 20 neurons followed by the ReLU activation function. Under a common policy $\pi_\theta = \{\pi_{\theta_i}\}_{i=1}^n$, critic networks jointly estimate the global value function $V_\pi(s)$ for all $s \in \mathcal{S}$, where the dimension of the output layer is 1. While the output of actor network corresponds to the probability of choosing each possible action option and the dimension of the output layer is 5 as there are 5 given actions in total. We set the step size for the critic network as $1 \times 10^{-3}$ and the step size for the actor network as $1 \times 10^{-4}$ for all the compared algorithms. Moreover, we set $\rho = 1 \times 10^{-1}$ in SLAM-AC. The results have been shown in Figure 1.

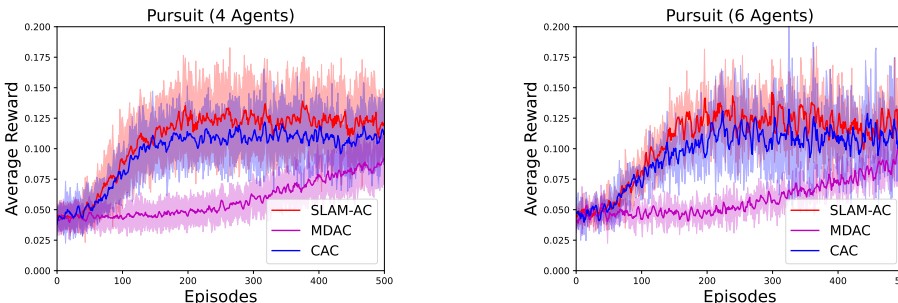

Figure 4: The averaged reward versus the learning process on the pursuit-evasion game. **(With only LL Consensus.)**

## C.3 Pursuit-Evasion Game

In this experiment, the "capture" reward for each agent is set to be 5 when a pursuer successfully catches an evader. Additionally, the pursuer will receive a small reward set to be $0.1$ when the pursuer encounters an evader at its current location. The environment is divided by a $15 \times 15$ grid where there exist obstacles in this two-dimensional (2D) grid such that the agents cannot pass through. Hence, the global state of the pursuit-evasion game consists of three images (binary matrices) with the size of $15 \times 15$. Consequently, the dimension of the global state is $3 \times 15 \times 15$. These three images (binary matrices) respectively present the location of the pursuers, evaders, and obstacles in this 2D grid.

Since the observation of each agent is a 3-channel image, two convolutional neural networks (CNNs) are each agent respectively maintained at each agent, including two convolutional layers, one max-pooling layer, and one fully connected layer for both the actor and critic learners. The ReLU activation function is utilized in each hidden layer of actor network and critic networks. The output of critic network targets approximating the value function $V_\pi(s)$ for all $s \in \mathcal{S}$, where the dimension of the output layer is 1. The output dimension of actor network is 5 which corresponds to the number of possible actions. In each CNN, the raw images (3-channel location matrices), whose dimension is $3 \times 15 \times 15$, are processed by two convolutional layers and one max-pooling layer first and then passed through a fully connected layer as the output layer. The detailed structure diagrams used for the actor and critic networks are shown in Figure 3. For all algorithms, we set the step size for the critic network as $1 \times 10^{-3}$ and the step size for the actor network as $1 \times 10^{-4}$. Moreover, we set $\rho = 5$ in SLAM-AC.

Additional results are shown in Figure 4, where only critic neural networks are used for consensus for estimating the network value function through the communication channel. All the algorithms adopt the same settings. It can be observed from these figures that our proposed algorithm, SLAM-AC, outperforms the state-of-the-art algorithms w.r.t. both convergence speed and achievable average rewards.

## C.4  Scalability, robustness, and extendability of SLAM in MARL

In this section, we add more numerical results on MARL and decentralized MAML problems respectively to showcase the superiority of the proposed learning framework and efficiency of SLAM, including scalability, robustness, and extendability.

Based on the problem setup and parameters chosen shown in Section C, we compare the scalability of the three algorithms on a different number of agents and test the performance of SLAM by varying the hyper-parameters.

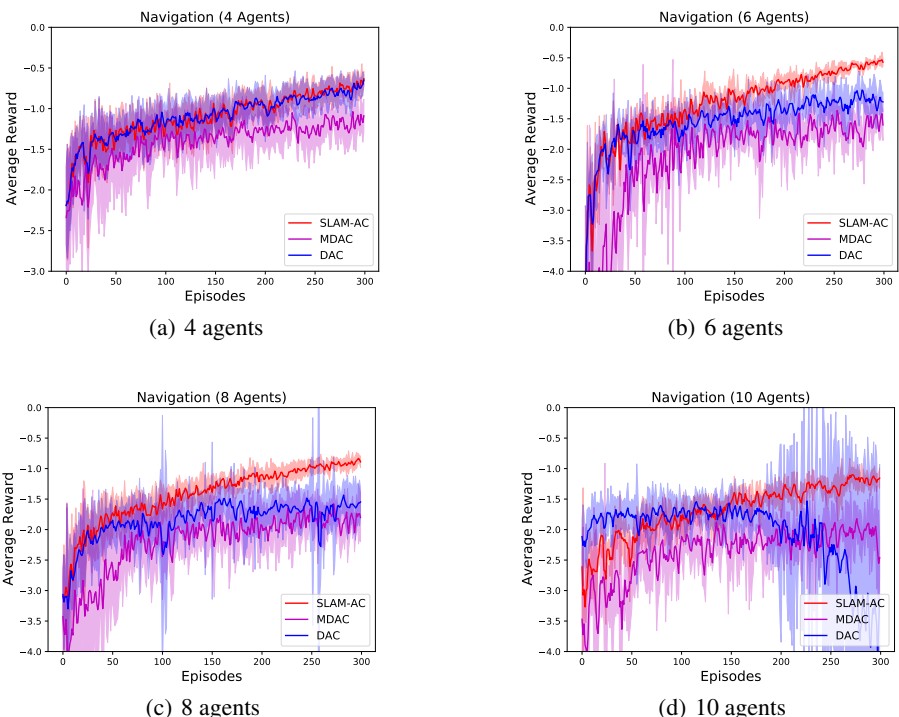

Figure 5: The averaged reward versus the learning process on the cooperative navigation task over different numbers of agents.

### C.4.1  Scalability

It is well known that the variance of the multi-agent policy gradient estimate grows as the number of agents increases [18], implying that the difficulty of solving large-scale multi-agent problems rises. We use the same set of step sizes for each algorithm and test their performances by increasing the number of agents from 4 to 10. It can be observed in Figure 5 that SLAM-AC consistently performs well and converges stably with minimum variances compared with the other two existing methods, where the average reward achieved by the classic MARL algorithm, DAC, has higher variances and even diverges when the number of agents is increased to 10, and the reward obtained by MDAC is relatively stable compared to DAC as the multiple consensus steps in the inner loop reduces the variances, but it is still lower than SLAM-AC. Therefore, it is concluded that SLAM-AC scales better than the other two due to the averaged variance over the network.

### C.4.2  Effects on Hyper-parameters

Besides the scalability of SLAM, we have also evaluated the numerical performance of SLAM by varying the hyper-parameters, including the actor and critic step sizes and the penalty term. It can be seen in Figure 6 that when the step sizes and penalty term shrink by a half and one-tenth, SLAM-AC converges slower while keeping a similar convergence behavior.

### C.4.3  Extendability of SLAM-L with PPO

The proposed SLAM based learning framework is generic, and is amenable to incorporating other optimization techniques at each agent to improve the performance of learning models. In applications of MARL, we further consider two algorithms, denoted as SLAM-PPO and Dec-PPO. Both of them apply proximal policy optimization

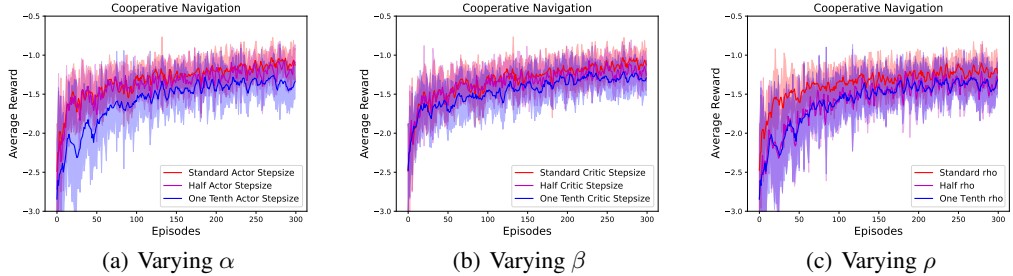

(a) Varying $\alpha$ (b) Varying $\beta$ (c) Varying $\rho$

Figure 6: The averaged reward versus the learning process by varying hyper-parameters of SLAM on the cooperative navigation task.

(PPO) instead of the vanilla policy gradient to update their actors (upper-level parameters). In the updates of the critic, the difference is that SLAM-PPO uses SLAM-L to minimize the temporal difference (TD) errors while Dec-PPO directly utilizes the decentralized gradient descent for policy evaluation. We test these two algorithms on the navigation task, where the $\epsilon$ clip threshold in PPO is set as 0.2. It can be observed from Figure 7 that SLAM-PPO converges to higher reward values with less variance compared with DEC-PPO, which is similar to the numerical results by employing policy gradient updates for the actor networks.

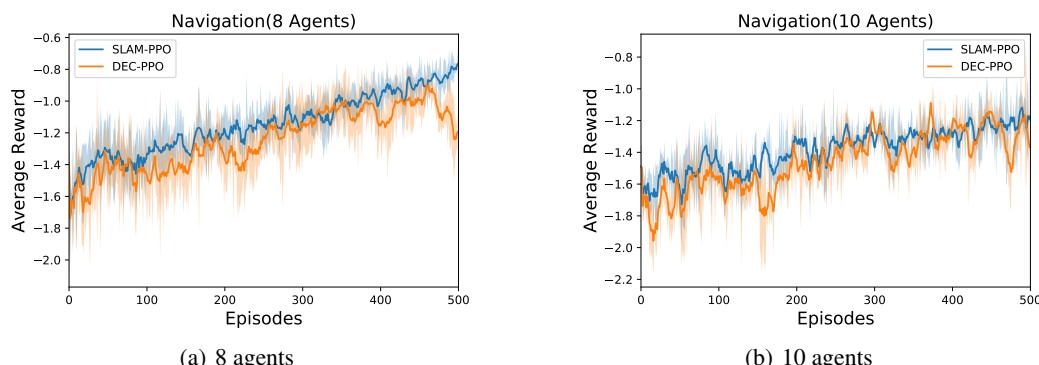

(a) 8 agents (b) 10 agents

Figure 7: The averaged reward versus the learning process on the cooperative navigation task with PPO for policy improvment.

## C.5 Decentralized Meta Learning

We further test the performance of the proposed SLAM with application to a multi-agent MAML problem (9). Following the ANIL structure [6], we partition the weights of the neural network at each agent as two parts denoted by $\mathbf{x}_i$ and $\mathbf{y}_i$, where the UL optimization problem is to extract the reusable latent space across the connected agents while the LL optimization problem is adopted for adaption of the local model to individual learning tasks.

In this numerical experiment, a two-layer neural network is used, where the numbers of neurons at the hidden and output layers are 32 and 10 respectively and the activation function is sigmoid. A 2-norm regularization term with parameter 0.01 is added to the LL loss function. The communication topology is generated by a random Erdős–Rényi graph. We divide the MNIST dataset as $n$ parts, where each of them only includes 128 data samples for the training. The (standard) initial UL and LL step sizes of SLAM-U are 0.01 and 0.1, and the mini-batch size for gradient estimate is 32.

### C.5.1 Linear Speed-up

From the numerical results shown in Figure 8, it can be observed that as the number of agents increases, SLAM-U converges faster in terms of the data samples passed. Setting 93.5% test accuracy as a threshold, we measure the number of data sampled passed w.r.t. the different numbers of agents and report the results in Table 2 and Figure 8(b). If we assume that the computational evaluations of the stochastic gradient estimate are the same at each agent, Figure 8(b) shows that there at least exists a linear speed-up w.r.t. the number of agents in terms of the computational workload, which is consistent with the theoretical analysis.

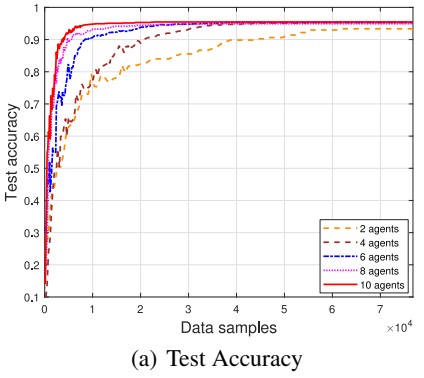
(a) Test Accuracy

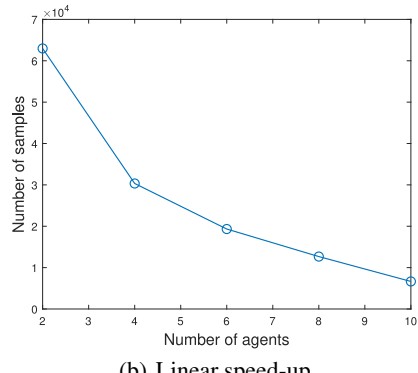
(b) Linear speed-up

Figure 8: The test accuracy versus the numbers of data samples passed at each agent on the decentralized metal-learning task.

Table 2: Linear speed-up of SLAM-U w.r.t. the number of agents.

| number of agents ($n$) | 2 | 4 | 6 | 8 | 10 |
|---|---|---|---|---|---|
| required number of samples passed | 62976 | 30336 | 19328 | 12672 | 6656 |

### C.5.2 Effects on Hyper-parameters

Similar to the MARL case, we use different step sizes and penalty parameters to test the robustness of SLAM-U against the changes of these hyper-parameters. From Figure 9, it can be seen that if the step sizes and $\rho$ decrease, SLAM-U converges relatively slower, and the performance of SLAM-U is more sensitive to the UL step size while very robust to the LL step size and $\rho$.

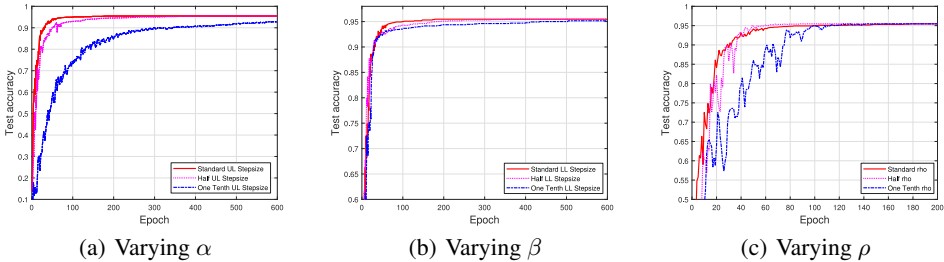
(a) Varying $\alpha$     (b) Varying $\beta$     (c) Varying $\rho$

Figure 9: The test accuracy versus the numbers of epochs on the decentralized metal-learning task.