# OpenReview forum: "A Stochastic Linearized Augmented Lagrangian Method for Decentralized Bilevel Optimization"
_NeurIPS.cc/2022/Conference — NeurIPS 2022 Accept_

### Official Review · Reviewer_6ivX · 2022-07-21

**Rating:** 6
**Confidence:** 3
**Soundness:** 3 good
**Presentation:** 2 fair
**Contribution:** 3 good

**Summary:**

In the study, the bilevel optimization framework is provided based on the linearized augmented lagrangian method, the detailed algorithms are provided, and the experimental results indicate the effectiveness of the method.

**Questions:**

1. In Page 6, Algorithm 1, how do the coefficient factors $\boldsymbol{\lambda}=\omega_{k}$ improve the performance? Could the study provide the analysis?
2. Can the study be extended to some strong baselines, e.g. MACPO and MAPPO-L [Gu, et al., 2021], HATRPO and HAPPO algorithms [Kuba, et al., 2021a], MAT [Wen, et al., 2022] ? More importantly, could the study provide the comparison experiments with HATRPO and HAPPO algorithms [Kuba, et al., 2021a], MAT [Wen, et al., 2022] ?
3. In the experimental section, Why is the performance of parameter sharing algorithms better than that of parameter non sharing? According to the the references [Kuba, et al., 2021a] and [Kuba, et al., 2021b], the the parameter sharing setting could be detrimental to the experimental performance.

[Gu, et al., 2021] Gu, S., Kuba, J. G., Wen, M., Chen, R., Wang, Z., Tian, Z., ... & Yang, Y. (2021). Multi-agent constrained policy optimisation. arXiv preprint arXiv:2110.02793.

[Wen, et al., 2022] Wen, M., Kuba, J. G., Lin, R., Zhang, W., Wen, Y., Wang, J., & Yang, Y. (2022). Multi-Agent Reinforcement Learning is a Sequence Modeling Problem. arXiv preprint arXiv:2205.14953.

[Kuba, et al., 2021a] Kuba, J. G., Chen, R., Wen, M., Wen, Y., Sun, F., Wang, J., & Yang, Y. (2021). Trust region policy optimisation in multi-agent reinforcement learning. arXiv preprint arXiv:2109.11251.

[Kuba, et al., 2021b] Kuba, J. G., Wen, M., Meng, L., Zhang, H., Mguni, D., Wang, J., & Yang, Y. (2021). Settling the variance of multi-agent policy gradients. Advances in Neural Information Processing Systems, 34, 13458-13470.


**Ethics Review Area:**

["I don’t know"]

**Limitations:**

The code is not found in the study, if it can be provided to examine the effectiveness of the method, that would be better.

**Strengths And Weaknesses:**

### Update after Rebuttal

I appreciate the authors' detail replies, and I have become largely persuaded by the authors and have raised my rating from 5 to 6.

Strengths: The theoretical part and the algorithms are good and sound solid.

Weaknesses:
1. In Page 4, line 167 and Page 7, line 253, the writing of the function may be not right, please check them.
2. In Page 5, line 198, line 199, the coefficient $\alpha$, $\beta$, $\rho$, $\gamma$ for the method's performance may need to be further investigated, ablation experiments may be necessary for the study.
3. Please explain the meaning of $g_{k}\left(\mathbf{x}^{r}, \mathbf{y}_{k}^{r}\right)$, currently, it's not clear.

---

> ### Author Response · Authors · 2022-08-02
> **Response to Questions**
>
> *Q1: In Page 6, Algorithm 1, how do the coefficient factors $\boldsymbol{\lambda}=\boldsymbol{\omega}_k$ improve the performance? Could the study provide the analysis?*
>
> **R1.** Please note that $\boldsymbol{\lambda}$ and $\boldsymbol{\omega}_k$ respectively denote the dual variables (Lagrange multipliers) for enforcing the UL and LL consensus constraints w.r.t. the primal variables $\mathbf{x}$ and $\mathbf{y}_k$. These two classes of dual variables are not equal as they are updated seperately with correspondence to different consensus processes.
>
> *Q2: Can the study be extended to some strong baselines, e.g. MACPO and MAPPO-L [Gu, et al., 2021], HATRPO and HAPPO algorithms [Kuba, et al., 2021a], MAT [Wen, et al., 2022] ? More importantly, could the study provide the comparison experiments with HATRPO and HAPPO algorithms [Kuba, et al., 2021a], MAT [Wen, et al., 2022] ?*
>
> **R2.** We truly thank reviewer 6ivX for bringing to our attention these recent related works on MARL.
>
> - First, we have added these references in the introduction part of this paper as follows.
>
> > ... the policy of each agent needs to be learned locally by certain efficient iterative methods, such as multi-agent deep deterministic policy gradient (MADDPG), trust region methods [Kuba, et al., 2021a], optimal baseline based variance reduced policy gradient [Kuba, et al., 2021b], and/or improved by more advanced techniques,  e.g., constrained policy optimization [Gu, et al., 2021] and large sequence models [Wen, et al., 2022].
>
> Also, please check the introduction section in our revised version of this paper.
>
> - We also further consider two algorithms based on proximal policy optimization (PPO), denoted as SLAM-PPO and Dec-PPO. Both of them apply PPO instead of the vanilla policy gradient to update their actors (upper-level parameters). In the updates of the critic, the difference is that SLAM-PPO uses SLAM-L to minimize the temporal difference (TD) errors while Dec-PPO directly utilizes the decentralized gradient descent for policy evaluation. We test these two algorithms on the navigation task, where the $\epsilon$ clip threshold in PPO is set as 0.2. It can be observed from Figure 5 that SLAM-PPO converges to higher reward values with less variance compared with DEC-PPO, which is similar to the numerical results with employing policy gradient updates for the actor networks.
>
> https://i.imgur.com/L5JWsI5.png (or please see Figure 5 in the appendix)
>
> *Q3: In the experimental section, Why is the performance of parameter sharing algorithms better than that of parameter non sharing? According to the the references [Kuba, et al., 2021a] and [Kuba, et al., 2021b], the the parameter sharing setting could be detrimental to the experimental performance.*
>
>
> **R3.** That's a good point. The parameter sharing strategy is inspired by the  ANIL type of multi-task learning techniques. Please kindly note that here we adopt the partial parameter sharing strategy rather than the fully sharing one. The partially shared parameters are used for extracting the permutation invariant latent space that is reusable across the network. We suppose that if the features of MARL tasks exist some shared latent space the partial parameter sharing method would improve the generalization performance of the model. In our numerical results, there seems no significant improvement benefited from the parameter sharing strategy.
>
>
> >[Gu, et al., 2021] Gu, S., Kuba, J. G., Wen, M., Chen, R., Wang, Z., Tian, Z., ... & Yang, Y. (2021). Multi-agent constrained policy optimisation. arXiv preprint arXiv:2110.02793.
>
> >[Wen, et al., 2022] Wen, M., Kuba, J. G., Lin, R., Zhang, W., Wen, Y., Wang, J., & Yang, Y. (2022). Multi-Agent Reinforcement Learning is a Sequence Modeling Problem. arXiv preprint arXiv:2205.14953.
>
> >[Kuba, et al., 2021a] Kuba, J. G., Chen, R., Wen, M., Wen, Y., Sun, F., Wang, J., & Yang, Y. (2021). Trust region policy optimisation in multi-agent reinforcement learning. arXiv preprint arXiv:2109.11251.
>
> >[Kuba, et al., 2021b] Kuba, J. G., Wen, M., Meng, L., Zhang, H., Mguni, D., Wang, J., & Yang, Y. (2021). Settling the variance of multi-agent policy gradients. Advances in Neural Information Processing Systems, 34, 13458-13470.

---

> ### Author Response · Authors · 2022-08-02
> **Response to Weaknesses**
>
> We thank reviewer 6ivX for their helpful comments and questions.
>
> **Weaknesses:**
> *Q1: In Page 4, line 167 and Page 7, line 253, the writing of the function may be not right, please check them.*
>
> **R1** Many thanks for the careful reading. We have revised the notations $F_i()$ and $f()$ as $F_i(x_i, y^*_{i,1}(x_i), \ldots,y^*_{i,m}(x_i);\xi)$ and $f_i(x_i, y^*_{i,k}(x_i)),\forall i,k$. Please see the revised version of this paper.
>
> *Q2: In Page 5, line 198, line 199, the coefficient $\alpha$, $\beta$, $\rho$, $\gamma$ for the method's performance may need to be further investigated, ablation experiments may be necessary for the study.*
>
> **R2** $1/\alpha$ and $1/\beta$ are the stepsizes of updating sequences $\mathbf{x}^r$ and $\mathbf{y}^r$, while $\rho/\gamma$ is the step size of the dual update. In the revised version (in the appendix), we further tested the numerical robustness of these step sizes on both MARL and MAML problems. To be more specific, we keep the current step sizes as the standard ones (as we have already tuned them as large as possible) and then decrease them.
>
> - Decentralized MAML
> It is shown in Figure 7 that when the  step sizes become smaller, the convergence of SLAM-U is slower, which is consistent with the classic theory of numerical algorithms.
>
> https://i.imgur.com/HeWh3FT.png (or please see Figure 7 in the appendix)
>
>
> - MARL
>  From Figure 4, we can have the similar conclusion that small step sizes will result in slower convergence rates.
>
> https://i.imgur.com/PLwemCL.png (or please see Figure 4 in the appendix)
>
>
> *Q3: Please explain the meaning of $g_k(\mathbf{x}^r,\mathbf{y}^r_k)$, currently, it's not clear.*
>
> **R3** Function $g_k(\mathbf{x},\mathbf{y}_k)$ is defined in eq. (2c), and it represents the $k$th lower level loss function. $g_k(\mathbf{x}^r,\mathbf{y}^r_k)$ denotes the function evaluated at $(\mathbf{x}^r,\mathbf{y}^r_k)$.

---

### Official Review · Reviewer_r52d · 2022-07-25

**Rating:** 8
**Confidence:** 3
**Soundness:** 4 excellent
**Presentation:** 4 excellent
**Contribution:** 4 excellent

**Summary:**

This paper has studied a generic form of the DBO (Decentralized Bilevel Optimization) problem, which has three major variants (constraints on upper level consensus, lower level consensus and both) that encompass several popular machine learning problems, such as MAML and MARL. For this problem, the paper proposes SLAM (a stochastic linearized augmented Lagrangian method), a single-timescale and single-loop algorithm that can find the ϵ-KKT points at a rate of O(1/(nϵ^2)).

**Questions:**

No Question.

**Limitations:**

The authors have adequately addressed the limitations and potential negative societal impact of their work.

**Strengths And Weaknesses:**

Strength 1. This paper studies an important machine learning problem, whose formulation is generic enough to have MARL and MAML as its special cases.

Strength 2. Theoretically, the major novelty is an augmented-Lagrangian variant. The authors have derived recursion of the successive dual variables to quantify the consensus errors (in terms of primal variables) from both UL and LL optimizations. The proof results are shown in a compact manner, with sufficient explanation, which presents good readability.

Strength 3. On the emparical side, the authors tested the performance of SLAM numerically on a MARL scenario and found that SLAM outperformed the traditional MARL algorithms w.r.t. convergence speed and achievable rewards.

Weakness. The numerical results are gained from toy examples. Thus, for real scenarios, the scalability (in terms of the number of agents, the action space, etc.) of this method is questionable.

---

> ### Author Response · Authors · 2022-08-02
> **Response to Weaknesses**
>
> We thank reviewer r52d for your positive comments and recognizing the importance of this work, and for your helpful comments and questions.
>
> In the revised version of this paper, we further added **two new** numerical results of showing the scalability of SLAM.  To be more specific, we give the following observation and discussions.
>
> - Decentralized ANIL [Raghu, et al., 2020]
>
> This problem can be formulated as eq.(9) in the paper. Here, we partition the weights of the neural network at each agent as two parts denoted by $\mathbf{x}_i$ and $\mathbf{y}_i$, where the UL optimization problem is to extract the reusable latent space across the connected agents while the LL optimization problem is adopted for adaption of the local model to individual learning tasks, and formulate the problem given in eq. (9).
>
> In this numerical experiment, a two-layer neural network is used, where the numbers of neurons at the hidden and output layers are 32 and 10, respectively, and the activation function is sigmoid. A 2-norm regularization term with parameter 0.01 is added to the LL loss function. The communication topology is generated by a random Erdos--Renyi graph. We divide the MNIST dataset into $n$ parts, where each of them only includes 128 data samples for training. The initial UL and LL step sizes of SLAM-U are $0.01$ and $0.1$, and the mini-batch size for gradient estimate is 32.
>
> https://i.imgur.com/Mmxv9GV.png (or please see Figure 6 in the appendix)
>
>
> From the numerical results shown above, it can be observed that as the number of agents increases, SLAM-U converges faster in terms of the data samples passed. Setting a 93.5\% test accuracy as a threshold, we measure the number of data sampled passed w.r.t. the different numbers of agents and report the results in Table 2 and Figure 6 (b). If we assume that the computational evaluations of the stochastic gradient estimate are the same at each agent, Figure 6 (b) shows that there at least exists a linear speed-up w.r.t. the number of agents in terms of the computational workload, which is consistent with our theoretical analysis.
>
> https://i.imgur.com/Lx1x4cN.png (or please see Table 2 in the appendix.)
>
> - MARL
>
> It is well known that the variance of the multi-agent policy gradient estimate becomes larger as the number of agents increases [Kuba, et al., 2021b], implying that the difficulty of solving large-scale multi-agent problems rises. We use the same set of step sizes for each algorithm and test their performances by increasing the number of agents from 4 to 10. It can be observed in Figure 3 that SLAM-AC consistently performs well and converges stably with minimum variances compared with the other two existing methods. Note that:
> - The average reward achieved by the classic MARL algorithm, DAC, has higher variances and even diverges when the number of agents is increased to 10;
> - The reward obtained by MDAC is relatively stable compared to DAC as the multiple consensus steps in the inner loop reduces the variances, but it is still lower than SLAM-AC.
>
> Therefore, it is concluded that SLAM-AC scales better than the other two due to the averaged variance over the network.
>
> https://i.imgur.com/x97DmKr.png (or please see Figure 3 in the appendix.)
>
> > [Raghu, et al., 2020] A. Raghu, et al. Rapid learning or feature reuse? towards understanding the effectiveness of MAML, ICLR, 2020.
>
> >[Kuba, et al., 2021b] Kuba, J. G., Wen, M., Meng, L., Zhang, H., Mguni, D., Wang, J., & Yang, Y. (2021). Settling the variance of multi-agent policy gradients. Advances in Neural Information Processing Systems, 34, 13458-13470.

---

### Official Review · Reviewer_a2eA · 2022-07-25

**Rating:** 5
**Confidence:** 3
**Soundness:** 3 good
**Presentation:** 3 good
**Contribution:** 2 fair

**Summary:**

This paper extends the bilevel optimization to the decentralized setting to include the popular problems in the multi-task learning and multi-agent reinforcement learning. In particular, the upper and lower optimization variable in the bilevel optimization problem need to achieve a consensus currently and each node (agent) can just communicate with its neighbors (w.r.t to the graph). Comparing with previous work with two timescales and two loops, the author derive a single timescale and one loop algorithm inspired by the augmented lagrangian method developed in the single-level optimization (and its decentralized version). Under some standard assumption (generally the assumption in single-level decentralized optimization and centralized bilevel-optimization ), the author provides the theoretical guarantee of the algorithm, in particularly, the convergence to the \epsilon-KKT points.  At last, the authors provide an empirical study on MARL problem including cooperative navigation task and pursuit-evasion game.

**Questions:**

Please see the comments in Strengths And Weaknesses.

**Limitations:**

Yes

**Strengths And Weaknesses:**

Pros:

1. The formulation of the decentralized bilevel optimization problem may cover several interesting problem in machine learning community. I guess this work may provide some insights to the meta-learning and reinforcement learning.

2. The authors provide a theoretical analysis on the proposed algorithm.

Cons:
1. I am sorry to say that I do not check every detail of the proof. One of my concern is that the proof technique may just combine the existing results, e.g., the result on single-level decentralized optimization and single-agent bilevel optimization. Can authors elaborate the novelty of their proof?
2. The empirical result is not that convincing. For instance, SLAM-AC in figure 1 and figure 2 does not outperform the baseline a lot (the shaded areas in some figure are even overlapped. ). In addition, the baseline is not strong enough, although I think it may be OK for a theoretical paper.

3. g in the assumption is strongly convex. It seems that in the experiment the author use neural networks to approximate the actor and critic (in the appendix).  Does this assumption limit the applicability of the algorithm?

---

> ### Author Response · Authors · 2022-08-02
> **Response to Weaknesses**
>
> We thank reviewer a2eA for your helpful comments and questions.
>
> *Q1: I am sorry to say that I do not check every detail of the proof. One of my concern is that the proof technique may just combine the existing results, e.g., the result on single-level decentralized optimization and single-agent bilevel optimization. Can authors elaborate the novelty of their proof?*
>
> **R1.** The key novelties of the technical proof mainly lie in the following two parts:
>
> - (Stochastic gradient estimate error coupling among UL and LL optimization variables). As the stochastic errors of the gradient estimate resulting from the two levels are coupled under the consensus constraint, using the classical bounded variance type of quantities as the measure is not enough to show the convergence. Here, mean-related terms (i.e., squares of the mean between two successive iterations of both primal and dual variables) are also used. To be more specific, we are able to show that $\mathbb{E}||\boldsymbol{\lambda}^{r+1}-\boldsymbol{\lambda}^r||^2$ and $\mathbb{E}||\boldsymbol{\omega}^{r+1}_k-\boldsymbol{\omega}^r_k||^2$ are upper bounded in terms of two classes of errors: 1) mean-related terms: $||\mathbb{E}[\boldsymbol{\lambda}^{r+1}-\boldsymbol{\lambda}^r]||^2$, $||\mathbb{E}[\mathbf{x}^{r+1}-\mathbf{x}^r-(\mathbf{x}^{r}-\mathbf{x}^{r-1})]||^2$, $||\mathbb{E}[\mathbf{y}^{r+1}_k-\mathbf{y}^r_k-(\mathbf{y}^{r}_k-\mathbf{y}^{r-1}_k)]||^2$, and variance-related terms: 2) $\mathbb{E}||\mathbf{x}^{r+1}-\mathbf{x}^r||^2$, $\mathbb{E}||\mathbf{y}^{r+1}_k-\mathbf{y}^r_k||^2$, up to some noise terms.
>
> -  (Consensus constraint violation controlled by the augmented Lagrangian related term). It is shown that only performing a stochastic primal update on the linearized augmented functions with the dual variable updates is sufficient to realize the consensus at either both levels or any level of this class of decentralized bilevel optimization problems. Technically, it is proven that the consensus errors $\mathbb{E}||\mathbf{A}\mathbf{x}^r||$ and $\mathbb{E}||\mathbf{A}\mathbf{y}^r||^2$ can be upper bounded by the quadractic terms in the form of $\mathbb{E}[\mathbf{x}^{r+1}-\mathbf{x}^r-(\mathbf{x}^{r}-\mathbf{x}^{r-1})]$ and $\mathbb{E}[\mathbf{y}^{r+1}_k-\mathbf{y}^r_k-(\mathbf{y}^{r}_k-\mathbf{y}^{r-1}_k)]$ that are built upon the augmented Lagrangian, where the construction of the merit/potential function in showing the convergence of the SLAM iterates is unique.
>
> To the best of our knowledge,  only with the support of these new theoretical results, we are able to develop this single-loop and communication-efficient bilevel optimization method while being able to achieve the same convergence rate as the classic stochastic gradient descent for solving only single-level general nonconvex problems.
>
> *Q2: The empirical result is not that convincing. For instance, SLAM-AC in figure 1 and figure 2 does not outperform the baseline a lot (the shaded areas in some figure are even overlapped. ). In addition, the baseline is not strong enough, although I think it may be OK for a theoretical paper.*
>
> **R2.** The MARL problem is a hard one comparing with the classic decentralized training problems as the variance of policy gradient grows linearly w.r.t. the number of agents [Kuba, et al., 2021b].
>
> From our new numerical results on scalability of the MARL algorithms, it is shown that SLAM-AC converges more stably compared with the other two methods, especially when the number of agents is 10.
>
> https://i.imgur.com/x97DmKr.png (or please see Figure 3 in the appendix)
>
>
> Also, in the revised version of our work, we further integrate the stronger baseline, proximal policy optimization (PPO), as the optimizer for the update of the actor network into our proposed SLAM to improve the performance of the MARL system.
>
> https://i.imgur.com/lrp4WU0.png (or please see Figure 5 in the appendix)
>
> *Q3: g in the assumption is strongly convex. It seems that in the experiment the author use neural networks to approximate the actor and critic (in the appendix). Does this assumption limit the applicability of the algorithm?*
>
> **R3.** We agree that this assumption would constrain the applicability of the obtained theoretical guarantees, but the proposed framework of this decentralized bilevel algorithm is still general. In practice, if the critic network is over parameterized or some regularization term is added, the lower level loss function may satisfy some certain property, e.g., PL condition, one-point strong monotonicity, or variational inequality, so that the gradient type of algorithms can still work well numerically and even theoretically, which will be left as future works.

---

### Official Review · Reviewer_wDpj · 2022-07-29

**Rating:** 7
**Confidence:** 2
**Soundness:** 3 good
**Presentation:** 3 good
**Contribution:** 3 good

**Summary:**

The paper proposes an algorithm for decentralized bilevel optimization, that converges to $\epsilon$-KKT points at a rate of $\mathcal{O}(1/(n\epsilon^2))$, where $n$ is the number of nodes. This is a linear speedup in $n$ compared to prior work. The algorithm is evaluated empirically on a navigation task and a pursuit-evasion game.

**Questions:**

- Could you comment more on practical considerations to aid judging the efficiency improvement? [** post-rebuttal ** : partially addressed in rebuttal via new experiments]
- Could you give some more intuition on what leads to higher rewards in existing experiments? [** post-rebuttal ** : addressed]
- It somewhat seems to me that the pursuit experiment shows faster convergence, while the navigation experiment shows convergence to better reward. Is this correct? If so, could you explain more what the intuition is behind the distinction in behavior? [** post-rebuttal ** : addressed]

**Limitations:**

- I believe the paper would benefit from more numerical experiments so as to capture wider range of behavior. [** post-rebuttal ** : addressed via two new experiments -- see rebuttal + new version of paper]

- Small comment: formatting of paragraph "Applications of Bilevel Optimization" (lines 28-63) is unpolished in my opinion.  Especially "RL:" after the introductory sentence on 53-55 feels a bit out of place/abrupt, similarly "Meta-learning:" after first sentence in 28-31.

**Strengths And Weaknesses:**

Strengths:
- paper is clear
- inclusion of empirical behavior is nice
- the problem studied is very relevant to current trends in other subfields of ML (e.g. multi-agent RL, multi-task learning)

Weaknesses:
- it is unclear to me in general how much of a difference the linear speed-up w.r.t. $n$ makes. Since this is a main novel contribution, I think it would be nice to have more in-depth discussion (with concrete examples), as well as a discussion of how tight the hidden rate dependencies of the proposed algorithm are. [** post-rebuttal ** : partially addressed: two new experiments added]

---

> ### Author Response · Authors · 2022-08-02
> **Response to Questions**
>
> *Q1: Could you comment more on practical considerations to aid judging the efficiency improvement?*
>
> **R1**  There are at least two aspects that can be concluded for improving the efficiency based on the above **two new** numerical experiments:
>
> - In applications of supervised learning problems, a linear speed-up achieved by SLAM w.r.t. the number of agents can save computational workload or gradient evaluations per agent.
> - For MARL problems, SLAM is able to achieve high reward values with less variance and converge stably in terms of hyperparameter tuning.
>
> *Q2: Could you give some more intuition on what leads to higher rewards in existing experiments?*
>
> **R2** By leveraging the primal-dual update on the linearized augmented Lagrangian, SLAM-AC can control the trade-off between the two levels of the optimization and consensus process in a balanced way so that it is able to provide more accurate value function estimations under the (LL) consensus constraint in comparison with existing works.
>
> *Q3: It somewhat seems to me that the pursuit experiment shows faster convergence, while the navigation experiment shows convergence to better reward. Is this correct? If so, could you explain more what the intuition is behind the distinction in behavior?*
>
> **R3** The criteria for calculating the rewards between the two environments are a little different. In the navigation experiment, the rewards are computed based on the distance between the destination and agents, which are continuous variables. In the pursuit experiment, the rewards are computed based on the fact that any two pursuers simultaneously arrive at the evader’s location, which is not continuous in terms of distance. So, the achieved rewards in the pursuit experiment are not as sensitive as the ones in the navigation experiments.
>
> Even though  SLAM can perform policy improvement by using more accurate function value estimations over the consensus network compared with the other methods intuitively, SLAM only shows the advantage of a faster convergence in this experiment. In the appendix (D.3), we further adjusted the step sizes to be smaller and obtain the similar results as shown in the navigation experiment in the sense that SLAM-AC has both faster convergence and higher rewards.

---

> > ### Comment · Reviewer_wDpj · 2022-08-08
> > **Thank you!**
> >
> > Thank you for the replies and clarifications! Regarding Q3, could you maybe move the appendix (D.3) that has both faster convergence and higher rewards to the main paper? Regarding Q1, the new experiments are great and will strengthen the paper! (see my other reply)

---

> > > ### Author Response · Authors · 2022-08-09
> > > **Many thanks for your suggestion**
> > >
> > > We are grateful for your suggestion and recognition of the significance of the new numerical results. In the updated version, we have moved the results that show the superiority of SLAM-AC with both faster convergence and higher achievable rewards from the appendix to the main content of this paper.

---

> ### Author Response · Authors · 2022-08-02
> **Response to Weaknesses**
>
> We thank reviewer wDpj for your helpful comments and questions.
>
> **Weaknesses:**
>
> it is unclear to me in general how much of a difference the linear speed-up w.r.t. $n$ makes. Since this is a main novel contribution, I think it would be nice to have more in-depth discussion (with concrete examples), as well as a discussion of how tight the hidden rate dependencies of the proposed algorithm are.
>
> **Response:** Achieving an $\mathcal{O}(\frac{1}{\sqrt{nT}})$ convergence rate means the total complexity of finding an $\epsilon$-approximate first-order stationary point of general nonconvex problems is upper bounded by $\mathcal{O}(\frac{1}{\epsilon^2})$, which indicates that the computational load (computation of stochastic gradient counts) per agent is $\mathcal{O}(\frac{1}{n\epsilon^2})$ [X. Lian, et al. 2017]. In other words, the equivalent total variance of the stochastic gradient estimate is reduced w.r.t. the number of agents.
>
> In the revised version of our paper, we have provided a **new** concrete example of decentralized ANIL [Raghu, et al. 2020] for demonstrating: 1) the linear speed-up that SLAM can numerically achieve on a real dataset; and 2) the tightness of the convergence rate bound obtained from our theoretical analysis. We also give **new** numerical results regarding the stability of SLAM-AC in applications of MARL, which further strengthens the variance improvement that can be achieved by our proposed learning framework. To be more specific, we give the following observation and discussions.
>
> - Decentralized ANIL [Raghu, et al. 2020]
>
> This problem can be formulated as eq.(9) in the paper. Here, we partition the weights of the neural network at each agent as two parts denoted by $\mathbf{x}_i$ and $\mathbf{y}_i$, where the UL optimization problem is to extract the reusable latent space across the connected agents while the LL optimization problem is adopted for adaption of the local model to individual learning tasks, and formulate the problem given in eq. (9).
>
> In this numerical experiment, a two-layer neural network is used, where the numbers of neurons at the hidden and output layers are 32 and 10, respectively, and the activation function is sigmoid. A 2-norm regularization term with parameter 0.01 is added to the LL loss function. The communication topology is generated by a random Erdos--Renyi graph. We divide the MNIST dataset into $n$ parts, where each of them only includes 128 data samples for training. The initial UL and LL step sizes of SLAM-U are $0.01$ and $0.1$, and the mini-batch size for gradient estimate is 32.
>
> https://i.imgur.com/Mmxv9GV.png (or please see Figure 6 in the appendix)
>
>
> From the numerical results shown above, it can be observed that as the number of agents increases, SLAM-U converges faster in terms of the data samples passed. Setting a 93.5\% test accuracy as a threshold, we measure the number of data sampled passed w.r.t. the different numbers of agents and report the results in Table 2 and Figure 6 (b). If we assume that the computational evaluations of the stochastic gradient estimate are the same at each agent, Figure 6 (b) shows that there at least exists a linear speed-up w.r.t. the number of agents in terms of the computational workload, which is consistent with our theoretical analysis.
>
> https://i.imgur.com/Lx1x4cN.png (or please see Table 2 in the appendix.)
>
> - MARL
>
> It is well known that the variance of the multi-agent policy gradient estimate becomes larger as the number of agents increases [Kuba, et al., 2021b], implying that the difficulty of solving large-scale multi-agent problems rises. We use the same set of step sizes for each algorithm and test their performances by increasing the number of agents from 4 to 10. It can be observed in Figure 3 that SLAM-AC consistently performs well and converges stably with minimum variances compared with the other two existing methods. Note that:
> - The average reward achieved by the classic MARL algorithm, DAC, has higher variances and even diverges when the number of agents is increased to 10;
> - The reward obtained by MDAC is relatively stable compared to DAC as the multiple consensus steps in the inner loop reduces the variances, but it is still lower than SLAM-AC.
>
> Therefore, it is concluded that SLAM-AC scales better than the other two due to the averaged variance over the network.
>
> https://i.imgur.com/x97DmKr.png (or please see Figure 3 in the appendix.)
>
>
> > [Lian, et al., 2017] X. Lian, et al. Can decentralized algorithms outperform centralized algorithms? a case study for decentralized parallel stochastic gradient descent, NeurIPS, 2017.
>
> > [Raghu, et al., 2020] A. Raghu, et al. Rapid learning or feature reuse? towards understanding the effectiveness of MAML, ICLR, 2020.
>
> >[Kuba, et al., 2021b] Kuba, J. G., Wen, M., Meng, L., Zhang, H., Mguni, D., Wang, J., & Yang, Y. (2021). Settling the variance of multi-agent policy gradients. Advances in Neural Information Processing Systems, 34, 13458-13470.

---

> > ### Comment · Reviewer_wDpj · 2022-08-08
> > **Thank you!**
> >
> > Thank you for thoroughly describing and presenting the new experiments. This does address some of my main reservations so I increased my contribution score from 2 to 3, and my general score from 6 to 7.

---

> > > ### Author Response · Authors · 2022-08-09
> > > **Many thanks for your positive feedback to our response**
> > >
> > > We truly thank you for recognizing the contributions of this work and increasing the scores.

---

### Meta-Review · Area_Chair_sFXg · 2022-09-01

**Recommendation:** Accept
**Confidence:** Certain

**Metareview:**

This paper proposes a new algorithm for distributed bilevel optimization with solid theoretical results and empirical validation in multi-agent RL examples. During the rebuttal/discussion period, authors were able to provide additional experiments and addressed most of reviewers' concerns. All are in favor for acceptance and appreciate the combination of theory and strong experiments.

**Award:**

No

---

### Decision · Program_Chairs · 2022-09-14

Accept